# RiskZero: Plan More to Risk Less with a Learned Model

Yousef Yassin [1]    Junfeng Wen [1]

## Abstract

*AlphaZero* and *MuZero* have demonstrated super-human performance across a range of strategic tasks. Yet their reliance on maximizing expected returns limits their use in real-world settings, where even high-return policies may incur rare but catastrophic failures. We introduce *RiskZero* to address this limitation; the first *MuZero*-family method for risk-sensitive decision-making, and planning with *zero* prior knowledge of environment dynamics. *RiskZero* learns distributional quantities to estimate trajectory-level risk, guiding search toward policies that explicitly avoid rare but severe outcomes. We establish theoretical convergence to optimal, stationary risk-sensitive policies and validate our approach on environments designed to test risk-sensitive learning from pixels, as well as on larger-scale combinatorial tasks. Across all settings, *RiskZero* consistently outperforms state-of-the-art risk-sensitive baselines, and improves sample efficiency, providing a general framework for safer and reliable model-based reinforcement learning under uncertainty.

## 1. Introduction

We make decisions in an uncertain world, but we need not be uncertain about the decisions we make. Faced with the unknown, humans plan to guide behavior by considering their options and anticipating possible futures. They plan to *reduce risk*, and to increase the likelihood of achieving their goals. Planning, in this sense, is not only for seeking reward but also for *how much* one desires avoiding the undesirable.

This capacity for informed decision-making has inspired extensive research into integrating planning within intelligent agents (Schrittwieser et al., 2020; Zhao et al., 2021). Tree-based planning, in particular, has achieved remarkable success, surpassing human performance in domains such as Chess, Atari (Silver et al., 2016; 2017), and Poker (Brown & Sandholm, 2018). Meanwhile, the demand for *robust* decision-making under uncertainty has driven growing interest in *risk-sensitive* policy learning (Dabney et al., 2018a; Fei et al., 2021), which seeks policies that optimize explicitly for risk rather than expected return, often by modeling the full *distribution* of future returns. Bridging these two directions—planning *with* the return distribution—opens an exciting avenue toward risk-aware agents better suited to the challenges of safe, real-world decision-making.

*AlphaZero* showed that a single algorithm combining search with deep reinforcement learning (RL) can achieve superhuman performance across diverse complex tasks (Silver et al., 2017). However, it relied on a black-box model of the environment. *MuZero* generalized this idea by learning its own world model, enabling planning even when dynamics are unknown, complex, or costly to simulate (Schrittwieser et al., 2020). Despite these advances, both approaches remain tied to the standard RL objective of maximizing expected return, a criterion that often falls short in real-world settings.

Expected return can fail because the average is intrinsically risk-neutral. It assumes recovery is possible and that multiple trials are available—assumptions that rarely hold in practice (Du et al., 2022). Many real-world applications are instead sensitive to even a single failure, where success hinges on a single policy execution (Hayes et al., 2023). In such settings, risk-neutral policies are inadequate, as a high expected return can still mask catastrophic outcomes arising from stochasticity. Hence, our approach, like others, is motivated by the recognition that expectation, a single value, hides risk by averaging over rare but severe events.

Risk-sensitive RL (RSRL) shifts focus from maximizing expected return to mitigating *worst-case* outcomes by optimizing alternative criteria such as entropic risk (Borkar, 2002; Fei et al., 2021), value-at-risk (VaR) (Chow et al., 2018a; Jung et al., 2022), and conditional value-at-risk (CVaR) (Tamar et al., 2015), which capture both the frequency and severity of rare adverse events. Among these, coherent risk measures such as CVaR are preferred as they enjoy compelling theoretical properties (Artzner et al., 1999). However, optimizing such criteria can incur substantial computational challenges, confining most RSRL methods to relatively simple environments. In practice, distributional

Code: https://github.com/yyassin/riskzero
[1]Carleton University, Ottawa, Canada. Correspondence to: Yousef Yassin <yousef.yassin@carleton.ca>.

*Proceedings of the 43ʳᵈ International Conference on Machine Learning*, Seoul, South Korea. PMLR 306, 2026. Copyright 2026 by the author(s).

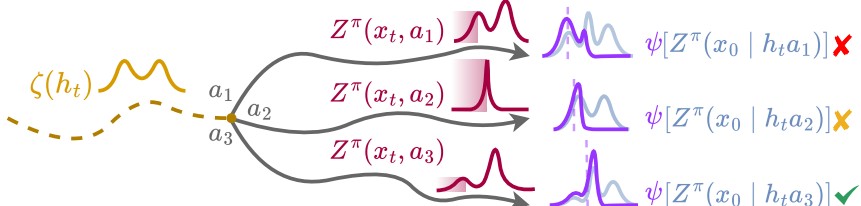

*Figure 1.* **Unbiased risk-aversion requires the trajectory,** combining the *accumulated return distribution* with each candidate action's *future return distribution* $Z^\pi(x_t, a_i)$ for action selection. Naive consideration of *future risk* alone, by applying $\psi$ to $Z^\pi(x_t, a_i)$ (i.e., expectation of the shaded region), misidentifies $a_2$ as safest. Proper evaluation considers the full *trajectory return distribution*, revealing $a_3$ as risk-optimal. Risk is computed as the expectation (dashed lines) of a *distorted trajectory return* that emphasizes adverse outcomes.

RL is commonly used as access to distributions facilitates risk estimation. Early approaches optimized greedily with respect to future risk (Dabney et al., 2018a; Urpí et al., 2021; Shen et al., 2023), but were later shown to be biased and lack convergence guarantees (Lim & Malik, 2022). More recent methods (Zhou et al., 2023) remove this bias by modeling trajectory-level risk, but at the expense of sample efficiency.

We address these challenges with **RiskZero**, a novel model-based extension of *MuZero* that learns distributional rewards, and *future returns*, together with the distribution of *accumulated returns*, to enable *trajectory-level* risk estimation and planning under arbitrary coherent *risk-measures* such as CVaR (Fig. 1). We first show that ❶ greedy policies suffice for optimal decision-making under coherent risk measures and that ❷ risk-sensitive objectives admit unbiased expectation-based surrogates, yielding ❸ a general framework for risk-sensitive planning. Building on these insights, we ❹ introduce the first *MuZero*-family method capable of risk-sensitive planning and ❺ provide theoretical guarantees establishing convergence to a class of optimal risk-sensitive policies. Empirically, we show that ❻ *RiskZero* achieves state-of-the-art performance across multiple risk measures on domains where risk aversion is salient, ranging from a discrete mini-grid to image-based environments (a modified Space Invaders task from MinAtar (Young & Tian, 2019)) and large discrete combinatorial problems (stochastic Maximum Independent Set and Bipartite Matching), while substantially improving sample efficiency and consistency.

## 2. Preliminaries

As a prerequisite to our risk-sensitive approach, we begin by reviewing the notations and foundations of risk in RL.

**Agent Interaction Model.** We consider a finite-horizon Markov decision process (MDP) (Puterman, 2014) of length $T$ with state space $\mathcal{X}$, discrete action space $\mathcal{A}$, random rewards $R(x, a)$, transition probabilities $P(\cdot \mid x, a)$, and discount factor $\gamma \in (0, 1]$. Following prior work in planning, we focus on deterministic dynamics $M$, denoted by $x_{t+1} = M(x_t, a_t)$ where $t$ indexes time steps. A trajectory $\tau \in \mathcal{T}$ is a sequence $\tau = x_0 a_0 x_1 \ldots a_{T-1} x_T$ generated by the agent's environment interactions. Each state-ending prefix

defines a history $h_t = x_0 a_0 x_1 \ldots a_{t-1} x_t$, and we denote the space of all histories by $\mathcal{H}$. In the deterministic setting, we use $h_t a$ and $h_t a x'$ interchangeably, where $x' = M(x_t, a)$.

The agent acts according to a policy $\pi : \mathcal{H} \to \Delta_{\mathcal{A}}$ that specifies a distribution over actions conditioned on the history. The return at time $t$ is $Z^\pi(x_t) = \sum_{i \geq 0} \gamma^i R(x_{t+i}, a_{t+i})$ so that $Z^\pi(x_0)$ represents the total random return from the initial state under policy $\pi$. Standard (risk-neutral) RL seeks to maximize the expected return, $\max_\pi \mathbb{E}[Z^\pi(x_0)]$. In contrast, our objective is to mitigate adverse outcomes by maximizing a risk-sensitive criterion, e.g., CVaR, of $Z^\pi(x_0)$.

**Quantifying Risk.** For a bounded random variable $Z$ with cumulative distribution function $F_Z(z) = \mathbb{P}(Z \leq z)$, we denote by $F_Z^{-1}(\omega) = \inf_z\{z \mid \omega \leq F_Z(z)\}$, $\omega \in [0, 1]$ its generalized inverse (i.e., quantile function). Adopting the RL convention of interpreting $Z$ as *gain* (return), risk measures then quantify performance in the presence of adversity.

Yaari (1987) showed that risk preferences over $Z$ admit utility representations as *distorted expectations* of outcomes

$$\psi[Z] = \mathbb{E}_{\omega \sim g'_\psi}[F_Z^{-1}(\omega)] = \int_0^1 F_Z^{-1}(\omega) dg_\psi(\omega), \quad (1)$$

where $g_\psi : [0, 1] \to [0, 1]$ is a monotone non-decreasing *distortion function*. Rather than modifying outcome values directly, $g_\psi$ reshapes cumulative probabilities, thereby encoding attitudes toward risk (Wang, 1996), as illustrated by the distorted distributions in Fig. 1 (right). This formulation generalizes the risk-neutral expectation, recovered when $g_\psi$ is the identity. In this work, we focus on *coherent* risk measures, satisfying axioms widely regarded as desirable for rational risk-sensitive decision-making (Artzner et al., 1999). A canonical example is $\alpha$-CVaR[1], obtained by choosing $g_\psi(\omega) = \min\{\omega/\alpha, 1\}$. Further details are in App. A.3.

**Problem Formulation.** Our interest lies in finding a risk-aware policy that maximizes the risk-sensitive utility

$$\max_\pi \ \psi[Z^\pi(x_0)], \quad (2)$$

capturing a *static* notion of risk by evaluating across the *full trajectory*. An alternative line of work considers *dynamic*

---

[1]We adopt the convention that $\alpha$-CVaR corresponds to the *left* tail, averaging the worst $\alpha\%$ of outcomes; others may use $1-\alpha\%$.

or *nested* risk (Shapiro et al., 2021), assessed recursively at each time step. While theoretically appealing, dynamic risk is harder to interpret and leads to overly optimistic or pessimistic behavior; hence, the static formulation is more common in practice (Lim & Malik, 2022; Hau et al., 2023).

As in MDPs under expectation, prior work on risk-sensitive decision-making often uses *greedy* policies, e.g., (Lim & Malik, 2022; Zhou et al., 2023; Moghimi & Ku, 2025) In Prop. 3.1, we show more directly that coherence is sufficient for the existence of *optimal* greedy policies. Still, *discovering* such policies remains algorithmically challenging as coherent risk measures are not generally *time-consistent*; optimal decisions may depend on realized returns, substantially enlarging the effective state space and rendering learning less practical. To improve tractability, we follow prior work (Lim & Malik, 2022; Zhou et al., 2023) and restrict attention to *stationary* policies, which assign the same action preferences to identical inputs. Traditional RL considers policies stationary with respect to *state*, and so do not consider realized returns (see App. A.4 for further discussion). However, even under this restriction, nonlinearity of $\psi$ means that learning remains inherently non-Markovian as the solution to (2) does not generally align with maximizing a local notion of *future* risk at each step (Zhou et al., 2023).

**Trajectory-level Risk.** *RiskZero* sidesteps this difficulty by shifting the lens from stepwise risk to *trajectory-level* return distributions. For a given state–action pair, the distributional perspective defines the random return as $Z^\pi(x_t, a_t) = \sum_{t \geq 0} \gamma^i R(x_{t+i}, a_{t+i})$ where $x_{t+1} \sim P(\cdot|x_t, a_t)$ and $a_t \sim \pi(\cdot \mid x_t)$ (Bellemare et al., 2017), leading to a distributional analog of the Bellman equation

$$Z^\pi(x_t, a_t) \stackrel{D}{=} R(x_t, a_t) + \gamma Z^\pi(X', A'), \qquad (3)$$

with $X' \sim P(\cdot|x_t, a_t)$, $A' \sim \pi(\cdot|X')$, and $\stackrel{D}{=}$ denoting equality in distribution. In practice, we parameterize $Z^\pi$ by quantiles learned via quantile temporal-difference (QTD) regression (Rowland et al., 2024) (see App. A.1 for further discussion).

While empirically effective, prior work shows that optimizing only for *future risk* leads to biased optimization, failing to improve risk overall (Lim & Malik, 2022; Zhou et al., 2023). *RiskZero* accordingly adopts a broader view on risk, building on *Trajectory Q-Learning* (TQL), and explicitly conditions on the history—hence, on the *distribution* of accumulated returns—to estimate and optimize *trajectory-level risk* (Zhou et al., 2023). Rather than optimize future risk, TQL instead proposes a *history*-relied operator $\mathcal{T}^*_{\psi,h}$

$$\mathcal{T}^*_{\psi,h} Z^\pi(h_t, a) \stackrel{D}{=} \sum_{i=0}^{t} \gamma^i R(x_i, a_i) + \gamma^{t+1} Z^\pi(x', A'), \quad (4)$$
$$A' = \arg\max_{a'} \psi[Z^\pi(h_{t+1}, a')], \qquad (5)$$

with $h_{t+1} = h_t a_t x'$ and $x' = M(x_t, a_t)$, resulting in policies stationary with respect to *history*. We will call such policies *history*-stationary going forward. Notably, the greedy

selection in (5) optimizes risk along the *trajectory*, rather than the *future* alone, enabling dynamic-programming (DP) updates that directly optimize (2) and converge to optimal risk-sensitive policies. However, convergence is not guaranteed from arbitrary $Z^\pi$ initializations, and history conditioning induces state-space growth that renders TQL sample-inefficient in complex environments (Wang et al., 2024a).

**MuZero.** *RiskZero* overcomes these limitations by combining trajectory-level risk modeling with planning, extending *MuZero* (Schrittwieser et al., 2020). *MuZero* learns an abstract predictive MDP of the environment and plans in this abstract state space via Monte Carlo tree search (MCTS) (Kocsis & Szepesvári, 2006). To avoid ambiguity, we use subscripts $t$ to index real environment timesteps and superscripts $k$ for unrolled steps in the abstract MDP, with $\Delta_A$ denoting the set of probability distributions over a set $A$.

Five components parameterize the abstract MDP. A *representation* model $h_\phi : \mathcal{H} \to \mathcal{S}$ maps a real history $h_t$ to an abstract root state $s_t^0$. A *dynamics* model $g_\phi : \mathcal{S} \times \mathcal{A} \to \mathcal{S}$ predicts successor abstract states, while a *reward* model $r_\phi : \mathcal{S} \times \mathcal{A} \to \mathbb{R}$ predicts immediate rewards. A *policy* model $\pi_\phi : \mathcal{S} \to \Delta_A$ produces action distribution priors, and a *value* model $v_\phi : \mathcal{S} \to \mathbb{R}$ estimates state values.

At each real timestep $t$, MCTS is performed in the abstract MDP starting from $s_t^0$, producing a search policy $\pi_t$ used to select the real action $a_t$. Training aims to align planning in the abstract MDP with real-world planning, such that search heuristically improves upon the policy prior. In practice, however, models are noisy, and shallow search often fails to produce meaningful improvement (Danihelka et al., 2022).

**Planning with Gumbel.** To obtain convergence guarantees for *RiskZero*, we employ *Gumbel*-based policy improvement in place of heuristic action selection. *Gumbel MuZero* (Danihelka et al., 2022) modifies root action selection in MCTS via the Gumbel Top-$K$ trick (Vieira, 2014; Kool et al., 2019). At non-root nodes, action values for unvisited children are *completed* via interpolation, after which actions are selected deterministically by matching visit counts $N(a)$ to the improved policy (Grill et al., 2020). This procedure guarantees policy improvement in expectation when action-value estimates are accurate, enabling strong performance with few simulations. While this guarantee is inherently risk-neutral, we show next that it suffices for learning risk-sensitive policies when values reflect trajectory-level risk.

## 3. Risk-Sensitive Policy Improvement

We seek to leverage *Gumbel*'s policy improvement guarantee to efficiently learn risk-sensitive policies. Doing so is not immediately clear as the guarantee holds in expectation whereas our objective is risk-sensitive. We resolve this tension by our adoption of a trajectory-level view on risk.

This section presents a general framework showing that, despite the absence of per-iteration guarantees in true risk, optimizing expected risk suffices to recover optimal risk-sensitive policies under mild conditions. This retains the tractability and convergence of expectation-based policy improvement while targeting risk-sensitive optimality. The framework here will be concretely instantiated in Sec. 4.

We start by showing that coherence admits deterministic optimal policies for the risk-sensitive objective in (2),

> **Proposition 3.1.** *Let $\psi$ be a coherent risk measure over the return $Z^\pi(x_0)$ of a finite MDP. Then there exists an optimal deterministic policy $\pi^\star$ such that*
>
> $$\psi[Z^{\pi^\star}(x_0)] = \max_\pi \psi[Z^\pi(x_0)].$$

A proof is provided in App. E.1. By this result, we may restrict attention to greedy optimal policies, and later show that our approach converges to such policies by its updates. Principled greed must apply at the *trajectory* level, motivating a definition for the history-conditioned trajectory return

$$Z^\pi(x_0 \mid h_t) \stackrel{D}{:=} \sum_{i=0}^{t-1} \gamma^i R(x_i, a_i) + \gamma^t Z^\pi(x_t). \quad (6)$$

**Expected Risk.** A policy $\pi$ naturally induces a distribution over possible trajectories in the environment, each with its own trajectory risk due to reward stochasticity. Accordingly, we may then consider the *expected risk* of such a policy

$$v_\psi^\pi(x_0) := \sum_\tau d^\pi(\tau) v_\psi^\pi(\tau), \quad v_\psi^\pi(\tau) := \psi[Z^\pi(x_0 \mid \tau)], \quad (7)$$

where $d^\pi(\tau)$ is the probability of observing $\tau$ under $\pi$. Prop. 3.1 shows that optimizing the objective in (2) reduces to identifying a trajectory $\tau$ that attains optimal risk. Our key insight is that optimizing expected risk is sufficient to do so.

We define the risk action-value and value of a history $h$ as

$$q_\psi^\pi(h, a) := v_\psi^\pi(ha), \quad v_\psi^\pi(h) := \sum_a \pi(a \mid h) q_\psi^\pi(h, a), \quad (8)$$

where we again use $h_t a$ and $h_t a x'$ interchangeably in deterministic settings, with $x' = M(x_t, a)$. Then, conditioned on $h$, the policy $\pi$ can be understood to induce a distribution over trajectory *completions*, and unrolling this recursion recovers (7). We note that $v_\psi^\pi(h)$ is *not* obtained by applying $\psi$ to the mixture distribution $Z^\pi(x_0 \mid h)$. Rather, it corresponds to the expectation, under $\pi$, of $\psi[Z^\pi(x_0 \mid \tau)]$, taken over all $h$-prefixed trajectories; distinguishing the risk of the expected trajectory from the expectation of risk. Despite this distinction, we will next show that the latter serves as a practical and effective surrogate for optimizing the former.

**Surrogate Risk-Sensitive Objective.** Directly optimizing $\psi[Z^\pi(x_0)]$ is challenging (see Sec. 6) and fails to scale beyond simple environments. Contrary to expectations, increasing the probability of actions with higher risk-sensitive action-values $q_\psi^\pi$ does not, in general, improve overall risk.

> **Proposition 3.2.** *Let $\pi$ and $\pi'$ be policies. An improvement in expected risk, $v_\psi^{\pi'}(x_0) \geq v_\psi^\pi(x_0)$, does not imply true risk improvement, $\psi[Z^{\pi'}(x_0)] \geq \psi[Z^\pi(x_0)]$.*

See App. E.2 for a proof. This limitation may seem to weaken the usefulness of expected risk at first glance. However, Prop. 3.2 only rules out a guarantee of *per-iteration* improvement in $\psi$; it does not preclude eventual improvement or convergence to the optimal risk-sensitive policy. In fact, for stationary policies, we show that optimizing (2) reduces to optimizing expected risk—the two problems are equivalent and share a deterministic optimum.

> **Proposition 3.3.** *Let $\psi$ be a coherent risk measure and $\Pi_H$ the class of history-stationary policies. A deterministic policy $\pi^* \in \Pi_H$ is optimal in expected risk, $\pi^* \in \arg\max_{\pi \in \Pi_H} v_\psi^\pi(x_0)$, if and only if it is optimal in true risk, $\pi^* \in \arg\max_{\pi \in \Pi_H} \psi[Z^\pi(x_0)]$.*

Thus, procedures converging to deterministic policies under expected risk can serve as surrogate optimizers for true risk. This fact follows by dropping the expectation at convergence in deterministic environments (see App. E.3 for a proof).

**Risk Optimization with Gumbel.** Concretely, we use *Gumbel MuZero* as a surrogate optimizer to produce *RiskZero*, which applies wherever *MuZero* does and can, in principle, plan for *safer* decisions under various risk measures.

We describe how to estimate risk-sensitive action-values in Sec. 4; here, we assume them given. Planning then closely follows *Gumbel MuZero*. At the root, $m \leq |\mathcal{A}|$ actions are sampled without replacement from the softmax prior using independent Gumbel noise $g \in \mathbb{R}^{|\mathcal{A}|}$ and the Gumbel Top-$K$ trick (Kool et al., 2019). The real action $a_t \in \mathcal{A}_{\text{topm}}$ is then chosen by allocating a fixed simulation budget via sequential halving (Karnin et al., 2013), and satisfies

$$a_t = \arg\max_{a \in \mathcal{A}_{\text{topm}}} \left[ g(a) + \theta_a + \sigma(q_\psi^\pi(a)) \right], \quad (9)$$

where $\theta_a$ are $\pi$'s logits, and $\sigma(q)$ is a strictly monotonically increasing function linear in $q$. In practice, we set $\sigma(q) = (c_{\text{visit}} + \max_b N(b)) c_{\text{scale}} q$ with $c_{\text{visit}} = 50$ and $c_{\text{scale}} = 1.0$, following *Gumbel MuZero* (Danihelka et al., 2022).

At non-root nodes, actions follow an improved policy derived from *completed* action values (Grill et al., 2020)

$$\text{completed}Q_\psi(a) = \begin{cases} q_\psi^\pi(a), & \text{if } N(a) > 0 \\ v_\psi^\pi, & \text{otherwise.} \end{cases} \quad (10)$$

$$\pi' = \text{softmax}(\theta + \sigma(\text{completed}Q_\psi)), \quad (11)$$

and actions are selected deterministically by mimicking $\pi'$ using the visit counts. We additionally use the improved policy as a distillation target for training, yielding the update

$$\theta^{(t+1)} = \theta^{(t)} + \xi \sigma \left( \text{completed}Q_\psi^{(t)} \right), \xi > 0 \quad (12)$$

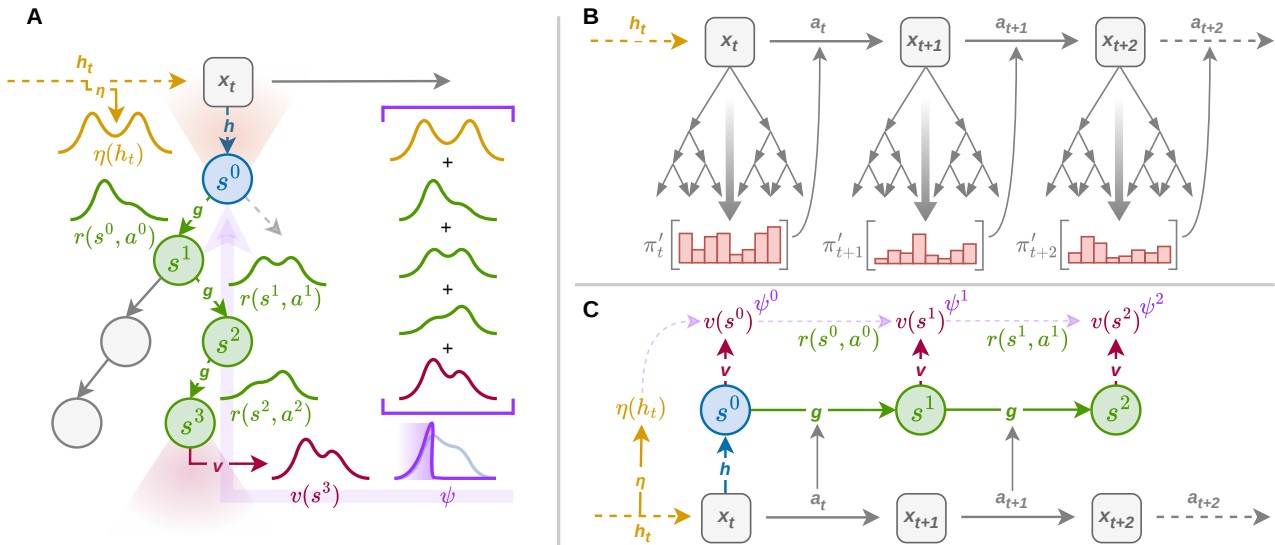

*Figure 2.* **Planning-based risk aversion with a learned model**. **(A)** Simulations combine the historical return $\eta(h_t)$, predicted rewards $r(s_t^k, a_t^k)$, and leaf value $v(s_t^\ell)$ distributions to estimate trajectory risk $\psi$. **(B)** MCTS yields an improved policy $\pi_t'$ that guides action selection. **(C)** The model is unrolled for $K$ steps with distributional predictions trained end-to-end; risk $\psi^k$ is estimated at each step.

With accurate estimates, this procedure ensures policy improvement in expected risk at both root and non-root nodes.

> **Proposition 3.4.** *Let $\pi'$ denote the policy induced by Gumbel selection at the root (9) and the improved policy (11) at non-root nodes, with parameters updated according to (12). Then the resulting policy satisfies*
>
> $$v_\psi^{\pi'}(x_0) \geq v_\psi^\pi(x_0).$$

A proof can be found in App. E.4. Leveraging this guarantee, and viewing the update in (12) as a policy-gradient step under a softmax policy, we show that *RiskZero* converges in expectation to a deterministic policy optimal in expected risk (see App. F). By Prop. 3.3, this then implies convergence to an optimal stationary risk-sensitive policy.

> **Theorem 3.1** (Global convergence). *Let rewards $R$ be bounded and let risk measure $\psi$ be preserving of boundedness. Assume the update in (12) is followed under a softmax policy with a positive linear isomorphism $\sigma$, and that every history $h$ is seen with positive probability. Then for all $h$, $v_\psi^{(t)}(h) \to v_\psi^*(h)$ as $t \to \infty$.*

## 4. *RiskZero*

*RiskZero* decomposes the trajectory return into two components, seen in Fig. 2A: the ***accumulated return*** along the history and the ***future return*** under policy $\pi$. Conditioned on a history, it learns to estimate both, yielding a complete characterization of the ***trajectory return*** distribution. These estimates, refined by ***immediate reward*** predictions, are integrated into search for efficient ***risk-sensitive*** policy learning.

**Models for Risk-Sensitive Search.** We adopt *MuZero*'s *representation*, *state dynamics*, and *policy* models unchanged. To support risk sensitivity, we replace the remaining models with distributional counterparts; the *reward dynamics* model $r_\phi : \mathcal{S} \times \mathcal{A} \to \Delta_\mathbb{R}$ predicts immediate reward *distributions*, while the *value* model $v_\phi : \mathcal{S} \to \Delta_\mathbb{R}$ predicts future return distributions. We further introduce a *historical return* model $\eta_\phi : \mathcal{H} \to \Delta_\mathbb{R}$ to predict accumulated return distributions along the history, enabling trajectory-level return estimation.

**Estimating Trajectory Return.** We propose decomposing the trajectory return $\zeta$ into accumulated and future components. The latter is modeled by $v_\phi$, and the former follows a similar recursion, motivating a backward TD operator $\mathcal{T}_B$

$$\mathcal{T}_B \zeta(h_t) \overset{D}{=} \mathbf{1}_{t \neq 0} \zeta(h_{t-1}) + \gamma^{t-1} R(x_{t-1}, a_{t-1}), \quad (13)$$

which directly inherits QTD's convergence guarantees.

Given quantiles, sampling from the trajectory return distribution follows naturally via the inverse transform theorem; mapping a uniform random variable through the quantile function recovers a sample from the according distribution (see App. A.2 for details). In practice, we sample the trajectory return by sampling the value, reward, and historical return models and summing samples index-wise. This scheme extends naturally to $n$-step returns

$$Z(x_0 \,|\, h_{t+n}) \overset{D}{=} \zeta(h_t) + \sum_{i=0}^{n-1} \gamma^{t+i} R(x_i, a_i) + \gamma^{t+n} Z(x_{t+n}), \quad (14)$$

resulting in more stable training and enabling search.

**Model Training.** Training enforces *distributional value equivalence* with predictions trained to match true out-

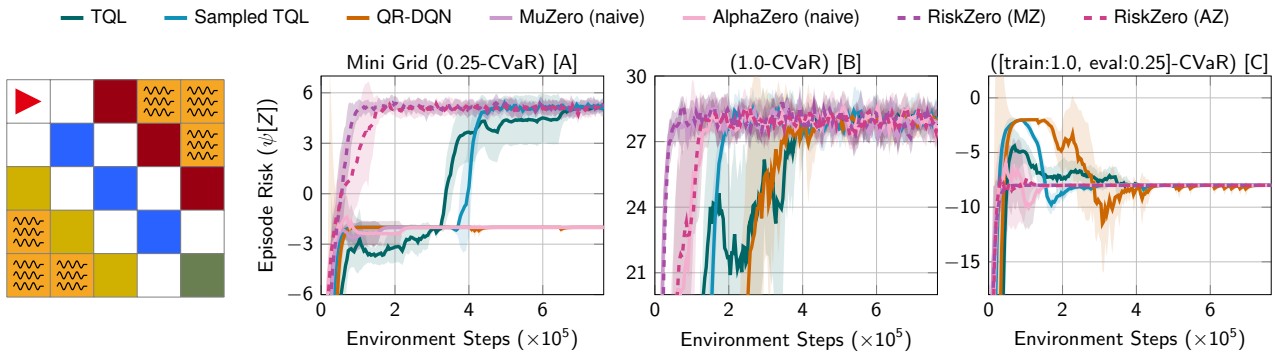

*Figure 3.* **Mini Grid results.** Episode risk $\psi[Z]$ on mini-grid averaged over 5 random seeds; risk is measured by rolling out the evaluation policy a number of times and applying $\psi$ on the resulting empirical distribution. Shaded regions denote standard errors. *RiskZero* matches *TQL*'s optimal policy under 0.25-CVaR with $3\times$ fewer samples (A), and maintains its advantage under risk-neutral optimization (B, C).

come distributions observed during environment interaction, $\eta_t \approx \zeta(h_t)$, $\pi_t^k \approx \pi'_{t+k}$, $v_t^k \approx Z(x_{t+k})$, and $r_t^k \approx R_{t+k}$. The historical return is used, and thus learned, only at the tree root.

As in *MuZero*, training proceeds via end-to-end unrolling (Fig. 2C). *RiskZero* extends this framework with distributional learning and an additional historical-return loss. For trajectory batches sampled from the replay buffer $\mathcal{D}$, we generate policy, value, reward, and historical-return targets $(\pi'_{t:t+K}, Z_{t:t+K}, R_{t:t+K}, \zeta_t)$, then jointly optimize by initializing $s_t^0$ and unrolling $g_\phi$ for $K$ steps, yielding the loss

$$\mathcal{L} = \mathbb{E}_{\mathcal{D}}\Big\{ \sum_{k=0}^{K} l^p(\pi_t^k, \pi'_{t+k}) + c_1 \sum_{k=0}^{K} l^v(v_t^k, Z_{t+k}) \\ + c_2 \sum_{k=0}^{K} l^r(r_t^k, R_{t+k}) + c_3 l^\eta(\eta_t, \zeta_t) \Big\}, \quad (15)$$

with loss coefficients $c_i \in \mathbb{R}$. The policy loss $l^p$ uses cross-entropy, while the value, reward, and historical return losses $(l^v, l^r, l^\eta)$ use the Quantile Huber loss $\rho$ (Dabney et al., 2018b). We derive targets $Z_t$ and $\zeta_t$ via $n$-step returns, $R_t$ is the sampled immediate reward, and $\pi'_t$ is the improved tree policy from (11), obtained following search (Fig. 2B).

**Risk-Sensitive Search.** *RiskZero* modifies MCTS to reason about risk within its abstract MDP. At each real timestep $t$, simulations construct a tree policy $\pi_t$ via *selection*, *expansion*, and *backup* (Fig. 2A). The abstract root is initialized not only with the latent state $s_t^0 = h_\phi(x_{\leq t})$, but also with the historical return distribution $\eta_t = \eta_\phi(x_{\leq t}, a_{<t})$.

Selection proceeds by drawing $D$ samples from the historical return at the root $\eta_{t,1:D} \sim \eta_t$. As search descends the tree, samples of immediate rewards are collected at each transition, $r_{t,1:D}^k \sim r_\phi(s_t^k, a_t^k)$, until reaching a leaf $s_t^{\ell-1}$. Upon expansion, we sample both the immediate reward and future return, $r_{t,1:D}^{\ell-1}$ and $v_{t,1:D}^\ell \sim v_\phi(s_t^\ell)$. These are combined according to (14) to produce $D$ trajectory return samples

$$z_j(x_0 \mid h_t^\ell) = \eta_{t,j} + \sum_{k=0}^{\ell-1} \gamma^{t+k} r_{t,j}^k + \gamma^{t+\ell} v_{t,j}^\ell, \quad (16)$$

for $j \in [D]$, and where $h_t^\ell := h_t a_t^1 s_t^1 \dots a_t^{\ell-1} s_t^\ell$. We sort the

samples $z_j$ and average indices according to the distorted quantiles in (1), yielding an approximation of $\psi[Z^\pi(x_0 \mid h_t^\ell)]$ that is then backed up along the search path to obtain estimates $\hat{q}_\psi^\pi(h, a)$. Please see App. B for the full algorithm.

## 5. Experiments

We first evaluate *RiskZero* on a Risky Mini Grid (Fig. 3) to confirm its ability to identify and converge to optimal risk-sensitive policies, and then on larger image-based (Fig. 4) and combinatorial (Fig. 6) problems to assess whether it scales to do so efficiently, and under diverse risk preferences.

**Baselines.** We evaluate two variants of *RiskZero*, built on *MuZero* and *AlphaZero*, both of which use *Gumbel* action selection. Our primary baseline is *TQL*, the only existing method that explicitly optimizes trajectory-level risk with theoretical guarantees. We consider both the original formulation (Zhou et al., 2023) and a variant that leverages our novel trajectory decomposition from Sec. 4, which we refer to as *Sampled TQL*. To ablate the importance of optimizing over entire trajectories rather than future returns alone, we compare against *QR-DQN* (Dabney et al., 2018b) and naive *RiskZero* variants that do not sample historical returns at the root. We also evaluate the CVaR-based method of Lim & Malik (2022), but omit these results as performance is very similar to *QR-DQN*. Results are averaged over five random seeds, and hyperparameters are provided in App. C.

**Mini Grid Results.** We first evaluate risk-sensitive policy learning on the stochastic Mini Grid task from (Zhou et al., 2023) (Fig. 3), a controlled toy setting in which optimal risk-sensitive policies and values can be computed exactly, enabling a clear sanity check of both convergence and solution quality. The agent moves right or down in a $5 \times 5$ grid from the upper-left to the bottom-right, incurring a $-1$ step cost and receiving random, cell-dependent rewards (yellow: 10 w.p. 0.75; blue: 2; red: 40 w.p. 0.3; orange: $-10$).

Under 0.25-CVaR optimization, *RiskZero* matches *TQL*'s

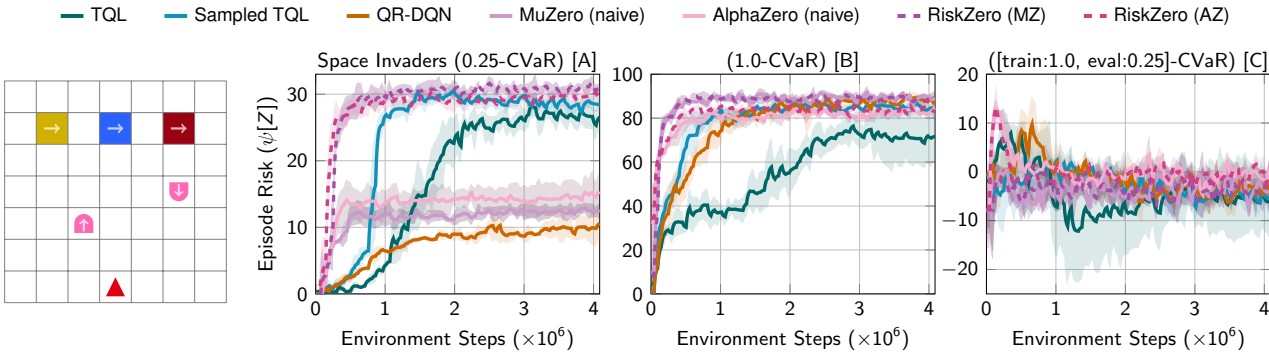

*Figure 4.* **Space Invaders results.** Episode risk averaged over 5 seeds; risk is measured by rolling out the evaluation policy a number of times and applying $\psi$ on the resulting empirical distribution. Shaded regions denote standard errors. *RiskZero* outperforms with fewest samples under 0.25-CVaR (A) and risk-neutral objectives (B), while risk-neutral policies underperform in risk-sensitive evaluation (C).

final performance while converging with $3\times$ fewer environment interactions (Fig. 3A). *TQL* initially converges to a suboptimal policy and only recovers after sufficient exploration. By contrast, *RiskZero* identifies the optimal policy early by leveraging lookahead search, which benefits from the more accurate value estimates near terminal states. This advantage persists under risk-neutral objectives (1.0-CVaR; Fig. 3B). However, the resulting risk-neutral policy exhibits substantially worse 0.25-CVaR performance (Fig. 3C). As expected, methods that naively optimize future risk converge to a suboptimal blue-cell policy: their DP formulation ignores trajectory-level risk, reinforcing locally favorable but globally inferior decisions. In particular, blue cells dominate near termination (CVaR of 2 versus 0), and this preference propagates backward through value updates.

**Space Invaders Results.** We repeat this experiment in a modified Space Invaders environment from MinAtar (Young & Tian, 2019) to assess scaling to longer horizons and image-based observations. The environment features three alien types with distinct rewards (Fig. 4): type 1 yields 40 w.p. 0.4 and $-10$ otherwise, type 2 yields 10 w.p. 0.75 and $-5$ otherwise, and type 3 yields a deterministic reward of 1. There are 8 waves, each starting with one alien of each type; eliminating any alien resets the full set. Other mechanics are preserved, namely avoiding alien projectiles and preventing them from reaching the floor (see App. C.1 for details).

*RiskZero* achieves the strongest performance and sample efficiency, converging to the optimal risk-sensitive policy with the fewest interactions (Fig. 4A), while *Sampled TQL* recovers the same policy with $\sim3\times$ more samples. *RiskZero* continues to dominate under risk-neutral preferences (Fig. 4B, C), whereas standard *TQL* is less efficient, exhibits $3\times$ higher variance, and converges to a suboptimal policy under both risk profiles. Again, naively optimizing future risk yields an overly pessimistic policy that favors type 3 aliens.

These results reaffirm that strong risk-sensitive performance in long-horizon tasks requires reasoning at the trajectory

level. Approaches that focus solely on future risk tend to converge to locally safe but globally suboptimal policies. By contrast, *RiskZero* integrates trajectory-aware optimization with look-ahead search to enable both correct action selection and substantially improved sample efficiency.

**Combinatorial Optimization Results.** Combinatorial optimization offers challenging benchmarks for RL and search-based methods via large state and action spaces. We evaluate on stochastic variants of two popular problems (Fig. 5):

- **Stochastic Bipartite Matching (SBM)**: Given a bipartite graph $G=(L\cup R, E)$ with stochastic edge weights $w : E \to \Delta_{\mathbb{R}}$, select a matching maximizing total weight.
- **Stochastic Maximum Independent Set (SMIS)**: Given a graph $G=(V, E)$ with stochastic vertex weights $w : V \to \Delta_{\mathbb{R}}$, select an independent set maximizing total weight.

For each problem, we generate 1,024 Erdős–Rényi (ER) $G_{n,m}$ (Erdős & Rényi, 1960) and Barabasi-Albert (BA) (Albert & Barabási, 2002) graphs[2], both commonly used to model real-world networks in RL benchmarking (Khalil et al., 2017; Abe et al., 2019; Ahn et al., 2020). The dataset is split evenly into training and test sets. SBM uses partitions $(|L|, |R|) \in \{(30, 10), (60, 30)\}$ and SMIS uses $|V| \in \{30, 60\}$, with edge densities varied from $10\%-40\%$.

We optimize a risk-sensitive objective evaluated over multiple realizations of the stochastic weights. In SBM, edges belong to one of two types: high-variance edges yield 20 w.p. 0.4 and $-7$ otherwise, while low-variance edges yield 6 w.p. 0.7 and $-3$ otherwise. Agents may also skip matching an edge, receiving a reward of 0 to avoid downside risk. In SMIS, vertices are similarly typed: type 1 yields 40 w.p. 0.3 and $-7$ otherwise, while type 2 yields 10 w.p. 0.7 and $-5$ otherwise; skipping a vertex again yields 0.

*RiskZero* consistently outperforms competing methods across both problems, graph sizes, and densities (Fig. 6),

---

[2]BA graph results, similar to ER graphs, appear in App. C.5.3.

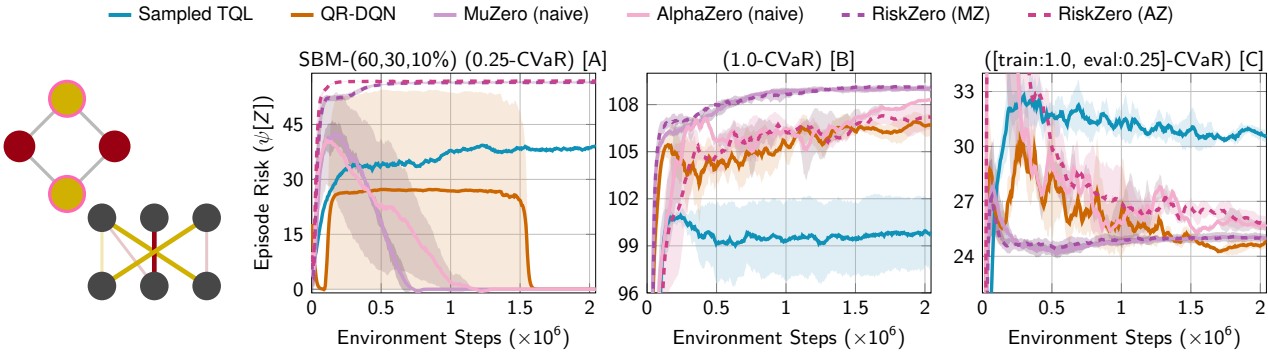

*Figure 5.* **Combinatorial optimization results.** SMIS selects a mutually non-adjacent node set (pink), while SBM assigns nodes from one partite set (top) to another (bottom), with node and edge types (red, yellow) inducing different reward distributions (left). We plot episode risk averaged over 5 seeds, computed by rolling out the policy as in our other experiments. *RiskZero* outperforms baselines, finding the optimal risk-sensitive (A) and risk-neutral policies (B), whereas risk-neutral policies underperform under risk-sensitive evaluation (C).

reliably converging to the optimal risk-sensitive (Fig. 5A) and neutral policies (Fig. 5B, C). We further evaluate on two other commonly used coherent risk measures, $\alpha$-Wang (Wang, 1996) and $\alpha$-POW (Dabney et al., 2018a), and observe similar trends; due to space, the corresponding results are reported in App. C.5. While *Sampled TQL* also learns risk-sensitive policies, it often fails to consistently reach the optimum, with performance degrading significantly on larger graphs. By contrast, *RiskZero* retains strong performance by pruning the search space using look-ahead search. Longer horizons and stochastic rewards also amplify return variance, preventing standard *TQL* from learning a meaningful policy; we therefore omit these results. Consistent with earlier experiments, naive methods that optimize future risk collapse to a trivial skip-all policy with zero total weight.

**Search Ablations.** We ablate the role of search in *RiskZero* on SBM (30,20,15%) by varying the planning method, Gumbel versus *MuZero* search, as well as the number of simulations $n$ and samples $D$ used during training and evaluation.

Gumbel planning, guaranteeing expected-risk improvement, consistently outperforms heuristic search (Fig. 6A), with as few as $n=2$ simulations exceeding all *MuZero* variants. Heuristic search requires substantially more simulations $n>16$ to approach similar performance. Both methods degrade with very few samples ($D=1,2$) but improve as $D$ increases (Fig. 6B), reflecting more accurate risk estimation during search. Gumbel search is more robust, exhibiting better performance with reduced sensitivity to sample count.

**Verification.** We assess the fidelity of the learned model by examining the accumulated return, value, and reward distributions in the Mini Grid environment. Results are provided in the appendix (App. C.5.5). Thanks to the environment's simple structure, the ground-truth quantile distributions can be computed exactly, enabling direct comparison; we plot the ground truth in dashed pink lines for comparison in Fig. 24. *RiskZero* recovers the correct quantile distributions, with learned distributions closely matching the ground truth.

## 6. Related Work

Prior RSRL work focuses primarily on model-free methods.

***Policy gradient*** methods are widely used for CVaR optimization (Tamar et al., 2015; Chow et al., 2018a; Tang et al., 2019; Yang et al., 2021; Ying et al., 2022). These approaches estimate $\alpha$-CVaR by optimizing the mean return of the worst $\alpha N$ trajectories from a batch of $N$ samples. While effective, they are sample-inefficient. Filtering discards the top $1-\alpha$ fraction of trajectories, which is particularly costly under strong risk aversion, and flat left tails lead to vanishing gradients that stall learning (Greenberg et al., 2022). By contrast, *RiskZero* provably converges to the optimal risk-aware policies while using *all* the samples it collects.

***Constrained RL*** tackles risk via safety constraints, often within a policy gradient framework. Lyapunov-based methods (Chow et al., 2018b) and Constrained Policy Optimization (Achiam et al., 2017) enforce auxiliary cost bounds, but introduce complexity and require careful constraint design tied to the chosen risk measure. *RiskZero* directly evaluates risk from learned distributions, avoiding explicit constraints and enabling simple adaptation to diverse risk preferences.

***Distributional RL*** methods such as QR-DQN, and IQN (Dabney et al., 2018b;a) learn full return distributions, with subsequent work selecting actions by maximizing future risk (Tamar et al., 2015; Keramati et al., 2020). However, because risk is often time-inconsistent, greedily optimizing future risk does not align with improving trajectory-level risk, often yielding suboptimal policies (Lim & Malik, 2022; Zhou et al., 2023). Proposed fixes include augmented MDPs that track accumulated returns (Lim & Malik, 2022; Moghimi & Ku, 2025; Pires et al., 2025) and trajectory-based approaches such as TQL (Zhou et al., 2023).

The state-augmentation and alternating optimization employed by the former group of methods can reduce sample efficiency. Accordingly, most of these approaches evalu-

Table 1. 0.25-CVaR return across various ER graph instances.

| Task | SBM | | | | | | SMIS | | | | | |
|---|---|---|---|---|---|---|---|---|---|---|---|---|
| Instance Size | 30–10 | | | 60–30 | | | 30 | | | 60 | | |
| Edge Densities (%) | 10 | 15 | 20 | 10 | 15 | 20 | 15 | 30 | 40 | 15 | 30 | 40 |
| QR-DQN | 0.0 | 0.0 | 0.0 | 0.0 | 0.0 | 0.0 | 0.0 | 0.0 | 0.0 | 0.0 | 0.0 | 0.0 |
| Sampled TQL | 6.96 | 9.98 | 12.00 | 38.78 | 47.75 | 41.80 | 13.22 | 8.19 | 6.66 | 10.58 | 14.29 | 11.18 |
| Risk AZ (N) | 0.0 | 0.0 | 0.0 | 0.0 | 1.51 | 0.0 | 0.0 | 0.0 | 0.0 | 0.0 | 0.0 | 0.0 |
| Risk MZ (N) | 0.0 | 0.0 | 0.0 | 0.0 | 0.0 | 0.0 | 0.0 | 0.0 | 0.0 | 0.0 | 0.0 | 0.0 |
| Risk AZ | 7.39 | 10.67 | 12.78 | **56.86** | **57.34** | **56.85** | **15.30** | **9.31** | 7.37 | **24.94** | 18.73 | 14.86 |
| Risk MZ | **7.40** | **10.68** | **12.79** | 56.48 | 57.16 | 56.58 | **15.30** | **9.31** | **7.38** | **24.94** | **18.74** | **14.92** |

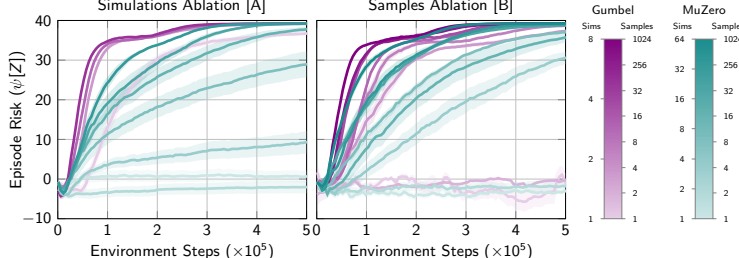

*Figure 6.* **More results and ablating search.** Risk over various graph instances (averaged over 5 seeds). *RiskZero* consistently achieves optimal policies across all sizes and densities (Tab. 1), while *Sampled TQL* degrades on larger instances. Gumbel planning outperforms heuristic search with as few as 2 simulations (A), and performance improves with more samples $D$ (B). See App. C.5 for more results.

ate in relatively simple and low-dimension environments (Moghimi & Ku, 2025; Pires et al., 2025), while others lack convergence guarantees (Lim & Malik, 2022), further motivating a planning-based approach that better scales.

Despite TQL's monotonic risk improvement, it lacks general guarantees and can be unstable in practice. This instability stems from modeling trajectory returns with a single historical network whose targets are derived from Monte Carlo samples and are bootstrapped from a jointly trained value network. Monte Carlo samples lead to higher variance targets at longer horizons. The historical network converges more slowly as a result, leading to poor action selection that hinders learning of the value network, which further worsens the historical network targets in turn. By contrast, our trajectory decomposition decouples these learning components, enabling more stable and scalable training.

*Model-based RL* gained prominence with AlphaGo, AlphaZero, and MuZero (Silver et al., 2016; 2017; Schrittwieser et al., 2020) which combine learned models with search for powerful policy improvement, with many extensions improving scalability and efficiency. Existing risk-aware variants either optimize expected utilities using classical MCTS (Hayes et al., 2023) or estimate CVaR via adversarial perturbations of learned dynamics (Rigter et al., 2021; 2023), adding complexity and limiting scalability. *RiskZero* is the first to unify these directions by planning directly with respect to arbitrary risks using learned return distributions and search, yielding stable and scalable risk-sensitive control.

## 7. Conclusion

We introduced a novel approach for planning-based risk aversion with *RiskZero*, the first *MuZero*-family method that plans directly with respect to risk using a learned model. By learning trajectory-level return distributions, and integrating them into Gumbel-based MCTS, *RiskZero* enables principled risk-sensitive planning compatible with diverse coherent risk measures. We showed that coherent risk measures admit deterministic optimal policies, that optimizing expected risk is sufficient for finding such policies, and that

*RiskZero*'s resulting policy updates provably converge to optimal stationary risk-sensitive policies. Across grid worlds, image-based control, and stochastic combinatorial optimization, *RiskZero* scales to achieve state-of-the-art risk-sensitive performance with substantial gains in sample efficiency.

**Limitations and Future Work.** More search is needed to fully realize the benefits of risk-sensitive planning. Our experiments focus on controlled environments for validation; scaling to more safety-critical domains remains an important direction for future work. We focus on stationary policies in deterministic dynamics with discrete actions. Extending to stochastic dynamics is a natural next-step, but it conflicts with Prop. 3.3, which relies on convergence to a deterministic $\pi^*$ so that expected and true risk coincide along a *fixed* trajectory. This equivalence breaks down under stochastic dynamics, inducing a gap that is non-trivial to bound.

Handling stochastic environments may therefore require revisiting the theory while preserving expectation-based updates, akin to prior work using state-augmented statistics for learning non-stationary policies in stochastic environments, e.g., (Moghimi & Ku, 2025; Pires et al., 2025). Combining such statistics with *Stochastic MuZero* (Antonoglou et al., 2021) is a promising direction for extending *RiskZero*.

More broadly, our approach is orthogonal to recent advances in deep search and may benefit from them. We focus on uncertainty intrinsic to the environment (*aleatoric*) rather than from estimation (*epistemic*). Modeling the latter, and reducing model misspecification more generally, may further improve robustness as in *RobustZero* (Li et al., 2025). Other natural extensions include continuous action spaces (*Sampled RiskZero*) (Hubert et al., 2021), improving memory (Pu et al., 2024), and efficiency (Wang et al., 2024b).

## Acknowledgements

We thank the anonymous reviewers for useful comments. YY gratefully acknowledges support from the Ontario Graduate Scholarship (OGS), Vector Scholarship in AI, and Canada Graduate Research Scholarship (CGRS M). JW acknowledges the support of NSERC, RGPIN-2024-05357.

## Impact Statement

This work advances reinforcement learning by introducing a general framework for risk-sensitive planning and, to the best of our knowledge, is the first to endow *MuZero*-family methods with principled risk-awareness. There are many potential societal consequences of our work, none which we feel must be specifically highlighted here.

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

## Outline of Appendix

The Appendix is structured as follows. In App. A, we provide extended background relevant to *RiskZero*, including a review of quantile regression and the quantile Huber loss (App. A.1), a brief overview of risk measures (App. A.3), and a discussion of stationary policies and time-consistency (App. A.4). App. B presents the full *RiskZero* algorithm in pseudocode, covering the training loop, self-play, Gumbel-based MCTS, and trajectory-level risk estimation by distortion. App. C details our experimental setup, describing environment specifications for the four risk-sensitive tasks (App. C), architectural details for our models (Apps. C.2 and C.3), and the hyperparameters used across all models and environments (App. C.4); we provide further empirical results related to our combinatorial optimization experiments in App. C.5. App. D establishes various supporting lemmas underlying our theoretical claims, beginning with mixture convexity of coherent risk measures (Cor. D.1) and auxiliary results used to prove policy improvement under Gumbel-based planning (Lems. D.2 to D.4). App. E provides short proofs for all the propositions mentioned in the main text, and App. F presents an argument for Thm. 3.1, establishing global convergence of *RiskZero*'s updates to an optimal stationary, risk-sensitive policy.

## A. Additional Background

### A.1. Quantile Regression and Quantile Huber Loss

**Quantile regression.** Quantile Regression DQN (QR-DQN; Dabney et al., 2018b) approximates the return distribution $Z^\pi(x, a)$ using a uniform mixture of $N$ learned quantiles. A parametric model $\phi : \mathcal{X} \times \mathcal{A} \to \mathbb{R}^N$ defines

$$Z_\phi(x, a) = \frac{1}{N} \sum_{i=1}^{N} \delta_{\phi_i(x,a)}, \tag{17}$$

where $\delta_x$ denotes the Dirac measure at $x \in \mathbb{R}$. Each $\phi_i$ is tied to a fixed quantile level $\omega_i$ and is trained via quantile TD regression (Koenker & Hallock, 2001; Rowland et al., 2024). Note that $Z_\phi$ is not itself a quantile distribution; rather, it is a probability density parameterized by its quantile locations. Learning thus amounts to positioning $N$ equally weighted particles to approximate $Z^\pi$.

From the CDF perspective, QR-DQN fixes uniformly spaced quantile levels and learns their supports, avoiding the need for a predefined value support as in categorical approaches (Bellemare et al., 2017). Each $\phi_i$ estimates the inverse CDF at the midpoint $\hat{\omega}_i = \frac{\omega_i + \omega_{i+1}}{2}$, yielding the optimal approximation under a constrained $p$-Wasserstein objective (Dabney et al., 2018b). For a transition $(x, a, r, x')$, the pairwise TD error between predicted and target quantiles is

$$\delta^{ij'} = r + \gamma \bar{Z}_{\phi'}(x', a' \mid \omega_j') - Z_\phi(x, a \mid \omega_i), \tag{18}$$

where $a' = \arg\max_a \mathbb{E}[\bar{Z}_{\phi'}(x', a)]$, and $\bar{Z}_{\phi'}$ is a target model to improve stability.

**Quantile Huber loss.** The standard quantile loss is non-smooth at zero, which can impair optimization with nonlinear function approximation. To address this, Dabney et al. (2018b) introduce the *quantile Huber loss*, a smooth variant based on the Huber loss (Huber, 1992)

$$\rho_\omega^\kappa(\delta) = \left| \omega - \mathbf{1}_{\delta < 0} \right| \frac{\mathcal{L}_\kappa(\delta)}{\kappa}, \tag{19}$$

$$\mathcal{L}_\kappa(\delta) = \begin{cases} \frac{1}{2}\delta^2, & \text{if } |\delta| \leq \kappa, \\ \kappa \left( |\delta| - \frac{1}{2}\kappa \right), & \text{otherwise.} \end{cases} \tag{20}$$

where $\kappa$ is the Huber threshold. The asymmetric weighting in (19) enforces the quantile ordering by penalizing over- and underestimation according to $\omega$. The loss is summed over predicted quantiles and averaged over target quantiles. Given $N$ quantiles, the total contribution of one step to the loss is

$$\mathcal{L}(x, a, r, x') = \frac{1}{N} \sum_{i=1}^{N} \sum_{j=1}^{N} \rho_{\hat{\omega}_i}^\kappa \left( \delta^{ij'} \right), \tag{21}$$

and the full loss is averaged over a minibatch of steps. Note that regression is performed with respect to $\hat{\omega}_i$ rather than $\omega_i$.

## A.2. Inverse Transform Sampling

Given the quantiles of a distribution, we may sample the underlying distribution itself. *RiskZero* leverages this idea of *inverse transform sampling* to factorize the trajectory-level return distribution, and estimate it during search. We state the inverse transform lemma below for completeness, justifying our factorization based approach.

**Lemma A.1.** *For any random variable $Z \in \mathbb{R}$, the random variable $F_Z^{-1}(U)$ has the same distribution as $Z$ where $F_Z^{-1}$ is the quantile distribution of $Z$ and $U$ is uniform on $[0, 1]$.*

*Proof.* Although the result is well-known, we provide a proof for completeness. Let $F_Z$ be the cumulative distribution function of an arbitrary random variable $Z \in \mathbb{R}$, and let $F_Z^{-1}$ denote its generalized inverse. We aim to show that the transformed variable $F_Z^{-1}(U)$ has cumulative distribution function $F_Z$, i.e.,

$$P(F_Z^{-1}(U) \le z) = F_Z(z).$$

Since $F_Z$ is non-decreasing, for any $u \in [0, 1]$ and $z \in \mathbb{R}$ we have $F_Z^{-1}(u) \le z \iff u \le F_Z(z)$ (by applying $F_Z$).

Hence

$$
\begin{aligned}
P(F_Z^{-1}(U) \le z) &= P(U \le F_Z(z)) \\
&= F_Z(z),
\end{aligned}
$$

where the final step uses the fact that $U$ is uniformly distributed on $[0, 1]$. $\qquad\square$

## A.3. Risk Measures

**Distortion Risk Measures.** A distortion risk *function* is a monotonic function $g_\psi : [0, 1] \to [0, 1]$ such that $g_\psi(0) = 0$ and $g_\psi(1) = 1$ (Dhaene et al., 2012; Wang et al., 2020). Such functions define a broad class of statistical functionals that encode risk preferences by nonlinearly distorting probability mass, thereby emphasizing either higher or lower risk events.

A distortion risk *measure* applies a distortion to the distribution of a random variable $Z$ and quantifies a risk-aware utility as a *distorted expectation* under the resulting perturbed distribution. The usual expectation is reweighted by $g'_\psi(F_Z(z))$

$$\psi[Z] = \int_{-\infty}^{\infty} z \, \frac{d}{dz}(g_\psi \circ F_Z)(z) \, dz = \int_{-\infty}^{\infty} z \, g'_\psi(F_Z(z)) f_Z(z) \, dz, \tag{22}$$

assuming $g_\psi$ is differentiable. Notably, these risk measures alter outcome likelihoods rather than rescaling to define utility.

For our purposes, given access to quantiles, it is more convenient to work with an equivalent quantile-based representation of (22). Applying the change of variables $\omega = F_Z(z)$ to (22) yields

$$\psi[Z] = \int_0^1 F_Z^{-1}(\omega) \, dg_\psi(\omega). \tag{23}$$

This formulation reveals that $\psi[Z]$ can be interpreted as the expectation of $F_Z^{-1}(\cdot)$ under a probability measure on $[0, 1]$ with cumulative distribution function (CDF) $g_\psi$, or density $g'_\psi$. In practice, sampling from $g'_\psi$ is conveniently performed using inverse transform sampling, making the quantile function $g_\psi^{-1}$ (i.e., inverse CDF) particularly useful for approximating $\psi$

$$\psi[Z] = \mathbb{E}_{\omega \sim g'_\psi}\left[F_Z^{-1}(\omega)\right] = \mathbb{E}_{u \sim U[0,1]}\left[F_Z^{-1}(g_\psi^{-1}(u))\right]. \tag{24}$$

We consider the following distortion functions in our experiments:

- **Mean.** The identity distortion

$$g_{\mathrm{mean}}^{-1}(\omega) = \omega, \tag{25}$$

  recovers the risk-neutral expectation by weighting all quantiles uniformly.

- $\alpha-$**CVaR.** Conditional Value at Risk (CVaR) (Rockafellar et al., 2000) corresponds to the average over the $\alpha$-*left* tail of the distribution, with distortion function

$$g_{\alpha-\mathrm{CVaR}}(\omega) = \min\{\omega/\alpha, 1\}. \tag{26}$$

  We note that some authors instead define $\alpha$-CVaR with respect to the $(1-\alpha)$-left tail. Since the derivative of $g_{\alpha-\mathrm{CVaR}}$ vanishes for $\omega > \alpha$, it suffices to consider the linear restriction $g_{\alpha-\mathrm{CVaR}}(\omega) = \omega/\alpha$ on $[0, \alpha]$, yielding the inverse

$$g_{\alpha-\mathrm{CVaR}}^{-1}(\omega) = \alpha\omega, \qquad \alpha \in [0, 1]. \tag{27}$$

- $\alpha-$**Wang.** The Wang transform (Wang, 1996) shifts quantiles in Gaussian space

$$g_{\alpha-\mathrm{Wang}}^{-1}(\omega) = \Phi\left(\Phi^{-1}(\omega) + \alpha\right), \qquad \alpha \in \mathbb{R}, \tag{28}$$

  where $\Phi$ denotes the standard Gaussian cumulative distribution function. Positive $\alpha$ induces risk-seeking behavior, while negative $\alpha$ induces risk aversion.

- $\alpha-$**POW.** The power distortion (Dabney et al., 2018a; Bernard et al., 2024) is defined as

$$g_{\alpha-\mathrm{POW}}^{-1}(\omega) = \begin{cases} \omega^{\frac{1}{1+|\alpha|}}, & \alpha \geq 0, \\ 1 - (1-\omega)^{\frac{1}{1+|\alpha|}}, & \alpha < 0, \end{cases} \tag{29}$$

  which induces risk-seeking ($\alpha > 0$) or risk-averse ($\alpha < 0$) preferences through a simple reweighting of quantiles.

The resulting distortion risk measures are each coherent as all three distortion risk functions are concave (Bernard et al., 2024). We discuss coherence further in the next section (this particular link to concavity of $g$ is given by Cor. A.1).

**Coherent Risk Measures.** Coherent risk measures were introduced by Artzner et al. (1999) to address limitations of earlier risk criteria, such as VaR. Namely, they are designed to satisfy intuitive axioms that reflect rational behavior under risk.

Formally, a risk measure $\psi$ is *coherent* if it satisfies the following properties (Artzner et al., 1999; Lam et al., 2022):

1. *Normalization:* $\psi[0] = 0$.

2. *Positive homogeneity:* $\psi[\alpha Z] = \alpha\psi[Z]$ for all $\alpha \geq 0$.

3. *Translation invariance:* $\psi[Z + \alpha] = \psi[Z] + \alpha$ for all constants $\alpha \in \mathbb{R}$.

4. *Monotonicity:* If $\mathbb{P}(Z_1 \leq Z_2) = 1$, then $\psi[Z_1] \leq \psi[Z_2]$.

5. *Super-additivity:* $\psi[Z_1 + Z_2] \geq \psi[Z_1] + \psi[Z_2]$.

Note that our setting treats $Z$ as a *gain* random variable representing total random return, in contrast to the financial literature, which typically models $Z$ as a *loss* variable. Accordingly, we aim to *maximize* a risk-aware utility of rewards, i.e., $\max \psi[Z]$, whereas the financial formulation seeks to *minimize* the risk of losses. As a result, the axioms of risk measures are inverted relative to the loss-minimization setting. Specifically, *super-additivity* under return maximization corresponds to *sub-additivity* under loss minimization. Monotonicity remains the same as smaller losses imply reduced *risk* of loss.

Let $X$ be a *loss* random variable, then its associated risk is given by Sereda et al. (2010, Section 3) as the Choquet integral

$$\rho_g[X] = \int_0^1 F_X^{-1}(\omega)d(1 - g(1 - \omega)) = \int_0^1 F_X^{-1}(1 - \omega)dg(\omega) \tag{30}$$

which again corresponds to a distorted expectation. However, this formulation differs from $\psi$ in (23) as the distortion is being applied to a loss distribution and therefore exaggerates the upper tail (i.e., larger losses) whereas $\psi$ applies the *same* distortion function (i.e., $g = g_\psi$) but exaggerates the lower tail. This distinction is reflected explicitly in the $g'(w)$-reweighted integrands $F_X^{-1}(1 - \omega)$ versus $F_X^{-1}(\omega)$. In what follows, we relate the two formulations and show that $\rho$ is coherent whenever $\psi$ is coherent and vice versa.

**Lemma A.2.** *Let $\rho_g$ be a distortion risk measure defined on losses and let $\psi$ be the associated measure defined on gains. Then, $\rho_g$ is coherent if and only if $\psi$ is coherent.*

*Proof.* By the properties of Choquet integrals, we have that properties 1-3 for coherence are satisfied for both $\rho$ and $\psi$. Monotonicity for both $\rho_g$ and $\psi$ follows from the fact that $g$ is monotone non-decreasing (Sereda et al., 2010). It remains to show that $\rho_g$ is sub-additive if and only if $\psi$ is super-additive.

We can relate the gain and loss formulations via Sereda et al. (2010, Section 4, Property 4)

$$-\rho_g[X] = \rho_{\tilde{g}}[-X] \tag{31}$$

where $\tilde{g}(u) := 1 - g(1 - u)$. Note that the right-hand side is applying to gains (or negative losses), and is nothing but $\psi$. Specifically, let $Z := -X$ define a gain and therefore using $\tilde{g}$ in the place of $g$ in the first equality of (30) recovers (23)

$$
\begin{aligned}
\rho_{\tilde{g}}[-X] &= \int_0^1 F_{-X}^{-1}(\omega) d(1 - \tilde{g}(1 - \omega)) && \text{definition of } \rho, \\
&= \int_0^1 F_Z^{-1}(\omega) d(1 - (1 - g(\omega))) && \text{definitions of } Z \text{ and } \tilde{g}, \\
&= \int_0^1 F_Z^{-1}(\omega) dg(\omega) = \psi[Z].
\end{aligned}
$$

The same relationship is established by substituting the identity $F_X^{-1}(\omega) = -F_{-X}^{-1}(1 - \omega)$ into the right equality in (30).

Now suppose $\rho_g$ is sub-additive, then for loss random variables $X_1$ and $X_2$ (with corresponding gains $Z_1 := -X_1$ and $Z_2 := -X_2$), we have

$$
\begin{aligned}
& \rho_g[X_1 + X_2] \leq \rho_g[X_1] + \rho_g[X_2] \\
\iff & -\rho_{\tilde{g}}[-X_1 - X_2] \leq -\rho_{\tilde{g}}[-X_1] - \rho_{\tilde{g}}[-X_2] \text{ by (31)} \\
\iff & \rho_{\tilde{g}}[-X_1 - X_2] \geq \rho_{\tilde{g}}[-X_1] + \rho_{\tilde{g}}[-X_2] \\
\iff & \psi[Z_1 + Z_2] \geq \psi[Z_1] + \psi[Z_2]
\end{aligned}
$$

which establishes super-additivity of $\psi$. □

**Corollary A.1.** *The risk measure $\psi$ defined on gains is coherent if and only if its distortion function $g_\psi$ is concave.*

*Proof.* It is well known that $\rho_g$ is sub-additive, and thus coherent, if and only if $g$ is concave (Wirch & Hardy, 2001). □

### A.4. Stationary, Adaptive and Non-adaptive Policies.

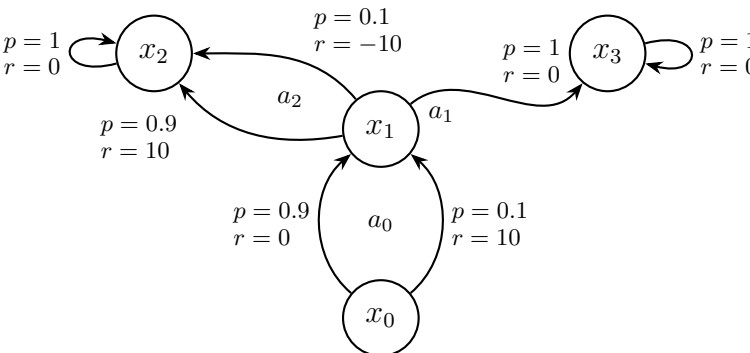

*Figure 7.* Example MDP from Stanko & Macek (2019) to convey the time-inconsistency of CVaR.

**Stationary Policies and Time-consistency.** A policy is said to be *stationary* (or *Markovian*) if it prescribes actions solely as a function of the current state, that is $\pi : \mathcal{X} \to \mathcal{A}$. This class of policies plays a central role in sequential decision-making,

as stationary policies are optimal whenever the return distribution is evaluated using a time-consistent measure, such as expectation (Ruszczyński, 2010). Although time consistency can be defined in several formal ways (Boda & Filar, 2006; Pflug & Pichler, 2016), its underlying intuition is straightforward. When a measure is time-consistent, optimal decision-making in a multi-step process can be made by focusing only on the future without the need to consider what happened in the past. By contrast, optimizing over a measure that is *time-inconsistent* generally requires considering the past, rendering standard policies, that are stationary with respect to the state, insufficient for optimal decision-making.

To illustrate this distinction, we consider the example from Stanko & Macek (2019), shown in Fig. 7 for $\psi \equiv 0.19$-CVaR. The MDP consists of an initial state $x_0$ with a single action $a_0$, followed by a state $x_1$ with actions $a_1$ and $a_2$; states $x_2$ and $x_3$ are terminal. We examine three policies: $\pi_1(x_1) = a_1$ and $\pi_2(x_1) = a_2$, which are the only two stationary policies, and a non-stationary policy $\pi_3$ that selects $a_2$ unless the agent receives a reward of 10 on the transition from $x_0$, in which case it selects $a_1$. We may compute the trajectory-risk under each policy with $\gamma = 1$

$$\mathrm{CVaR}_{0.19}(Z^{\pi_1}(x_0)) = \frac{0.19 \cdot 0}{0.19} = 0, \tag{32}$$

$$\mathrm{CVaR}_{0.19}(Z^{\pi_2}(x_0)) = \frac{0.09 \cdot (-10) + 0.01 \cdot 0 + 0.09 \cdot 10}{0.19} = 0, \tag{33}$$

$$\mathrm{CVaR}_{0.19}(Z^{\pi_3}(x_0)) = \frac{0.09 \cdot (-10) + 0.10 \cdot 10}{0.19} \approx 0.526. \tag{34}$$

It follows that the non-stationary policy $\pi_3$, which conditions on past rewards, strictly outperforms all stationary policies for the CVaR objective in Fig. 7. Hence, optimal decision-making with respect to time-inconsistent objectives generally requires policies that depend on historical information.

**Non-stationary Policies.** One approach to addressing time inconsistency is to explicitly account for realized returns at each state, typically by augmenting the state space with a continuous budgeting variable $y \in \mathbb{R}$ that represents confidence levels used for future risk evaluation (Bäuerle & Ott, 2011; Chow et al., 2015; Stanko & Macek, 2019; Bellemare et al., 2023; Moghimi & Ku, 2025; Pires et al., 2025; Moghimi et al., 2026). Policies derived from such formulations are *non-stationary* with respect to state, as their behavior also depends on $y$. They adapt their actions based on past reward realizations and are therefore termed *adaptive* in the stochastic optimization literature (Goemans & Vondrák, 2006; Ghuge et al., 2021).

While theoretically sound, the continuous nature of $y$ renders learning such adaptive policies difficult in many settings as each state must be visited under multiple histories and realizations of $y$, substantially reducing sample efficiency. Some existing adaptive methods (Chow et al., 2015; Stanko & Macek, 2019; Pires et al., 2025) are restricted to $\psi \equiv \mathrm{CVaR}$ and rely on computationally intensive procedures for optimization (Chow et al., 2015; Stanko & Macek, 2019). Others generalize beyond CVaR (Moghimi & Ku, 2025) but lack optimal convergence guarantees and rely on bilevel optimization that may further reduce sample efficiency (Bellemare et al., 2023; Moghimi & Ku, 2025; Pires et al., 2025; Moghimi et al., 2026).

**Non-adaptive Policies.** For these reasons, recent work has restricted attention to stationary policies when optimizing risk-sensitive objectives (Lim & Malik, 2022), despite their potential suboptimality in the general case. This restriction simplifies learning by ensuring that a policy's action in state $x$ is fixed and independent of realized rewards. Such policies are therefore *non-adaptive*; however, they are still not necessarily greedy with respect to *future* risk, as the underlying objective remains time-inconsistent (Lim & Malik, 2022).

Indeed, one may attempt to learn $Z^{\pi}(x, a)$ using the standard Bellman Backup $\mathcal{T}^*$

$$\mathcal{T}^*_{\psi} Z^{\pi}(x, a) \stackrel{D}{=} R + \gamma Z^{\pi}(X', A'), \tag{35}$$

$$A' = \arg\max_{a'} \psi[Z^{\pi}(X', a')], \tag{36}$$

that selects actions to optimize future risk (Dabney et al., 2018a). While empirically effective, prior work shows that optimizing only future returns fails to reduce risk across the trajectory, leading to biased optimization that may be arbitrarily poor (Zhou et al., 2023). Hence, $\mathcal{T}^*$ does not converge to the optimal stationary risk-sensitive policy—static *or* dynamic.

Although Lim & Malik (2022) seek a stationary optimal policy, their behavioral policy during learning still conditions on realized returns. This mismatch complicates analysis and leaves convergence behavior unresolved. Unbiased optimization instead requires accounting for accumulated returns along the history, starting from $x_0$ (Zhou et al., 2023).

In this work, we instead consider greedy action selection with respect to *trajectory* risk for both the target and behavioral policies. Actions are chosen greedily according to the return distribution induced along the entire trajectory, rather than with

respect to future risk alone. This approach is still non-adaptive: while the return distribution characterizes possible outcomes, the policy does not condition on *realized* rewards and therefore does not adjust its behavior based on past outcomes.

Instead, the policy is given return distributions over candidate action sequences and must effectively commit to one of these sequences. Consequently, both the behavioral and target policies are stationary with respect to *histories* rather than *states*. Every state-stationary policy can be realized as a history-stationary policy by conditioning only on the most recent state, making the former a strict subclass of the latter.

# B. Algorithm

We provide the pseudocode for *RiskZero*, as described in Sec. 4, in Algorithm 1 below.

---

**Algorithm 1:** RiskZero (Part 1)

---

**Global:** Replay buffer $\mathcal{D}$; Models: $h_\phi, g_\phi, r_\phi, \pi_\phi, v_\phi, \eta_\phi$

**Procedure** `TrainingLoop()`:

    $t \leftarrow 0, x_t \leftarrow$ `env.reset()`, $h_t \leftarrow (x_t,)$

    **for** *training_iterations* **do**

        $x_t, h_t \leftarrow$ `Selfplay(num_selfplay_iterations, ` $x_t, h_t$ `)`

        **for** *update_epochs_per_iteration* **do**

            | `UpdateModel()`

        **end**

    **end**

**Procedure** `SelfPlay(` $n$ `, ` $x_0$ `, ` $h_0$ `)`:

    **for** $t = 0$ **to** $n - 1$ **do**

        $s_t^0 \leftarrow h_\phi(h_t), \eta_t \leftarrow \eta_\phi(h_t), \pi_t^0 \leftarrow \pi_\phi(s_t^0)$

        $a_t, \pi_t' \leftarrow$ `GumbelMCTS(` $s_t^0, \pi_t^0, \eta_t$ `)`

        $x_{t+1}, r_t, d_t \leftarrow$ `env.step(` $a_t$ `)` // $d_t$ `indicates termination`

        $h_{t+1} \leftarrow (h_t, a_t, x_{t+1})$

        **if** $d_t = 1$ **then**

            $x_{t+1} \leftarrow$ `env.reset()`

            $h_{t+1} \leftarrow (x_{t+1},)$

        **end**

    **end**

    $\mathcal{D} \leftarrow \mathcal{D} \cup \{h_t, x_t, a_t, r_t, x_{t+1}, \pi_t', d_t\}_{t=0}^{n-1}$

    **return** $x_t, h_t$

**Procedure** `UpdateModel()`:

    Sample a mini-batch of sequences $\{(h_t, x_t, a_t, R_t, x_{t+1}, \pi_t', d_t)_{t=i}^{i+K}\} \sim \mathcal{D}$ // $K$ `is the unroll length`

    Compute target TD-values[a] $Z_{t:t+K}, \zeta_{t:t+K}$

    Set absorbing-state targets (following $d_t = 1$) as zero rewards/values, and uniform $\pi'$ (Schrittwieser et al., 2020).

    Optimize the world model and policy jointly according to Equation (15)

**Procedure** `GumbelMCTS(` $s_t, \pi_t, \eta_t$ `)`:

    Sample $|\mathcal{A}|$ Gumbel variables $(g \in \mathbb{R}^{|\mathcal{A}|}) \sim \text{Gumbel}(0)$

    $\mathcal{A}_{\text{topm}} \leftarrow$ sample $m$ actions without replacement from $\pi_t$ using $g$ and the Gumbel Top-$K$ trick (Kool et al., 2019)

    Get $a_t$ via Sequential Halving (Karnin et al., 2013) with $num\_sim$ `simulation(` $g, s_t, a, \pi_t, \eta_t$ `)` to $a \in \mathcal{A}_{\text{topm}}$

    Construct improved policy $\pi'$ at the root $s_t^0$ using Equation (11).

    **return** $a_t$ and $\pi'$

**Procedure** `DistortionRisk(` $z$ `)`:

    **Require:** $z \in \mathbb{R}^n$ (distribution sample values)

    **Require:** $g_\psi^{-1} : [0,1] \to [0,1]$ (inverse distortion function)

    **Require:** $M$ (number of quantile samples)

    Sort $z$ in ascending order: $z_{(1)} \leq \cdots \leq z_{(n)}$

    **for** $m = 1$ **to** $M$ **do**

        $q_m \leftarrow \dfrac{m - 0.5}{M}$

        $\tilde{q}_m \leftarrow g_\psi^{-1}(q_m)$

        $i_m \leftarrow \lceil \tilde{q}_m \cdot n \rceil - 1$

        $y_m \leftarrow z_{(i_m)}$

    **end**

    **return** $\dfrac{1}{M} \sum_{m=1}^{M} y_m$

---

[a] We also obtain target latent states $s_{t+1:t+K+1}^0$ by applying $h_\phi$ to real observations $x_t$, and use them to compute the self-consistency loss from (Ye et al., 2021).

---

**Algorithm 1:** RiskZero (Part 2)

---

**Procedure** Simulation(`g, `$s_t$`, a, `$\pi_t$`, `$\eta_t$`)`:

    `// The following process repeats for `*num_sim*` simulations,`
    `// where i denotes the simulation index.`
    **Require:** $N_i(s,a), Q_\psi^i(s,a)$
    Initialize root node $s_t^0 \leftarrow s_t$ and prior policy $\pi_t^0 \leftarrow \pi_t$ and action $a_t^0 \leftarrow a_t$
    Sample $\eta_{t,1:D} \sim \eta_t$ `// skipped for naive variants`
    $k \leftarrow 0$
    **while** $N_i(s_t^k, a_t^k) > 0$ **do**
        Sample $r_{1:D}(s_t^k, a_t^k) \sim r_t^k$
        $k \leftarrow k + 1$
        $s_t^k \leftarrow g_\phi(s_t^{k-1}, a_t^{k-1})$ `// can be saved and queried from the expansion step below`
        $a_t^k \leftarrow \arg\max_a \left( \pi'(a) - \frac{N_i(s_t^k, a)}{1 + \sum_b N_i(s_t^k, b)} \right)$ `// where `$\pi'$` is from (11) and the `$\theta_a$` are from `$\pi_t^k$
    **end**
    `// reached a leaf node (`$s_t^k, a_t^k$`)`
    Evaluate the leaf node $s_t^{\ell-1} := s_t^k$ using the learned model:
    $s_t^\ell \leftarrow g_\phi(s_t^{\ell-1}, a_t^{\ell-1}), r_t^{\ell-1} \leftarrow r_\phi(s_t^{\ell-1}, a_t^{\ell-1})$
    $\pi_t^\ell \leftarrow \pi_\phi(s_t^\ell), v_t^\ell \leftarrow v_\phi(s_t^\ell)$
    Sample $v_{1:D} \sim v_t^\ell$, and $r_{1:D}(s_t^{\ell-1}, a_t^{\ell-1}) \sim r_t^{\ell-1}$
    Compute trajectory distortion risk:
    $z_{1:D} \leftarrow \eta_{1:D} + \sum_{k=0}^{\ell-1} \gamma^{t+k} r_{1:D}(s_t^k, a_t^k) + v_{1:D}$ `// Samples are summed index-wise`
    $q_\psi \leftarrow$ `DistortionRisk(`$z_{1:D}$`)`
    **for** *each* $(s,a)$ *along the search path* **do**
        $Q_\psi^{i+1}(s,a) \leftarrow \dfrac{N_i(s,a) \, Q_\psi^i(s,a) + q_\psi}{N_i(s,a) + 1}$
        $N_{i+1}(s,a) \leftarrow N_i(s,a) + 1$
    **end**

---

# C. Experimental Details

## C.1. Environment Details

We introduce four environments to evaluate *RiskZero* across a range of risk-averse preferences, as well as to assess its overall performance and sample efficiency. Consistent with much of the prior planning literature (Silver et al., 2016; Schrittwieser et al., 2020; Li et al., 2025), all environments exhibit deterministic dynamics, while risk is introduced through stochastic reward structures. Specifically, certain actions yield riskier reward distributions—characterized by higher variance or the potential for negative outcomes—relative to others. Each environment is already described in Sec. 5; here, we provide additional implementation details related to the underlying MDP formulations.

**Risky Mini-Grid.** The environment consists of a 5×5 grid in which the agent can take one of two actions at each cell: move right or move downward (Fig. 3). Each action deterministically moves the agent one cell in the corresponding direction, except at grid boundaries, where the action results in a no-op. States are represented using normalized Cartesian coordinates. An episode terminates when the agent reaches the goal cell located at the bottom-right corner of the grid.

**Risky Space-Invaders.** This environment is a modified version of the MinAtar Space Invaders environment (Young & Tian, 2019), designed to introduce risk through stochastic rewards. The agent controls a cannon positioned at the bottom of the screen and fires projectiles at aliens located above (Fig. 4). Aliens move horizontally across the screen until one reaches a boundary, at which point all aliens move downward by one pixel and reverse direction. The agent's goal is to eliminate the aliens before they reach the floor; aliens also fire projectiles toward the agent, which must be avoided.

We modify the environment to include exactly three aliens at all times, each belonging to a distinct type. Shooting an alien yields a type-specific distributional reward, as described in Sec. 5. Upon being shot, all aliens immediately respawn at their

original locations, while the agent remains in its current position. This process repeats for up to eight waves. The action space consists of four discrete actions: no-op, move left, move right, and fire a projectile upward. State transitions are deterministic and correspond directly to the selected movement or firing action. Projectiles fired by the agent and by aliens move upward or downward, respectively, at a rate of one cell per time step.

Each state is represented as a $7\times7$ image with nine channels. The first four channels encode the positions of the agent and the three alien types, with one channel per type. The next two channels represent the positions of the agent's and aliens' projectiles. Two additional channels indicate the current horizontal movement direction of the aliens (left or right), with the active direction encoded at each alien's location. The final channel contains the normalized wave index (ranging from one to eight), providing an explicit signal of the agent's progress. An episode terminates if (i) the agent eliminates eight aliens, (ii) the agent is hit by an alien projectile, or (iii) an alien reaches the bottom of the screen.

**Stochastic Bipartite Matching.** The environment is defined by a bipartite graph $G = (L \cup R, E)$, with the goal of matching vertices from the $L$-set to those in the $R$-set. At each step, the agent is presented with a vertex from the $R$ partition and must match it to an *available* adjacent vertex in the $L$ partition. Accordingly, the action space at each step consists of all currently available neighbors of the presented $R$-vertex. Upon matching vertices $u \in L$ and $v \in R$, the agent receives a stochastic reward drawn from a distribution determined by the type of the edge $\{u, v\}$, as described in Sec. 5. The corresponding $u$ vertex is then no longer available for matching. An episode terminates once all vertices in the $R$ partition have been matched. To simplify the implementation, we guarantee feasibility at every step by ensuring each vertex in $R$ is connected to a dummy vertex in $L$ via an edge that yields zero reward.

The state representation encodes both the graph structure and the agent's current partial matching. Each vertex $v$ is associated with a feature vector $\mathbf{x}_v = [\text{to\_match} \,\|\, \text{vertex\_type}]$, where $\|$ denotes concatenation, $\text{to\_match}$ is a binary indicator equal to 1 if $v$ is the vertex to be matched at the current step and 0 otherwise, and $\text{vertex\_type}$ denotes the vertex partition (with 1 indicating one partition and 0 the other). Each edge $\{u, v\}$ is represented by a feature vector $\mathbf{e}_{uv} = [\text{edge\_type} \,\|\, \text{incident\_to\_match} \,\|\, \text{is\_selected}]$, where $\text{edge\_type}$ is a one-hot encoding of the edge type, $\text{incident\_to\_match}$ indicates whether the edge is incident to the vertex being matched at the current step, and $\text{is\_selected}$ indicates whether the edge has been selected in a prior step. Finally, the state also includes the ratio of matched vertices to the total number of vertices in the $R$ partition as a global measure of progress.

**Stochastic Maximum Independent Set.** The environment is defined by an undirected graph $G = (V, E)$, and the agent is tasked with creating an independent set $I \subseteq V$ containing no adjacent vertices. At each step, the agent selects a vertex $v \in V \setminus I$ to add to its independent set and receives a stochastic reward drawn from a distribution determined by the selected vertex's type, as described in Sec. 5. At the beginning of an episode, all vertices are available. Selecting a vertex marks it as chosen and renders all of its neighbors unavailable for subsequent selection. An episode terminates when no further vertices can be selected, that is, when all vertices are either selected or unavailable.

The state representation encodes the graph structure together with the agent's partially chosen independent set. Each vertex $v$ is associated with a feature vector $\mathbf{x}_v = [\text{selected} \,\|\, \text{available}, \text{type}]$, where $\text{selected}$ and $\text{available}$ are binary indicators denoting whether the vertex has been chosen and whether it remains eligible for selection, respectively; selected vertices are automatically marked as unavailable. The $\text{type}$ component is a one-hot encoding of the vertex's type. In addition, the state includes global features summarizing progress and performance following (Khalil et al., 2017): the ratio of unavailable vertices to the total number of vertices, the ratio of covered edges (i.e., edges incident to at least one selected vertex) to the total number of edges, and the ratio of selected vertices to the total number of vertices.

**Graph Generation.** We use two families of random graphs in our experiments: Erdős–Rényi graphs $G_{n,m}$ (Erdős & Rényi, 1960) and Barabási–Albert (Albert & Barabási, 2002), both commonly used to model real-world networks in RL benchmarking (Khalil et al., 2017; Abe et al., 2019; Ahn et al., 2020). ER graphs are generated by uniformly sampling an $m$-subset of the possible edges. BA graphs follow a preferential attachment mechanism controlled by a parameter $k$.

For SMIS, the BA process begins with an initial graph of $k$ nodes. Nodes are added sequentially until the desired graph size is reached. Each new node forms $k$ edges to existing nodes, where the probability of connecting to a node $v$ is proportional to its degree, i.e., the ratio of the degree of $v$ to the total degree of the graph. For bipartite graphs used in SBM, preferential attachment is applied when connecting vertices in the $R$ partition. Each vertex in $R$ is connected to $m/|R|$ vertices in $L$, yielding a total of $m$ edges.

## C.2. Architecture Details

We implement *RiskZero* by extending the *MuZero* and *AlphaZero* architectures, with environment-specific modifications to individual network components based on the underlying state representation. Fully connected networks are used for vector-based environments (MiniGrid), convolutional networks for image-based environments (Space Invaders), and graph networks for graph-structured environments (Stochastic Maximum Independent Set and Stochastic Bipartite Matching). We detail the general architectural design below, with graph-specific modifications described in App. C.3.

**Representation Network.** The representation network maps a fixed-length history of raw observations to a latent state embedding shared by the dynamics, policy, and value networks. For raw observations $\mathbf{x}_t$ at time $t$, the representation network produces a latent embedding $\mathbf{z}_t \in \mathbb{R}^d$, where $d$ is the latent state dimension.

Since rewards are independent of actions in our environments, we treat histories as sequences of states rather than sequences of both states and actions. Each observation in the history window $\mathbf{x}_{t-H+1}, \ldots, \mathbf{x}_t$ is first processed independently by a modality-specific feature extractor $\varphi(\cdot)$. The produced feature sequence is then aggregated using a gated recurrent unit (GRU) (Chung et al., 2014). The hidden state at the final timestep is mapped to the latent state representation via a learned linear transformation followed by a $\tanh$ nonlinearity, which serves a stabilization role

$$\mathbf{h}_{t,\_} \leftarrow \mathrm{GRU}(\varphi(\mathbf{x}_{t-H+1}), \ldots, \varphi(\mathbf{x}_t)), \qquad \mathbf{z}_t \leftarrow \tanh(\boldsymbol{\phi}_z \mathbf{h}_t),$$

where $\boldsymbol{\phi}_z \in \mathbb{R}^{d \times d}$ is a learned parameter matrix. When relevant to the observed rewards, additional information—such as the sequence of past actions—can be encoded and included in the history window (Zhou et al., 2023).

The feature extractor $\varphi(\cdot)$ is instantiated differently depending on the modality. Across all modalities, the final output of the representation network is a $d$-dimensional latent embedding used by the dynamics, policy, and value networks.

- **Vector inputs.** Observations are processed using two fully connected layers with hidden and output dimension $d$, each followed by a $\tanh$ activation.
- **Image inputs.** Observations are processed using two convolutional layers with $3 \times 3$ kernels and ReLU activations. Spatial resolution is reduced using non-overlapping $2 \times 2$ average pooling with stride 2, applied independently across channels. The pooled feature maps are flattened and progressively projected to dimension $d$ via fully connected layers.
- **Graph inputs.** Observations are processed using graph convolutional layers operating on vertex and, when applicable, edge features. Node embeddings are aggregated using a permutation-invariant pooling operation to create a pooled embedding. The pool and individual node embedding are concatenated and projected to $d$-dimensional embeddings for each vertex, serving as the latent state. For graph-based problems, the representation network processes single states rather than histories, as the history is implicitly encoded in the current partial solution (see App. C.3).

Aside from the choice of representation network and changes specific to graph-based environments, all other architectural components and training procedures are shared across environments.

**Policy, Value, and Return History Networks.** The policy network maps the latent state embedding $\mathbf{z}_t$ to action logits $\boldsymbol{\theta}_t \in \mathbb{R}^{|\mathcal{A}|}$ using two fully connected layers with hidden dimension $d$ and a ReLU activation. The value network mirrors this architecture but outputs the values for $N$ quantiles, parameterizing a distribution. The return history network is architecturally identical to the value network and operates only on the root latent state

$$\boldsymbol{\theta}_t \leftarrow \boldsymbol{\phi}_{\pi,2}\big(\mathbf{z}_t + \mathrm{ReLU}\big(\boldsymbol{\phi}_{\pi,1}(\mathbf{z}_t)\big)\big),$$
$$\mathbf{v}_t \leftarrow \boldsymbol{\phi}_{v,2}\big(\mathbf{z}_t + \mathrm{ReLU}\big(\boldsymbol{\phi}_{v,1}(\mathbf{z}_t)\big)\big),$$
$$\boldsymbol{\eta}_t \leftarrow \boldsymbol{\phi}_{\eta,2}\big(\mathbf{z}_t + \mathrm{ReLU}\big(\boldsymbol{\phi}_{\eta,1}(\mathbf{z}_t)\big)\big),$$

where $\mathbf{v}_t, \boldsymbol{\eta}_t \in \mathbb{R}^N$ are the value and return-history distribution quantiles, and $\boldsymbol{\phi}_{\pi,i}$, $\boldsymbol{\phi}_{v,i}$, and $\boldsymbol{\phi}_{\eta,i}$ ($i = 1, 2$) denote learned linear maps with appropriate dimensions.

**State Dynamics and Reward Network.** The dynamics network is specific to *MuZero* and embeds the selected action $a_t^k$, combining it with the current latent state using a GRU. A residual connection with the original latent state is applied before projecting to the next latent state. The reward head projects the GRU output to $N$ quantiles

$$\mathbf{h}_t^k, \mathbf{y}_t^k \leftarrow \mathrm{GRU}\big(\mathrm{enc}(a_t^k), \mathbf{z}_t^k\big), \qquad \mathbf{z}_t^{k+1} \leftarrow \boldsymbol{\phi}_g\big(\mathbf{h}_t^k + \mathbf{z}_t^k\big), \qquad \mathbf{r}_t^k \leftarrow \boldsymbol{\phi}_r\big(\mathbf{y}_t^k\big),$$

where $\mathrm{enc}(\cdot)$ is an action embedding, $\mathbf{h}_t^k, \mathbf{y}_t^k \in \mathbb{R}^d$ are the GRU hidden state and output, $\mathbf{z}_t^{k+1} \in \mathbb{R}^d$ is the predicted next latent state, $\mathbf{r}_t^k \in \mathbb{R}^N$ is the predicted immediate reward distribution, and $\boldsymbol{\phi}_g$ and $\boldsymbol{\phi}_r$ denote learned linear projections.

**State Consistency.** We additionally incorporate the state-consistency objective from *EfficientZero* (Ye et al., 2021) to regularize multi-step unrolled dynamics. Following *SimSiam* (Chen & He, 2021), the unrolled latent state $\mathbf{z}_t^k$ is encouraged to match a projected version of the representation of the true future observation $x_{t+k}$ sampled from the replay buffer.

The projection head is a residual network with two fully connected layers with hidden and output dimension $d$ and ReLU activation. We project both representations, with gradients stopped on the target branch, as in SimSiam.

We describe the modifications required to support learning on graph-structured environments in App. C.3. In brief, the state representation consists of embeddings for each *node*, computed using a graph network, rather than a single global embedding. The policy prior, value, and reward predictions are therefore derived from these node representations, while incorporating global structure through a pooled graph embedding and any auxiliary information provided by the environment.

For *MuZero*, the dynamics model is implemented as a GNN that updates all node embeddings, along with a representation of the auxiliary state information. The self-consistency objective then encourages the predicted latent node and auxiliary representations to match those obtained from the true future states.

**Quantile Networks.** The remaining algorithms employed in our experiments make use of quantile Q-networks (Dabney et al., 2018b). These architectures are similar to the value network described above, except they output quantiles $\mathbf{q}_t \in \mathbb{R}^{N \times |\mathcal{A}|}$ *for each action*. For graph-based environments, actions correspond to nodes, so we simply employ a value network on node embeddings, yielding a quantile distribution corresponding to each node.

## C.3. Graph-Based Environments

### C.3.1. GRAPH NEURAL NETWORK BACKGROUND

We use graph neural networks (GNNs) to model graph-structured state spaces while preserving permutation invariance. This section briefly reviews the GNN framework before we describe how it is integrated into our overall architecture.

GNNs generalize neural network computation to graph-structured data $G = (V, E)$, where $V$ is a set of nodes and $E \subseteq V \times V$ a set of edges. Each node carries features $\mathbf{x}_v \in \mathbb{R}^{d_x}$ and edges may carry features $\mathbf{e}_{uv} \in \mathbb{R}^{d_e}$. Nodes are also called *vertices*, and we will use both names interchangeably.

The core operation in GNNs is *message passing* (Gilmer et al., 2017); at each layer $\ell$, each node $v \in V$ aggregates information from its neighbourhood

$$\mathcal{N}(v) := \{u : \{u, v\} \in E\}$$

and updates its hidden state

$$\mathbf{h}_v^{(\ell)} = \text{UPDATE}\left(\mathbf{h}_v^{(\ell-1)}, \text{AGGREGATE}\left(\left\{\mathbf{h}_u^{(\ell-1)} : u \in \mathcal{N}(v)\right\}\right)\right), \tag{37}$$

with $\mathbf{h}_v^{(0)} = \mathbf{x}_v$. A non-linear activation is often applied to $\mathbf{h}_v^{(\ell)}$ prior to the next layer to increase expressivity.

After $T$ layers, the embedding $\mathbf{h}_v^{(T)}$ incorporates information from the $T$-hop neighborhood of node $v$ and ideally forms a semantic representation of the node. Choices for AGGREGATE and UPDATE yield different architectures; we consider two.

**GraphSAGE** (Hamilton et al., 2017) applies the update

$$\mathbf{h}_v^{(\ell)} = \phi_1^{(\ell)} \mathbf{h}_v^{(\ell-1)} + \phi_2^{(\ell)} \left[\mathbf{h}_v^{(\ell-1)} \| \sum_{u \in \mathcal{N}(v)} \mathbf{h}_u^{(\ell-1)}\right], \tag{38}$$

where $\phi_1^{(\ell)}, \phi_2^{(\ell)}$ are learned weight matrices and $\|$ denotes concatenation.

**GraphConv** (Morris et al., 2019) extends this update to incorporate edge features $\mathbf{e}_{uv} \in \mathbb{R}^{d_e}$ via

$$\mathbf{h}_v^{(\ell)} = \phi_1^{(\ell)} \mathbf{h}_v^{(\ell-1)} + \phi_2^{(\ell)} \sum_{u \in \mathcal{N}(v)} \mathbf{e}_{uv} \odot \mathbf{h}_u^{(\ell-1)}, \tag{39}$$

where $\odot$ denotes elementwise multiplication, allowing relational structure to modulate aggregation.

Restricting message passing to a fixed number of hops can limit the ability of GNNs to capture long-range dependencies. To mitigate this issue, some formulations—including ours—introduce a *global* auxiliary node connected to all vertices in the graph and initialized with a learnable embedding (Gilmer et al., 2017; Battaglia et al., 2018). This global node enables efficient propagation of global context to all nodes while preserving the original graph structure.

### C.3.2. ARCHITECTURE DETAILS

We adapt the *RiskZero* architecture described in App. C.2 to explicitly incorporate GNNs for structured state representations.

**Representation Function.** Each vertex $v \in V$ is associated with a feature vector defined in App. C.1. These vertex-level features are first projected into a latent space of dimension $d$ using two fully connected layers with ReLU activations, yielding initial node embeddings $\mathbf{h}_v^{(0)} \in \mathbb{R}^d$.

To facilitate global information flow, we augment the graph with a single learnable *register* vertex that is connected to all original vertices. The augmented graph is treated as undirected. We then apply four message-passing layers indexed by $l = 0, \ldots, 3$, each equipped with ReLU activations, layer normalization (LN) (Ba et al., 2016), and residual connections. When edge features are absent, we use GraphSAGE-style convolutions; when edge features are present, we use edge-aware graph convolutions that jointly update node and edge embeddings. We denote both operation by GConv with the understanding that edges are ignored for the former.

Node updates then take the form

$$\mathbf{h}_v^{(l+1)} \leftarrow \mathbf{h}_v^{(l)} + \mathrm{LN}\Big(\mathrm{ReLU}\Big(\mathrm{GConv}^{(l)}\Big(\mathbf{h}_v^{(l)}, \{\mathbf{h}_u^{(l)} : u \in \mathcal{N}(v)\}, \{\mathbf{e}_{uv}^{(l)}\}\Big)\Big)\Big), \quad l = 0, \ldots, 3, \tag{40}$$

where $\mathbf{e}_{uv}^{(l)}$ denotes edge embeddings when available (and is omitted otherwise). For edge-aware models, edge embeddings are updated in parallel via learned linear transformations with ReLU activation, layer normalization, and residual connections.

After the final message-passing layer, we construct graph-level, auxiliary, and node-level representations. Let $\mathbf{a}$ denote auxiliary (non-graph) features. We compute

$$\mathbf{p} \leftarrow \phi_p \sum_{v \in V} \mathbf{x}_v^{(4)}, \qquad \mathbf{a}' \leftarrow \phi_a \mathbf{a},$$

and define the final node embeddings as

$$\mathbf{u}_v \leftarrow \mathbf{h}_v^{(4)} + \mathrm{LN}\Big(\mathrm{ReLU}\Big(\phi_u \left[\mathbf{h}_v^{(4)} \,\|\, \mathbf{p} \,\|\, \mathbf{a}'\right]\Big)\Big), \tag{41}$$

where $\phi_p$, $\phi_a$, $\phi_u$ are learned linear maps. The register node is discarded after aggregation. When edge features are present, the final edge embeddings $\{\mathbf{e}_{uv}^{(4)}\}$ are retained for downstream use.

The resulting node embeddings $\{\mathbf{u}_v\}$, auxiliary embedding $\mathbf{a}'$, and (when applicable) edge embeddings are passed to subsequent components of the architecture. For graph-based *TQL* and *QRDQN* baselines, state–action values are predicted directly from the resulting node embeddings.

**Policy, Value, and Return History Networks.** In graph-based environments, actions correspond to selecting vertices. Given node embeddings $\{\mathbf{u}_v\}$ and auxiliary embedding $\mathbf{a}'$, we compute node-wise policy logits using a shared feed forward network applied to each node embedding concatenated with a pooled graph representation

$$\pi(\cdot \mid S) \leftarrow \phi_\pi \left( \left[\mathbf{u}_v \,\|\, \sum_{u \in V} \mathbf{u}_u \,\|\, \mathbf{a}'\right] \right),$$

where $\phi_\pi$ is a learned linear map and $S$ denotes the current partial solution.

Value-based heads operate only on the pooled graph representation and auxiliary embedding, and output $N$ quantiles

$$\mathbf{v} \leftarrow \phi_v \left( \left[\sum_{v \in V} \mathbf{u}_v \,\|\, \mathbf{a}'\right] \right) \in \mathbb{R}^N,$$

where $\phi_v$ is a learned map. The return history network is architecturally identical to the value network and is evaluated only at the root latent state.

**State Dynamics and Reward Networks.** The dynamics model conditions on a selected vertex $v_t \in V$, which represents the chosen action at time step $t$. A learned action embedding $\mathbf{e}_a \in \mathbb{R}^d$ is injected by adding it to the embedding of the selected node, followed by a residual FFN update

$$\tilde{\mathbf{u}}_v \leftarrow \mathbf{u}_v + \mathbf{1}\{v = v_t\}\,\mathbf{e}_a + \mathrm{ReLU}(\phi_s \mathbf{u}_v),$$

where $\phi_s \in \mathbb{R}^{d \times d}$ is a learned linear map.

A pooled graph summary is then computed as

$$\bar{\mathbf{p}} \leftarrow \sum_{v \in V} \tilde{\mathbf{u}}_v, \qquad \bar{\mathbf{p}} \leftarrow \bar{\mathbf{p}} + \mathrm{ReLU}(\phi_{\bar{p}} \bar{\mathbf{p}}),$$

where $\phi_{\bar{p}}$ is a learned linear map. The immediate reward distribution is predicted from a state-action embedding constructed using the pooled graph representation, auxiliary embedding, and the selected node $v_t$'s embedding

$$\mathbf{r}_t \leftarrow \phi_r(\tilde{\mathbf{u}}_{v_t} + \mathrm{ReLU}(\phi_{sa}[\bar{\mathbf{p}} \,||\, \mathbf{a}' \,||\, \tilde{\mathbf{u}}_{v_t}])) \in \mathbb{R}^N,$$

with $\phi_r, \phi_{sa}$ as learned linear maps.

To compute the next latent graph state, we reintroduce the register node and apply two additional message-passing layers using the same convolutional operators as in the representation network (see equation (40)). Auxiliary features are updated via a residual FFN

$$\mathbf{a}' \leftarrow \mathbf{a}' + \mathrm{ReLU}(\phi_{a'} \mathbf{a}'),$$

and node embeddings are fused with global context using the same pooled auxiliary residual update as in the representation function (see equation (41)). The updated node embeddings, together with the auxiliary embedding and edge embeddings (when present), define the next latent state used for planning.

**State Consistency.** For graph-based environments, we extend the SimSiam-style state-consistency objective to the full latent graph state used for planning. At each unrolled step, predicted node embeddings (and edge embeddings, when present), along with the auxiliary embedding, are encouraged to match corresponding projected embeddings of the true future state sampled from the trajectory. Similarity is measured using cosine similarity, and gradients are stopped on the target branch, following SimSiam. The final state-consistency loss is computed as the sum of the node-, edge-, and auxiliary-level losses.

### C.4. Hyperparameters

We optimize via Adam (Kingma, 2014) with $\beta_1 = 0.9$, $\beta_2 = 0.999$ and $\epsilon = 1 \times 10^{-5}$. Tabs. 2 to 4 summarize the hyperparameters used for *Risk Alpha/MuZero*, *QR-DQN*, and *TQL*, respectively, across all evaluated environments. We fix $D = 1024$ quantile samples to estimate risk, as we find this to be sufficient by ablation (Fig. 6). Other choices follow prior work. We use 64 quantiles (Dabney et al., 2018a), hidden size 256 (Zhou et al., 2023) (64 for graph environments (Abe et al., 2019), and adopt the *(Gumbel) MuZero* (Schrittwieser et al., 2020; Danihelka et al., 2022) hyperparameters otherwise. For each algorithm and environment, we use the largest feasible batch size and tune learning rate and replay buffer size via a small sweep.

*Table 2. RiskZero* hyperparameters for all environments.

| | Grid | Space Invaders | SBM | MIS |
|---|---|---|---|---|
| Hidden dimension $d$ | 256 | 256 | 64 | 64 |
| Discount $\gamma$ | 1.0 | 1.0 | 1.0 | 1.0 |
| # quantiles $N$ | 64 | 64 | 64 | 64 |
| Quantile samples $D$ | 1024 | 1024 | 1024 | 1024 |
| Simulations per step | 32 | 32 | 32 | 32 |
| Value loss coeff. | 0.25 | 0.25 | 0.25 | 0.25 |
| State-consistency coeff. | 2.0 | 2.0 | 10.0 | 10.0 |
| $n$-step return | 5 | 5 | 5 | 5 |
| Huber parameter | 1.0 | 1.0 | 1.0 | 1.0 |
| Learning rate | $5 \times 10^{-3}$ | $5 \times 10^{-3}$ | $5 \times 10^{-4}$ | $5 \times 10^{-4}$ |
| Min. learning rate | $10^{-3}$ | $10^{-3}$ | $10^{-5}$ | $10^{-5}$ |
| LR decay iterations | 200 | 200 | 200 | 200 |
| Max grad norm | 5.0 | 5.0 | 5.0 | 5.0 |
| Train batch size | 1024 | 1024 | 256 | 256 |
| Self-play batch size | 32 | 32 | 32 | 32 |
| Unroll sequence length | 5 | 5 | 5 | 5 |
| Replay buffer size | 6,144 | 102,400 | 32,768 | 32,768 |
| Target update interval | 5 | 10 | 5 | 5 |
| Number of training iterations | 1000 | 2000 | 2000 | 2000 |
| Iteration num. steps | 24 | 64 | 32 | 32 |

*Table 3.* Hyperparameters for QR-DQN across all environments.

| | Grid | Space Invaders | SBM | MIS |
|---|---|---|---|---|
| Hidden dimension $d$ | 256 | 256 | 64 | 64 |
| Discount $\gamma$ | 1.0 | 1.0 | 1.0 | 1.0 |
| # quantiles $N$ | 64 | 64 | 64 | 64 |
| Huber parameter | 1.0 | 1.0 | 1.0 | 1.0 |
| $n$-step return | 5 | 5 | 5 | 5 |
| Learning rate | $10^{-4}$ | $10^{-4}$ | $10^{-4}$ | $10^{-4}$ |
| Min. learning rate | $10^{-5}$ | $10^{-5}$ | $10^{-5}$ | $10^{-5}$ |
| LR decay iterations | 1000 | 2000 | 2000 | 2000 |
| Max grad norm | 5.0 | 5.0 | 0.5 | 0.5 |
| Target update interval | 10 | 10 | 50 | 50 |
| Replay buffer size | 76,800 | 102,400 | 65,536 | 65,536 |
| Batch size | 1024 | 1024 | 256 | 256 |
| Learning start steps | 100 | 50 | 64 | 64 |
| Iteration num. steps | 24 | 64 | 32 | 32 |
| $\epsilon_{\text{start}}$ | 1.0 | 1.0 | 1.0 | 1.0 |
| $\epsilon_{\text{final}}$ | 0.05 | 0.1 | 0.1 | 0.1 |
| $\epsilon$ anneal steps | 500 | 1000 | 1500 | 1500 |
| Number of training iterations | 1000 | 2000 | 2000 | 2000 |

*Table 4.* Hyperparameters for TQL across all environments.

| | Grid | Space Invaders | SBM | MIS |
|---|---|---|---|---|
| Hidden dimension $d$ | 256 | 256 | 64 | 64 |
| Discount $\gamma$ | 1.0 | 1.0 | 1.0 | 1.0 |
| # quantiles $N$ | 64 | 64 | 64 | 64 |
| Quantile samples $D$ | 1024 | 1024 | 1024 | 1024 |
| Huber parameter | 1.0 | 1.0 | 1.0 | 1.0 |
| $n$-step return | 5 | 5 | 5 | 5 |
| History length | 10 | 10 | – | – |
| Learning rate | $10^{-4}$ | $10^{-4}$ | $10^{-4}$ | $10^{-4}$ |
| Min. learning rate | $10^{-5}$ | $10^{-5}$ | $10^{-5}$ | $10^{-5}$ |
| LR decay iterations | 1000 | 2000 | 2000 | 2000 |
| Max grad norm | 5.0 | 5.0 | 0.5 | 0.5 |
| Target update interval | 25 | 25 | 50 | 50 |
| Training epochs / iter | 50 | 50 | 20 | 20 |
| Replay buffer size | 76,800 | 102,400 | 65,536 | 65,536 |
| Batch size | 1024 | 1024 | 256 | 256 |
| Learning start steps | 100 | 50 | 64 | 64 |
| Iteration num. steps | 24 | 64 | 32 | 32 |
| $\epsilon_{\text{start}}$ | 1.0 | 1.0 | 1.0 | 0.5 |
| $\epsilon_{\text{final}}$ | 0.05 | 0.05 | 0.1 | 0.1 |
| $\epsilon$ anneal steps | 500 | 1000 | 1500 | 1500 |
| Number of training iterations | 1000 | 2000 | 2000 | 2000 |

## C.5. Additional Results

### C.5.1. EXPERIMENTS WITH OTHER RISK MEASURES

We perform additional SBM experiments with different coherent risk-measures in Figs. 8 and 9. *RiskZero* retains its performance, demonstrating robustness to the choice of distortion function.

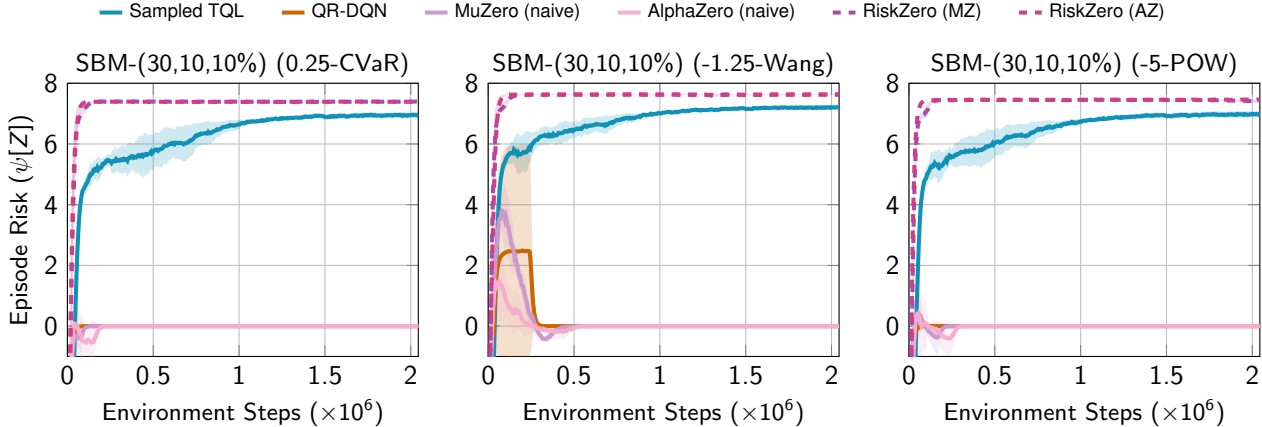

*Figure 8.* **Diverse coherent risk measures.** Mean episode risk on SBM-(30,10,10%) with 3 random seeds. Shades denote standard errors. *RiskZero* achieves optimal risk-sensitive performance across $0.25$-CVaR (left), $-1.25$-Wang (center), and $-5$-POW (right) risk measures.

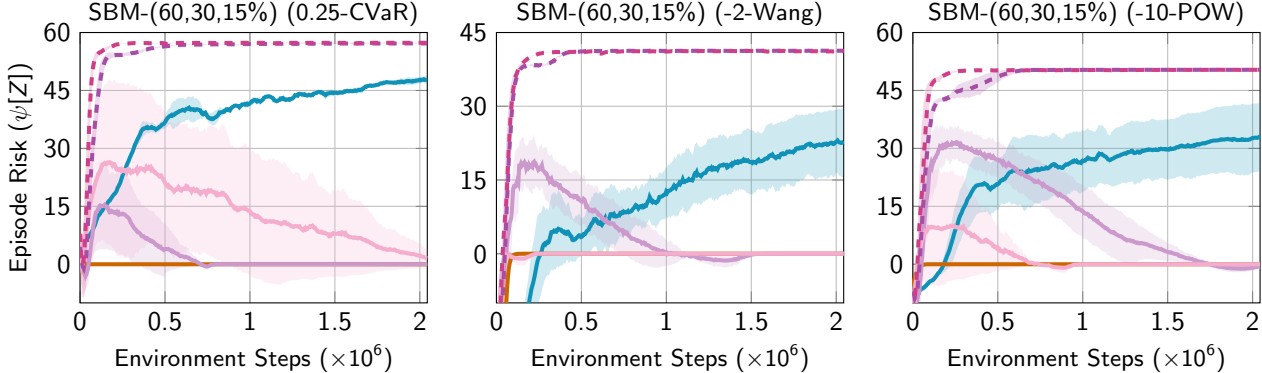

*Figure 9.* **Even more coherent risk measures.** Mean episode risk on larger SBM-(60,30,15%) instances with 3 random seeds. Shades denote standard errors. *RiskZero* achieves optimal performance across $0.25$-CVaR (left), $-2$-Wang (center), and $-10$-POW (right) risks.

### C.5.2. STOCHASTIC COMBINATORIAL OPTIMIZATION

We provide more learning-curves and standard-errors for our experiments on ER (and a few BA) graphs.

*Table 5.* Performance comparison of different methods on SBM instances on ER graphs with varying instance sizes (30-10 and 60-30 nodes) and edge densities (10%, 15%, 20%). Results show mean performance $\pm$ standard error across 5 random seeds. *RiskZero* significantly outperforms baseline methods, while QR-DQN and naive methods fail to learn effectively.

| | Task | SBM (0.25-CVaR) | | | | | |
|---|---|---|---|---|---|---|---|
| | Instance Size | 30–10 | | | 60–30 | | |
| | Edge Densities (%) | 10 | 15 | 20 | 10 | 15 | 20 |
| **Method** | QR-DQN | 0.0±0.0 | 0.0±0.0 | 0.0±0.0 | 0.0±0.0 | 0.0±0.0 | 0.0±0.0 |
| | Sampled TQL | 6.96±0.11 | 9.98±0.10 | 12.00±0.08 | 38.78±0.0 | 47.75±1.37 | 41.80±6.68 |
| | Risk AZ (N) | 0.0±0.0 | 0.0±0.0 | 0.0±0.0 | 0.0±0.0 | 1.51±2.14 | 0.0±0.0 |
| | Risk MZ (N) | 0.0±0.0 | 0.0±0.0 | 0.0±0.0 | 0.0±0.0 | 0.0±0.0 | 0.0±0.0 |
| | Risk AZ | 7.39±0.01 | 10.67±0.00 | 12.78±0.00 | 56.86±0.07 | 57.34±0.03 | 56.85±0.03 |
| | Risk MZ | 7.40±0.01 | 10.68±0.01 | 12.79±0.01 | 56.48±0.29 | 57.16±0.07 | 56.58±0.22 |

*Table 6.* Performance comparison on SMIS problems on ER graphs with instance sizes of 30 and 60 nodes across different edge densities (15%, 20%, 30%, 40%). *RiskZero* consistently achieves the highest performance, while naive methods are biased and learn a degenerate policy.

| | Task | SMIS (0.25-CVaR) | | | | | |
|---|---|---|---|---|---|---|---|
| | Instance Size | 30 | | | 60 | | |
| | Edge Densities (%) | 15 | 30 | 40 | 20 | 30 | 40 |
| **Method** | QR-DQN | 0.0±0.0 | 0.0±0.0 | 0.0±0.0 | 0.0±0.0 | 0.0±0.0 | 0.0±0.0 |
| | Sampled TQL | 13.22±2.62 | 8.19±0.72 | 6.66±0.46 | 10.58±8.01 | 14.29±0.55 | 11.18±0.15 |
| | Risk AZ (N) | 0.0±0.0 | 0.0±0.0 | 0.0±0.0 | 0.0±0.0 | 0.0±0.0 | 0.0±0.0 |
| | Risk MZ (N) | 0.0±0.0 | 0.0±0.0 | 0.0±0.0 | 0.0±0.0 | 0.0±0.0 | 0.0±0.0 |
| | Risk AZ | 15.30±0.02 | 9.31±0.03 | 7.37±0.01 | 24.94±0.01 | 18.73±0.01 | 14.86±0.04 |
| | Risk MZ | 15.30±0.01 | 9.31±0.01 | 7.38±0.01 | 24.94±0.01 | 18.74±0.02 | 14.92±0.03 |

*Table 7.* Performance comparison on SMIS problems on ER graphs under 1.0-CVaR objective with instance sizes of 30 and 60 nodes. All methods achieve reasonable performance under this risk-neutral setting, with *RiskZero* maintaining superior performance and lower variance compared to QR-DQN and Sampled TQL.

| | Task | SMIS (1.0-CVaR) | | | | | |
|---|---|---|---|---|---|---|---|
| | Instance Size | 30 | | | 60 | | |
| | Edge Densities (%) | 15 | 30 | 40 | 20 | 30 | 40 |
| **Method** | QR-DQN | 48.83±13.40 | 38.54±0.76 | 31.12±0.78 | 67.41±1.49 | 47.99±2.39 | 39.97±3.18 |
| | Sampled TQL | 51.81±6.77 | 33.00 ± 5.40 | 25.73±5.41 | 53.81±9.22 | 43.66±11.45 | 30.15±3.98 |
| | Risk AZ | 58.31±0.08 | 43.14 ± 0.05 | 36.90±0.05 | 76.75±0.09 | 60.18±0.11 | 48.34±0.38 |
| | Risk MZ | 58.40±0.04 | 43.20±0.08 | 36.96±0.03 | 77.36±0.02 | 60.15±0.02 | 48.45±0.05 |

*Table 8.* Performance comparison on SBM problems on ER graphs under 1.0-CVaR objective with instance sizes of 30-10 and 60-30. *RiskZero* achieves the best overall performance, particularly on larger instances (60–30), demonstrating robustness to the desired degree of risk aversion.

| | Task | SBM (1.0-CVaR) | | | | | |
|---|---|---|---|---|---|---|---|
| | Instance Size | 30–10 | | | 60–30 | | |
| | Edge Densities (%) | 10 | 15 | 20 | 10 | 15 | 20 |
| **Method** | QR-DQN | 31.21±0.10 | 35.02±0.14 | 36.29±0.09 | 106.70±0.77 | 111.35±0.86 | 113.00±0.14 |
| | Sampled TQL | 30.10±0.12 | 33.87±0.12 | 35.26±0.06 | 99.71±2.46 | 107.40±1.35 | 108.59±1.80 |
| | Risk AZ | 31.50±0.05 | 35.32±0.05 | 36.71±0.04 | 107.20±0.83 | 113.01±0.12 | 113.66±0.10 |
| | Risk MZ | 31.59±0.01 | 35.35±0.03 | 36.73±0.02 | 109.10±0.13 | 113.33±0.07 | 113.55±0.15 |

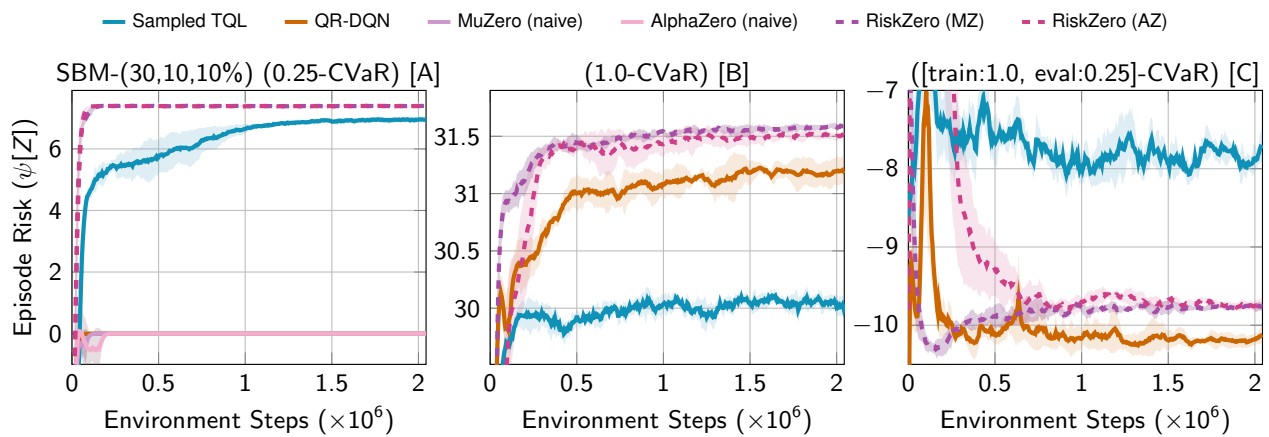

*Figure 10.* **SBM-(30,10,10%)** **results across risk profiles.** *RiskZero* converges to optimal policies under 0.25-CVaR (A) and risk-neutral 1.0-CVaR (B) objectives, while risk-neutral policies exhibit degraded performance under risk-sensitive evaluation (C).

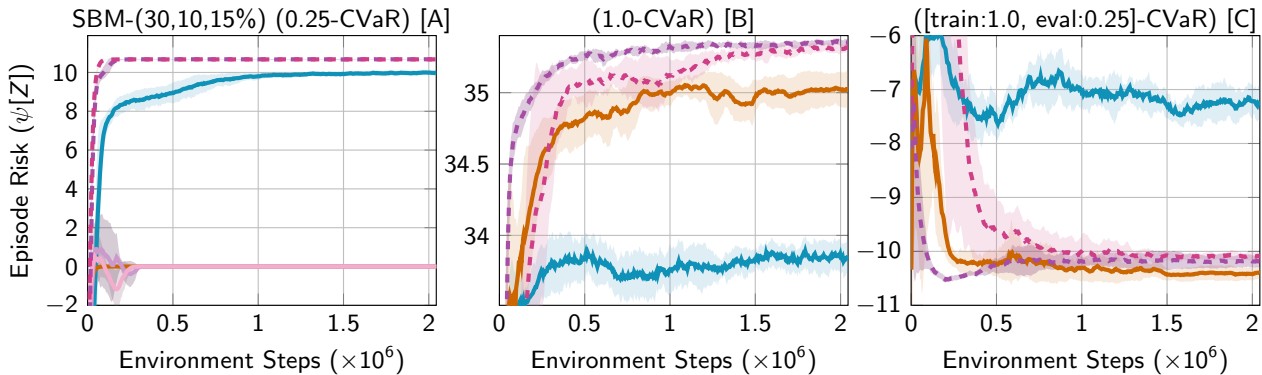

*Figure 11.* **SBM-(30,10,15%)** **results across risk profiles.** *RiskZero* converges to optimal policies under 0.25-CVaR (A) and risk-neutral 1.0-CVaR (B) objectives, while risk-neutral policies exhibit degraded performance under risk-sensitive evaluation (C).

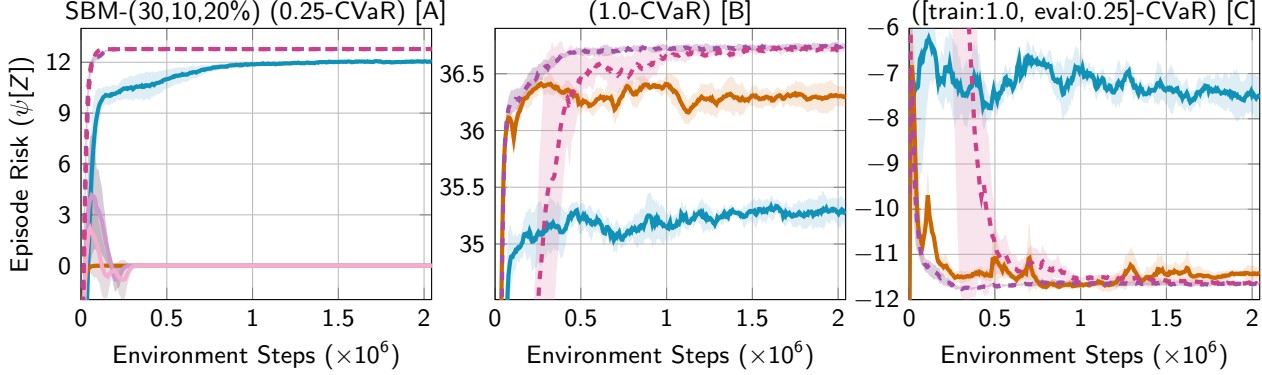

*Figure 12.* **SBM-(30,10,20%)** **results across risk profiles.** *RiskZero* converges to optimal policies under 0.25-CVaR (A) and risk-neutral 1.0-CVaR (B) objectives, while risk-neutral policies exhibit degraded performance under risk-sensitive evaluation (C).

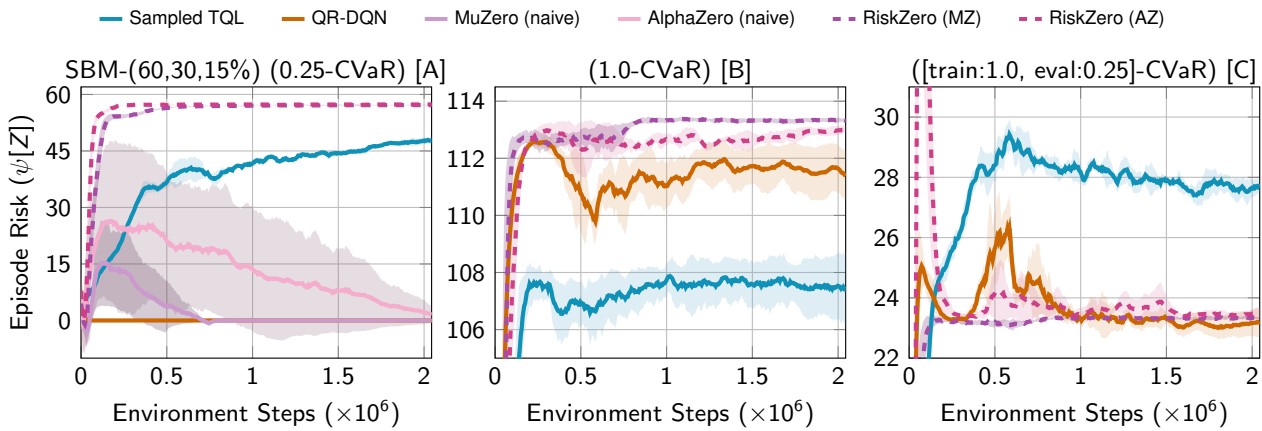

*Figure 13.* **SBM-(60,30,15%) results across risk profiles.** *RiskZero* converges to optimal policies under 0.25-CVaR (A) and risk-neutral 1.0-CVaR (B) objectives, while risk-neutral policies exhibit degraded performance under risk-sensitive evaluation (C).

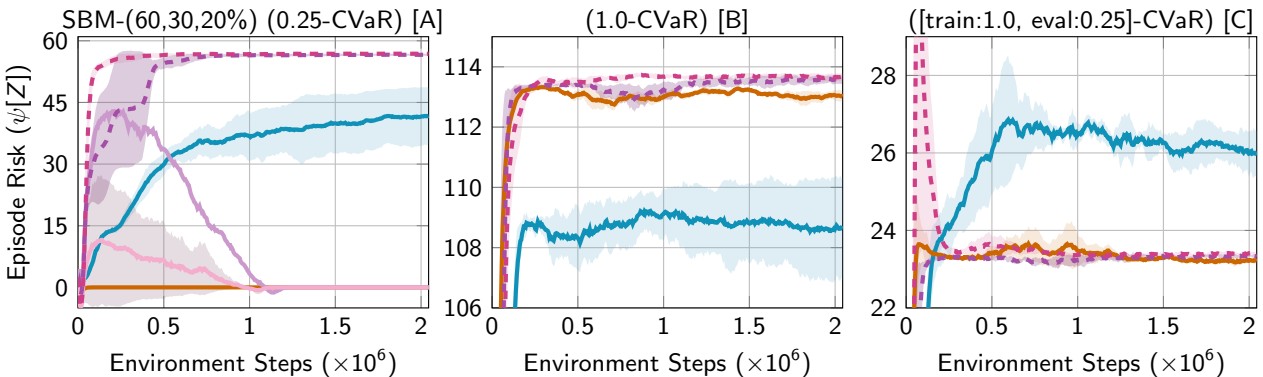

*Figure 14.* **SBM-(60,30,20%) results across risk profiles.** *RiskZero* converges to optimal policies under 0.25-CVaR (A) and risk-neutral 1.0-CVaR (B) objectives, while risk-neutral policies exhibit degraded performance under risk-sensitive evaluation (C).

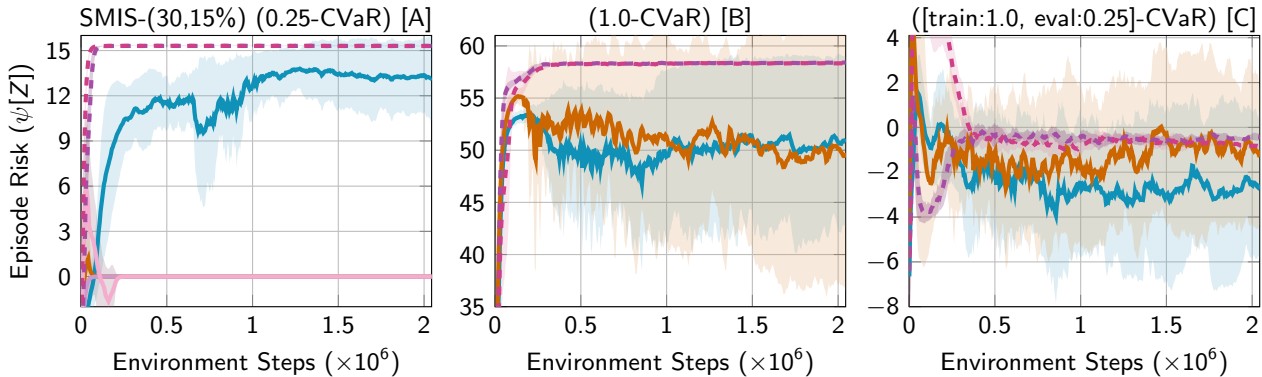

*Figure 15.* **SMIS-(30,15%) results across risk profiles.** *RiskZero* converges to optimal policies under 0.25-CVaR (A) and risk-neutral 1.0-CVaR (B) objectives, while risk-neutral policies exhibit degraded performance under risk-sensitive evaluation (C).

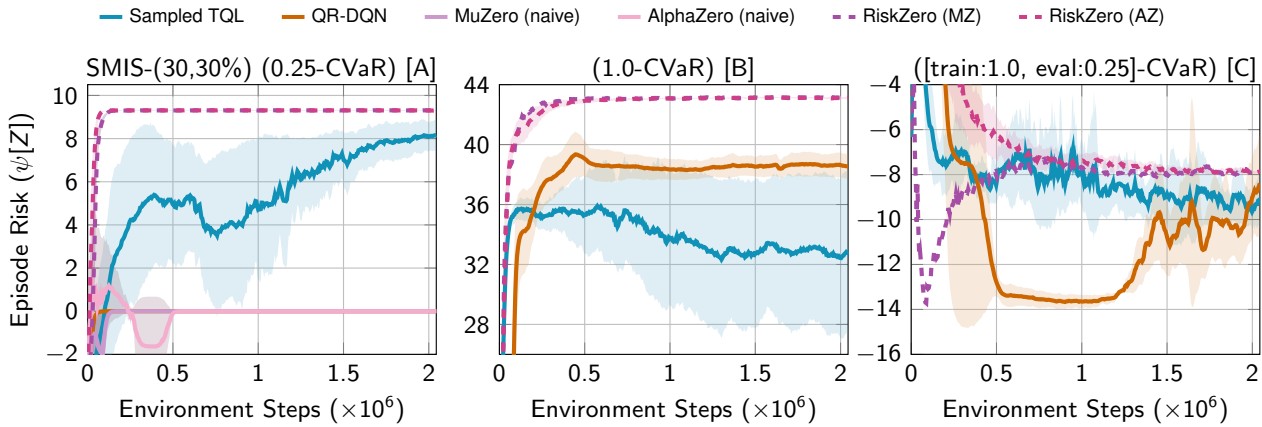

*Figure 16.* **SMIS-(30,30%) results across risk profiles.** *RiskZero* converges to optimal policies under 0.25-CVaR (A) and risk-neutral 1.0-CVaR (B) objectives, while risk-neutral policies exhibit degraded performance under risk-sensitive evaluation (C).

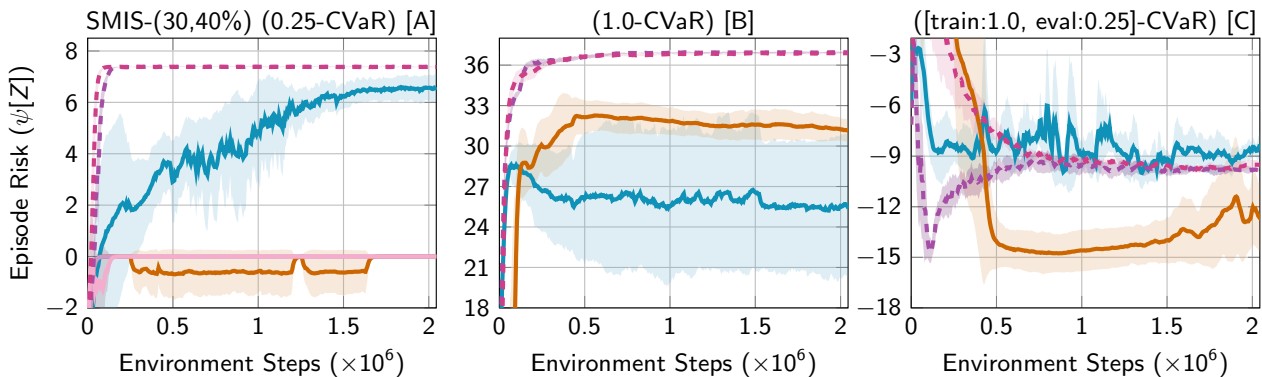

*Figure 17.* **SMIS-(30,40%) results across risk profiles.** *RiskZero* converges to optimal policies under 0.25-CVaR (A) and risk-neutral 1.0-CVaR (B) objectives, while risk-neutral policies exhibit degraded performance under risk-sensitive evaluation (C).

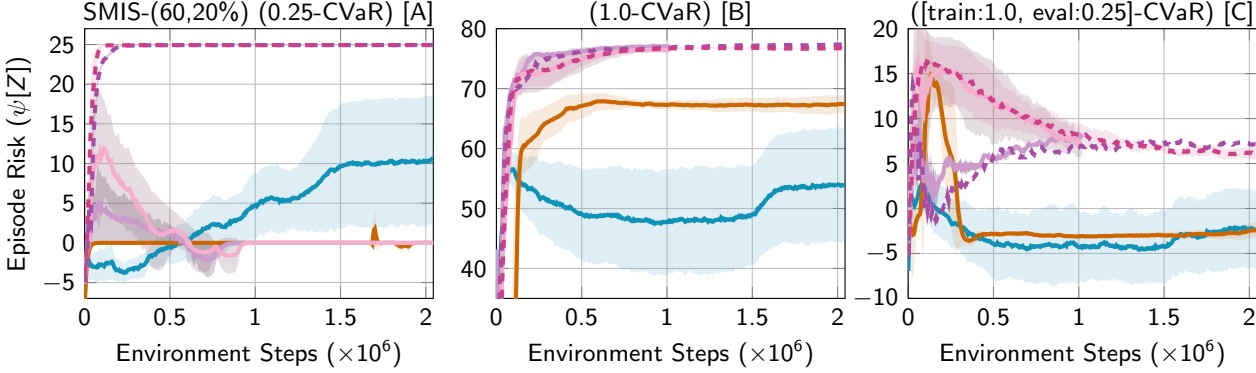

*Figure 18.* **SMIS-(60,20%) results across risk profiles.** *RiskZero* converges to optimal policies under 0.25-CVaR (A) and risk-neutral 1.0-CVaR (B) objectives, while risk-neutral policies exhibit degraded performance under risk-sensitive evaluation (C).

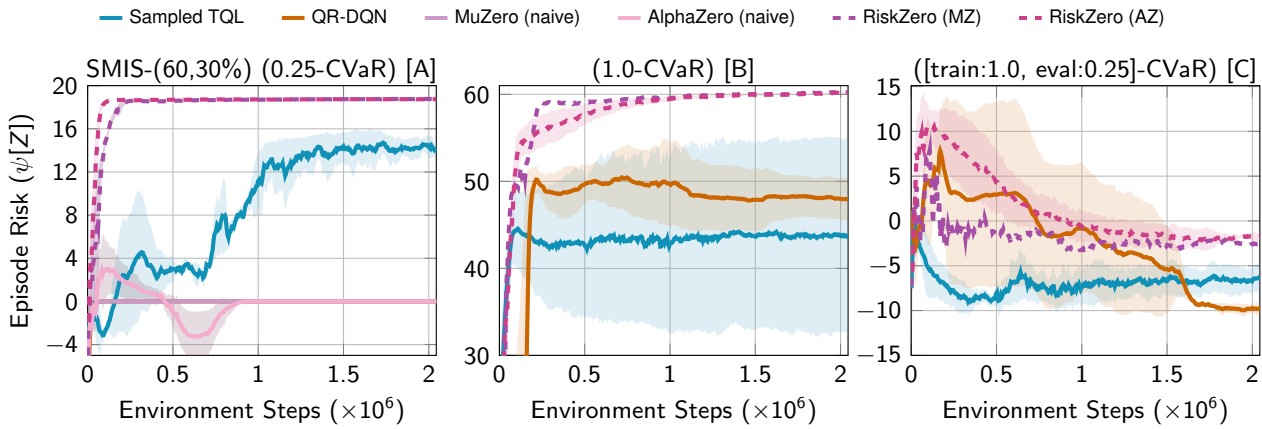

*Figure 19.* **SMIS-(60,30%) results across risk profiles.** *RiskZero* converges to optimal policies under 0.25-CVaR (A) and risk-neutral 1.0-CVaR (B) objectives, while risk-neutral policies exhibit degraded performance under risk-sensitive evaluation (C).

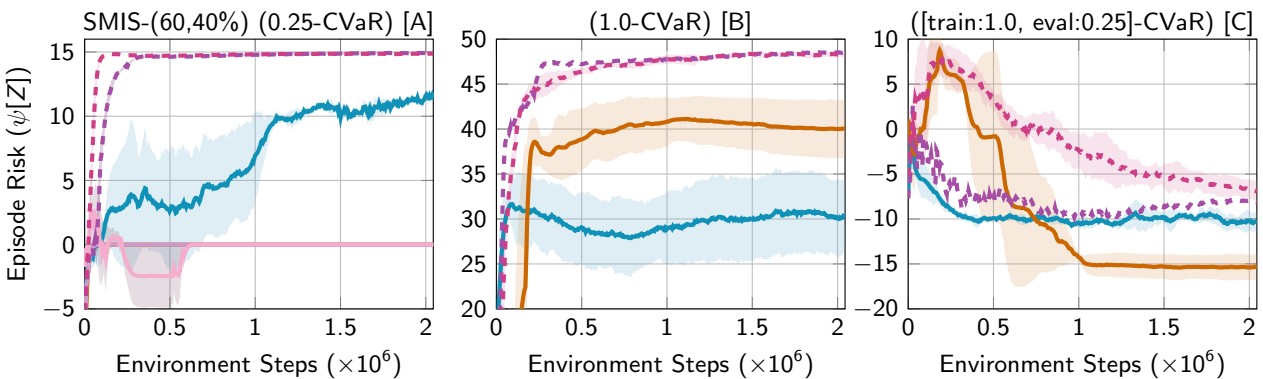

*Figure 20.* **SMIS-(60,40%) results across risk profiles.** *RiskZero* converges to optimal policies under 0.25-CVaR (A) and risk-neutral 1.0-CVaR (B) objectives, while risk-neutral policies exhibit degraded performance under risk-sensitive evaluation (C).

### C.5.3. EXPERIMENTS ON BARABÁSI–ALBERT GRAPHS

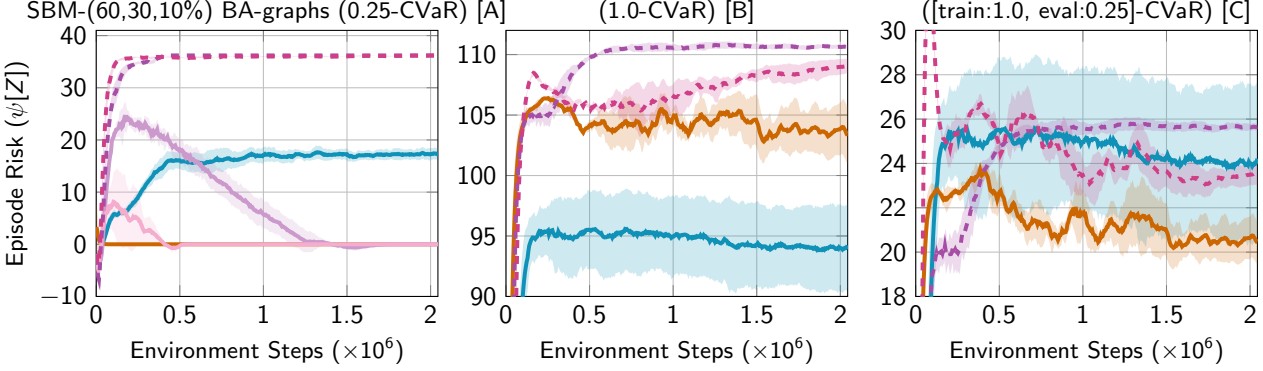

*Figure 21.* **SBM-(60,30,10%) results on BA graphs across risk profiles.** *RiskZero* converges to optimal policies under 0.25-CVaR (A) and risk-neutral 1.0-CVaR (B) objectives, while risk-neutral policies exhibit degraded performance under risk-sensitive evaluation (C).

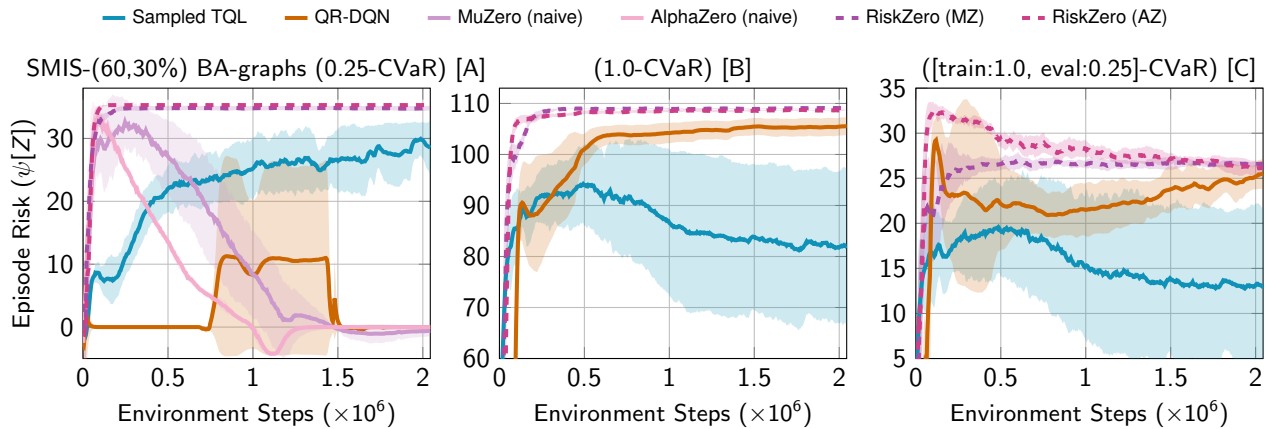

*Figure 22.* **SMIS-(60,30%) results on BA graphs across risk profiles.** *RiskZero* converges to optimal policies under 0.25-CVaR (A) and risk-neutral 1.0-CVaR (B) objectives, while risk-neutral policies exhibit degraded performance under risk-sensitive evaluation (C).

### C.5.4. ADDITIONAL ABLATION RESULTS

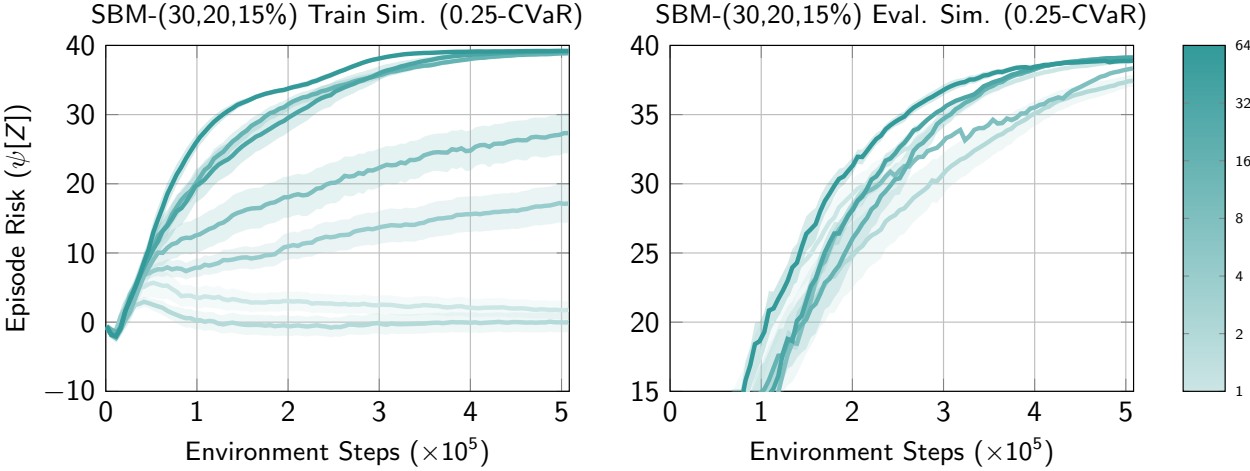

*Figure 23.* **Simulation budget ablation.** Mean episode risk on SBM-(30,20,15%) as a function of the number of MCTS simulations used during training (left; evaluation fixed at 32 simulations) and during evaluation (right; training fixed at 32 simulations). Increasing the simulation budget substantially improves training efficiency (left) and yields modest gains at evaluation time (right), with diminishing returns beyond 16 simulations in both cases.

### C.5.5. VERIFYING THE LEARNED MODEL

We verify the fidelity of the learned model by examining the learned distributions on Mini Grid. Thanks to the environment's simple structure, the ground-truth quantile distributions can be computed exactly, enabling direct comparison.

***RiskZero* learns the correct quantile distributions.** For the risk-averse model, which converges to the path visiting all yellow cells, Fig. 24 shows the learned distributions at the root of the search tree along this trajectory. The top row depicts accumulated return estimates, the second value estimates, and the third the immediate reward for the selected action. Ground-truth quantiles are shown with dashed pink lines. The learned quantiles closely match the ground truth.

The quantiles obtained by unrolling the learned dynamics model from state $x_4$ (the second yellow cell from the bottom) are shown in Fig. 25; see Fig. 26 for unrolling from $x_0$. The final column shows the induced *trajectory return* distribution obtained by sampling from the distributions in each row. Ground-truth trajectory distributions are shown in pink in the last row and again closely match the learned estimates. Importantly, these distributions are produced by the learned dynamics model and used directly for planning. Their accuracy therefore indicates that *RiskZero* learns reliable trajectory-risk estimates

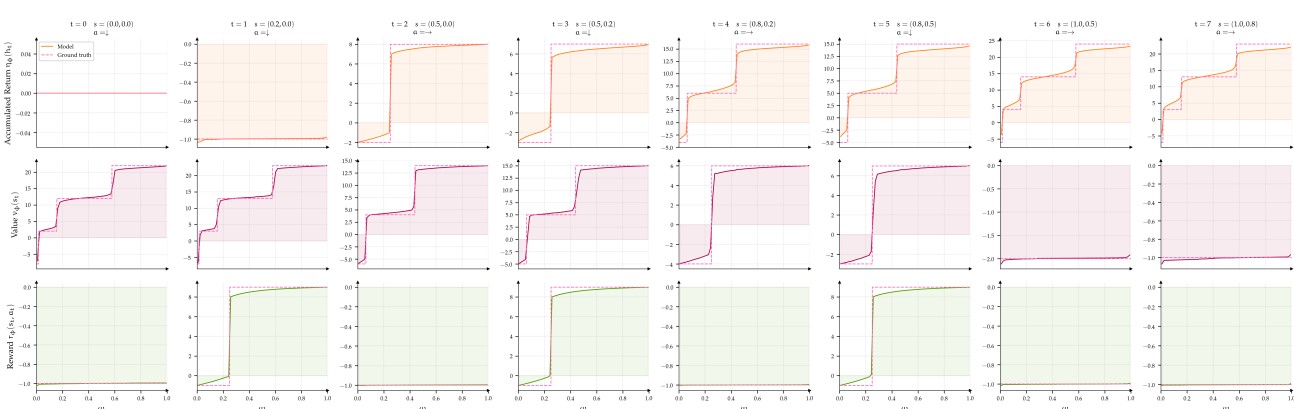

*Figure 24.* **Learned Distributions Along the Risk-Averse Trajectory** ($0.25$**-CVaR),** action sequence $\downarrow, \downarrow, \rightarrow, \downarrow, \rightarrow, \downarrow, \rightarrow, \rightarrow$. Each column corresponds to a state $x_t$ along this trajectory. *Top row:* Accumulated return $\eta_\phi(h_t)$. *Middle row:* State-value distributions $v_\phi(s_t)$. *Bottom row:* Immediate reward $r_\phi(s_t, a_t)$ for the selected action. Dashed pink lines denote ground-truth quantiles; the learned quantiles closely match the ground truth along the frequently visited trajectory.

to guide search toward safer trajectories.

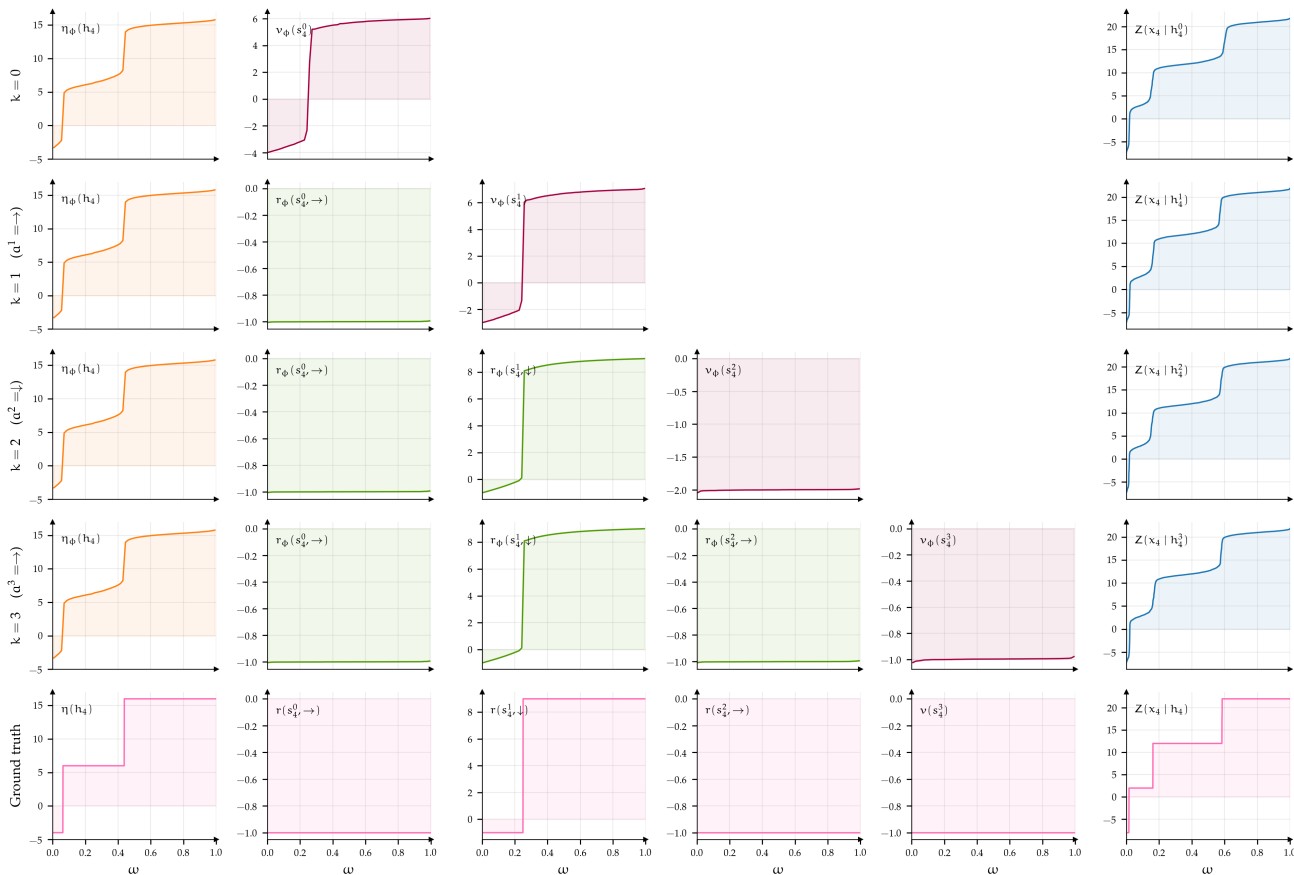

*Figure 25.* **Model Rollout from State** $x_4$**, Risk-Averse Trajectory** ($0.25$**-CVaR).** Each row corresponds to a planning depth $k = 0, \ldots, 3$ from $x_4$, with ground truth in the final row. Columns show the accumulated return $\eta_\phi(h_4)$, per-action reward distributions, state-value $v_\phi(s_4^k)$, and the induced trajectory return $Z(x_4 \mid h_4^k)$ (rightmost), obtained by sampling the distributions on the left. Ground-truth distributions (pink, final row) closely match the learned estimates.

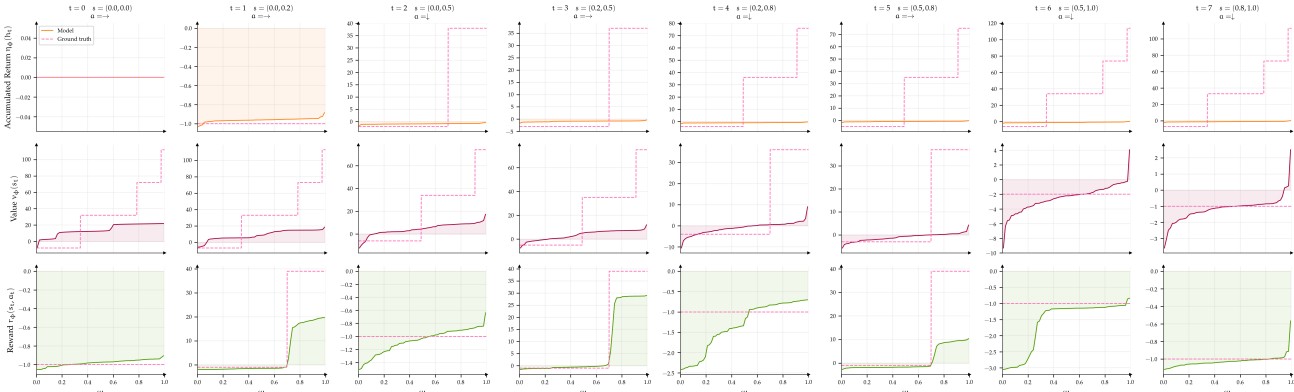

*Figure 26.* **Risk-Averse Model** (0.25-**CVaR**) **Rollout from State** $x_0$**, Risk-Averse Trajectory.** Each row corresponds to a planning depth $k = 0, \dots, 7$, with ground truth in the final row. Columns show the accumulated return $\eta_\phi(h_0)$, per-action reward distributions, state-value $\nu_\phi(s_0^k)$, and trajectory return $Z(x_0 \mid h_0^k)$ (rightmost). Learned estimates closely match the ground truth along the frequently visited trajectory.

*Figure 27.* **Risk-Averse Model** (0.25-**CVaR**) **Evaluated Along the Risk-Neutral Trajectory** (red-cell path, action sequence $\rightarrow, \rightarrow, \downarrow, \rightarrow$ $, \downarrow, \rightarrow, \downarrow, \downarrow$). Each column corresponds to a state $x_t$. *Top row:* Accumulated return $\eta_\phi(h_t)$. *Middle row:* State-value distributions $\nu_\phi(s_t)$. *Bottom row:* Immediate reward $r_\phi(s_t, a_t)$. Dashed pink lines denote ground-truth quantiles for the plotted trajectory. Value distributions deviate from ground truth along this infrequently visited trajectory, while immediate reward distributions remain somewhat accurate.

Quantile estimates are reasonably less accurate in regions of the state space visited less frequently. To illustrate this, we

evaluate the risk-averse model along the optimal *risk-neutral* trajectory (the red-cell path) from $x_4$, shown in Fig. 27. The value distributions deviate from ground truth, which is expected because the learned policy differs from the plotted trajectory. Interestingly, the *immediate reward* distributions remain reasonably accurate across most states.

This suggests that *RiskZero* can leverage accurate reward models to derive improved value estimates through planning, even when state-value predictions are imperfect. Since reward distributions are policy-independent, they may converge earlier and help guide search toward better decisions *faster*, speeding up convergence.

## D. Supporting Lemmas

This section presents lemmas and proofs that are used in the following section to establish the claims made in the main text.

We begin by establishing the convexity of coherent risk measures $\psi$ under *mixtures* of probability distributions, a property we refer to as *mixture convexity*. This property plays a key role in proving the existence of greedy optimal policies under $\psi$. Importantly, convexity under mixtures is fundamentally different from convexity under *sums* of random variables, as the latter corresponds to *aggregating* outcomes rather than *mixing* by randomizing over distributions.

To clarify this distinction, consider two random variables $X \sim [0; 0.5, \ 1; 0.5]$ and $Y \sim [0; 0.5, \ 2; 0.5]$. The *sum* $\lambda X + (1 - \lambda)Y$ follows the convolution of the two distributions, yielding

$$\lambda X + (1 - \lambda)Y \sim [0; 0.25, \ \lambda; 0.25, \ 2(1 - \lambda); 0.25, \ (2 - \lambda); 0.25].$$

Intuitively, this corresponds to drawing samples from *both* distributions and summing the resulting outcomes.

By contrast, *mixtures* operates at the distribution level. A convex mixture of $X$ and $Y$ is obtained by first sampling a Bernoulli variable with parameter $\lambda \in [0, 1]$ to select one of the two distributions, and then drawing a sample from this selection. The resulting mixture distribution is given by $\lambda F_X + (1 - \lambda)F_Y$. In the example above, the $\lambda$-mixture follows

$$X_\lambda Y \sim [0; 0.5, \ 1; 0.5\lambda, \ 2; 0.5(1 - \lambda)].$$

Notably, both the supports and distributions differ from that of the sum. In sequential decision-making, a policy $\pi$ induces a *mixture* over trajectories via randomized action selection. Hence, mixtures are the relevant notion for our analysis.

Both outcomes are realized for sums, hence *diversification* is generally preferred following *super-additivity* and *positive homogeneity* (App. A.3). In contrast, only a single outcome is realized for the mixture. Intuitively, introducing the possibility of a riskier outcome increases overall risk, thereby encouraging greedy mixtures instead. We formalize this intuition below.

**Lemma D.1.** *Let $\psi$ be a coherent risk measure with associated distortion function $g_\psi$. Then $\psi$ is mixture convex: for any gain random variables $Z_1, \ldots, Z_n$ (following laws $F_i$, $i = 1, \ldots, n$) and any convex mixture $Z_\lambda$ following the law*

$$F_\lambda = \sum_{i=1}^{n} \lambda_i F_i, \quad \lambda_i \geq 0, \ \sum_{i=1}^{n} \lambda_i = 1,$$

*it holds that*

$$\psi[Z_\lambda] \leq \sum_{i=1}^{n} \lambda_i \psi[Z_i].$$

*Proof.* From Sereda et al. (2010, Section 3)[3], we have that $\psi[Z]$ admits the following Choquet integral representation

$$\psi[Z] = -\int_{-\infty}^{0} g_\psi(F_Z(z))dz + \int_{0}^{\infty} [1 - g_\psi(F_Z(z))]dz \tag{42}$$

Since $\psi$ is coherent, $g_\psi$ is concave (Cor. A.1). Then, by Jensen's inequality, for any $z$,

$$g_\psi(F_\lambda(z)) \geq \sum_{i=1}^{n} \lambda_i g_\psi(F_i(z)). \tag{43}$$

---

[3]Using the fact that $\psi[Z] = \rho_{\tilde{g}}[-X]$ from Lem. A.2.

Substituting into the Choquet representation yields

$$\psi[Z_\lambda] = -\int_{-\infty}^0 g_\psi(F_\lambda(z))dz + \int_0^\infty \left[1 - g_\psi(F_\lambda(z))\right] dz \tag{44}$$

$$\leq \sum_{i=1}^n \lambda_i \left(-\int_{-\infty}^0 g_\psi(F_i(z))dz + \int_0^\infty \left[1 - g_\psi(F_i(z))\right]dz\right) \tag{45}$$

$$= \sum_{i=1}^n \lambda_i \psi[Z_i], \tag{46}$$

completing the proof. □

An immediate consequence of mixture convexity is the existence of deterministic optimal policies, since any stochastic mixture of actions can do no better than selecting a greedy action.

**Corollary D.1.** *Let $\psi$ be a coherent risk measure, and consider any gain random variables $Z_1, \ldots, Z_n$ and any convex mixture $Z_\lambda$ following the probability law*

$$F_\lambda = \sum_{i=1}^n \lambda_i F_i, \quad \lambda_i \geq 0, \ \sum_{i=1}^n \lambda_i = 1,$$

*it holds that*

$$\psi[Z_\lambda] \leq \max_{1 \leq i \leq n} \psi[Z_i].$$

*Proof.* The result follows directly from the definition of convex sums. □

Our next claims focus on planning with Gumbel. First, we show that a local improvement at a given history propagates to an improvement in expected risk from the initial state, and thus across the entire trajectory.

**Lemma D.2.** *Let $\pi'$ and $\pi$ be policies with different logits at some history $h'_t$ such that $v_\psi^{\pi'}(h'_t) \geq v_\psi^\pi(h'_t)$, and equal logits otherwise, at all other histories $h(\neq h'_t) \in \mathcal{H}$. Then*

$$v_\psi^{\pi'}(x_0) \geq v_\psi^\pi(x_0).$$

*Proof.* Expanding from the initial state, we have from (8)

$$v_\psi^\pi(x_0) = \sum_{a_0 \in \mathcal{A}} \pi(a_0 \mid x_0) \, q_\psi^\pi(x_0, a_0) \tag{47}$$

$$= \sum_{a_0 \in \mathcal{A}} \pi(a_0 \mid x_0) \sum_{a_1 \in \mathcal{A}} \pi(a_1 \mid x_1) \, q_\psi^\pi(h_1, a_1), \tag{48}$$

where $h_1 = x_0 a_0 x_1$ and $x_1 = M(x_0, a_0)$. Unrolling this expansion to time $t$ yields

$$v_\psi^\pi(x_0) = \sum_{a_0 \in \mathcal{A}} \pi(a_0 \mid x_0) \sum_{a_1 \in \mathcal{A}} \pi(a_1 \mid x_1) \cdots \sum_{a_t \in \mathcal{A}} \pi(a_t \mid h_t) \, q_\psi^\pi(h_t, a_t). \tag{49}$$

Equivalently, this can be expressed as an expectation over all partial trajectories

$$v_\psi^\pi(x_0) = \sum_{h_t \in \mathcal{H}_t} d^\pi(h_t) \sum_{a_t \in \mathcal{A}} \pi(a_t \mid h_t) \, q_\psi^\pi(h_t, a_t), \tag{50}$$

where

$$d^\pi(h_t) := \pi(a_0 \mid h_0) \, \pi(a_1 \mid h_1) \cdots \pi(a_{t-1} \mid h_{t-1}), \tag{51}$$

is the probability of observing the partial history $h_t$ under $\pi$.

By assumption, the logits of $\pi'$ differ from $\pi$ only at $h'_t$ so the probabilities $d^\pi(h_t)$ and $\pi(a_t \mid h_t)$, as well as the action values $q^\pi_\psi(h_t, a_t)$ remain unchanged for all $h_t \neq h'_t$. Hence, it suffices to show that

$$\sum_{a_t \in \mathcal{A}} \pi'(a_t \mid h'_t)\, q^{\pi'}_\psi(h'_t, a_t) \geq \sum_{a_t \in \mathcal{A}} \pi(a_t \mid h'_t)\, q^\pi_\psi(h'_t, a_t), \tag{52}$$

which follows from the premise $v^{\pi'}_\psi(h'_t) \geq v^\pi_\psi(h'_t)$, by definition of $v^\pi_\psi$. $\qquad\square$

Next, we show that sequential halving with Gumbel root selection yields a policy improvement in expected risk. The result follows from Danihelka et al. (2022) (Appendix B) and is included here for notational consistency.

> **Lemma D.3.** *Assume accurate risk-sensitive action-values $\hat{q}_\psi(h, a)$. Let $\pi'$ denote the policy induced by applying the Gumbel root selection rule (9). Then the resulting policy satisfies*
> $$v^{\pi'}_\psi(h) \geq v^\pi_\psi(h).$$

*Proof.* By the Gumbel-Max trick (Gumbel, 1954; Luce et al., 1959; Maddison et al., 2016; Jang et al., 2016), the value under policy $\pi$ can be written as

$$v^\pi_\psi(h) = \mathbb{E}_{a \sim \pi}[q^\pi_\psi(h, a)] = \mathbb{E}_{g \sim \mathrm{Gumbel}(0)}\left[q^\pi_\psi\left(h, \arg\max_a\{g(a) + \theta_a\}\right)\right] \tag{53}$$

where $\theta_a$ are the policy $\pi$'s logits. By definition of the Gumbel Top-$K$ trick (Kool et al., 2019), $\mathcal{A}_{\mathrm{topm}}$ consists of the $m$ actions with the largest $g(a) + \theta_a$ and so we may replace $\arg\max_a\{g(a) + \theta_a\}$ by $\arg\max_{a \in \mathcal{A}_{\mathrm{topm}}}\{g(a) + \theta_a\}$. Suppose Gumbel root selection picks action $a_t$, it then suffices to show

$$\mathbb{E}_{a_t \sim \pi'}[q^\pi_\psi(h, a_t)] \geq \mathbb{E}_{g \sim \mathrm{Gumbel}(0)}\left[q^\pi_\psi\left(h, \arg\max_{a \in \mathcal{A}_{\mathrm{topm}}}\{g(a) + \theta_a\}\right)\right]. \tag{54}$$

where the logits of $\pi'$ differ from $\pi$ only at the root, hence we can replace $q^{\pi'}_\psi(h, a_t)$ by $q^\pi_\psi(h, a_t)$ in the expectation on the left-hand side. From (9), the left-hand side is

$$\mathbb{E}_{g \sim \mathrm{Gumbel}(0)}\left[q^\pi_\psi\left(h, \arg\max_{a \in \mathcal{A}_{\mathrm{topm}}}\{g(a) + \theta_a + \sigma(q^\pi_\psi(h, a))\}\right)\right]. \tag{55}$$

Fixing any realization of $g$, it remains to show

$$q^\pi_\psi\left(h, \arg\max_{a \in \mathcal{A}_{\mathrm{topm}}}\{g(a) + \theta_a + \sigma(q^\pi_\psi(h, a))\}\right) \geq q^\pi_\psi\left(h, \arg\max_{a \in \mathcal{A}_{\mathrm{topm}}}\{g(a) + \theta_a\}\right), \tag{56}$$

which holds since $\sigma$ is monotonic increasing and therefore biases the maximization toward actions with larger $q^\pi_\psi(h, a)$. $\quad\square$

We next show that any update toward the improved policy in (11) yields a corresponding improvement in expected risk. The result follows from Danihelka et al. (2022) (Appendix C). We provide an alternative proof that is later helpful to establish Thm. 3.1 (in App. F, notably Lem. F.4), and also generalizes to step sizes other than $\xi = 1$.

> **Lemma D.4.** *Assume accurate risk-sensitive action-values $\hat{q}_\psi(h, a)$. Let $\pi'$ denote a softmax policy with logits updated according to (12). Then the resulting policy satisfies for all $h \in \mathcal{H}, a \in \mathcal{A}$*
> $$v^{\pi'}_\psi(h) \geq v^\pi_\psi(h) \quad \text{and} \quad q^{\pi'}_\psi(h, a) \geq q^\pi_\psi(h, a).$$

*Proof.* We will denote $\text{completed}Q_\psi^\pi(h,a)$ by $\tilde{q}_\psi^\pi(h,a)$.

Under the softmax parameterization and following the update in equation (12)

$$\pi'(a \mid h) = \frac{\exp(\theta_a^{\pi'})}{\sum_b \exp(\theta_b^{\pi'})} = \frac{\pi(a \mid h)\, e^{\xi\sigma\left(\tilde{q}_\psi^\pi(h,a)\right)}}{\sum_b \pi(b \mid h)\, e^{\xi\sigma(\tilde{q}_\psi^\pi(h,b))}}. \tag{57}$$

Let

$$W = \sum_a \pi(a \mid h)\, e^{\xi\sigma\left(\tilde{q}_\psi^\pi(h,a)\right)}, \tag{58}$$

so that

$$v_\psi^{\pi'}(h) = \sum_a \pi'(a \mid h) q_\psi^{\pi'}(h,a) = \frac{\sum_a q_\psi^\pi(h,a)\, \pi(a \mid h)\, e^{\xi\sigma\left(\tilde{q}_\psi^\pi(h,a)\right)}}{W}, \tag{59}$$

noting that $q_\psi^{\pi'}(h,a) = q_\psi^\pi(h,a)$ as the update only affects the action distribution at the current history $h$.

Multiplying by $W$ and subtracting $W v_\psi^\pi(h)$ yields the improvement gap

$$W\left(v_\psi^{\pi'}(h) - v_\psi^\pi(h)\right) = \sum_a q_\psi^\pi(h,a)\, \pi(a \mid h)\, e^{\xi\sigma(\tilde{q}_\psi^\pi(h,a))} - v_\psi^\pi(h) \sum_a \pi(a \mid h)\, e^{\xi\sigma\left(\tilde{q}_\psi^\pi(h,a)\right)} \tag{60}$$

$$= \sum_a \pi(a \mid h)\left(q_\psi^\pi(h,a) - v_\psi^\pi(h)\right) e^{\xi\sigma\left(\tilde{q}_\psi^\pi(h,a)\right)}. \tag{61}$$

Partition the action set into $\mathcal{A}_S = \{a : N(a) > 0\}$ and its complement $\bar{\mathcal{A}}_S$. From the definition of $\tilde{q}_\psi^\pi$ in equation (10), we get

$$\begin{aligned} W\left(v_\psi^{\pi'}(h) - v_\psi^\pi(h)\right) &= \sum_{a \in \mathcal{A}_S} \pi(a \mid h_t)\left(q_\psi^\pi(h,a) - v_\psi^\pi(h)\right) e^{\xi\sigma\left(q_\psi^\pi(h,a)\right)} \\ &\quad + \sum_{a \in \bar{\mathcal{A}}_S} \pi(a \mid h)\left(q_\psi^\pi(h,a) - v_\psi^\pi(h)\right) e^{\xi\sigma\left(v_\psi^\pi(h)\right)}. \end{aligned} \tag{62}$$

Noticing that the sum of advantages $\sum_{a \in \mathcal{A}} \pi(a \mid h)(q_\psi^\pi(h,a) - v_\psi^\pi(h)) = 0$, we have

$$e^{\xi\sigma\left(v_\psi^\pi(h)\right)} \sum_{a \in \bar{\mathcal{A}}_S} \pi(a \mid h_t)\left(q_\psi^\pi(h,a) - v_\psi^\pi(h)\right) = -e^{\xi\sigma\left(v_\psi^\pi(h)\right)} \sum_{a \in \mathcal{A}_S} \pi(a \mid h)\left(q_\psi^\pi(h_t,a) - v_\psi^\pi(h)\right). \tag{63}$$

Substituting this identity into the second sum from (62) gives

$$W\left(v_\psi^{\pi'}(h) - v_\psi^\pi(h)\right) = \sum_{a \in \mathcal{A}_S} \pi(a \mid h)\left(q_\psi^\pi(h,a) - v_\psi^\pi(h)\right)\left(e^{\xi\sigma(q_\psi^\pi(h,a))} - e^{\xi\sigma(v_\psi^\pi(h))}\right). \tag{64}$$

Since $e^{\xi\sigma(\cdot)}$ is strictly increasing in its argument, each factor $(e^{\xi\sigma(q_\psi^\pi(h,a))} - e^{\xi\sigma(v_\psi^\pi(h))})$ has the same sign as $(q_\psi^\pi(h,a) - v_\psi^\pi(h))$; hence every summand is nonnegative and the entire sum is $\geq 0$, with strict positivity unless $q_\psi^\pi(h,a) = v_\psi^\pi(h)$ for all sampled actions. Because $W > 0$, we conclude

$$v_\psi^{\pi'}(h) \geq v_\psi^\pi(h),$$

with strict inequality unless all advantages vanish. Monotonic improvement of action-values then follows from their definition in (8), as each state-action value is defined as the *value* of a successor state, and we have shown that these values are monotonically increasing. We note that one may generalize this argument to stochastic dynamics, by taking an expectation, but our focus is on deterministic dynamics. □

# E. Short Proofs

### E.1. Proof of Prop. 3.1

**Proposition 3.1.** *Let $\psi$ be a coherent risk measure over the return $Z^\pi(x_0)$ of a finite MDP. Then there exists an optimal deterministic policy $\pi^\star$ such that*

$$\psi[Z^{\pi^\star}(x_0)] = \max_\pi \psi[Z^\pi(x_0)].$$

*Proof.* Suppose, toward a contradiction, that no optimal deterministic policy exists. Then any optimal policy $\pi^*$ must mix over actions at some history, inducing a mixture over trajectories but Cor. D.1 shows that deterministically picking the trajectory that achieves the maximum risk does no worse than $\pi^*$, contradicting our initial assumption. □

### E.2. Proof of Prop. 3.2

**Proposition 3.2.** *Let $\pi$ and $\pi'$ be policies. An improvement in expected risk, $v_\psi^{\pi'}(x_0) \geq v_\psi^\pi(x_0)$, does not imply true risk improvement, $\psi[Z^{\pi'}(x_0)] \geq \psi[Z^\pi(x_0)]$.*

*Proof.* We provide a counterexample with $\psi$ as 0.25-CVaR. Consider an MDP with two available actions, $a_0$ and $a_1$, where the episode terminates immediately after the first step. Suppose:

- Action $a_0$ yields reward 0 with probability 0.5 and reward 1 with probability 0.5;
- Action $a_1$ yields reward $-1$ with probability 0.1 and reward 10 with probability 0.9.

Let $\pi$ be a deterministic policy that always selects $a_0$. Its return distribution is supported on $\{0, 1\}$ with equal probability, and hence

$$q_\psi^\pi(x_0, a_0) = \mathrm{CVaR}_{0.25}(Z^\pi(x_0)) = 0,$$

so $v_\psi^\pi(x_0) = 0$.

Since $a_1$ has a higher $\mathrm{CVaR}_{0.25}$ than $a_0$, one might consider increasing its selection probability. Let $\pi'$ be a policy that selects each action with equal probability. The $\mathrm{CVaR}_{0.25}$ of choosing $a_1$ is

$$q_\psi^\pi(x_0, a_1) = \frac{0.1(-1) + 0.15(10)}{0.25} = 5.6.$$

Therefore,

$$v_\psi^{\pi'}(x_0) = 0.5 \cdot 0 + 0.5 \cdot 5.6 = 2.8,$$

which strictly improves over $v_\psi^\pi(x_0) = 0$.

However, the return distribution under $\pi'$ is given by

- $-1$ with probability 0.05,
- 0 with probability 0.25,
- 1 with probability 0.25,
- 10 with probability 0.45.

The worst 25% outcomes consist of the mass at $-1$ (probability 0.05) and a portion of the mass at 0 (probability 0.20). Thus,

$$\psi[Z^{\pi'}(x_0)] = \mathrm{CVaR}_{0.25}(Z^{\pi'}(x_0)) = \frac{0.05(-1) + 0.20(0)}{0.25} = -0.2,$$

which is strictly worse than $\psi[Z^\pi(x_0)] = 0$.

Hence, improving the expected $\psi$-value does not imply an improvement in the CVaR of the full return distribution. □

**E.3. Proof of Prop. 3.3**

> **Proposition 3.3.** *Let $\psi$ be a coherent risk measure and $\Pi_H$ the class of history-stationary policies. A deterministic policy $\pi^* \in \Pi_H$ is optimal in expected risk, $\pi^* \in \arg\max_{\pi \in \Pi_H} v_\psi^\pi(x_0)$, if and only if it is optimal in true risk, $\pi^* \in \arg\max_{\pi \in \Pi_H} \psi[Z^\pi(x_0)]$.*

*Proof.* ($\Leftarrow$) By Prop. 3.1, there exists a deterministic policy that attains $\max_\pi \psi[Z^\pi(x_0)]$; let $\pi^*$ denote such a policy. Since $\pi^*$ is deterministic, it induces a single trajectory $\tau^*$ from $x_0$, and by (7)

$$v_\psi^{\pi^*}(x_0) = \sum_\tau d^{\pi^*}(\tau)\, \psi[Z^{\pi^*}(x_0 \mid \tau)] = \psi[Z^{\pi^*}(x_0 \mid \tau^*)] = \psi[Z^{\pi^*}(x_0)].$$

Hence,

$$v_\psi^{\pi^*}(x_0) = \max_\pi \psi[Z^\pi(x_0)],$$

implying $\pi^* \in \arg\max_\pi v_\psi^\pi(x_0)$ since $v_\psi$ is by definition a convex mixture over possible trajectory risks.

($\Rightarrow$) For the converse, suppose $\pi^* \in \arg\max_\pi v_\psi^\pi(x_0)$, but assume toward contradiction that

$$\pi^* \notin \arg\max_\pi \psi[Z^\pi(x_0)].$$

Let $\pi'$ be a deterministic policy attaining the maximum on the right-hand side (by Prop. 3.1). As above

$$v_\psi^{\pi'}(x_0) = \psi[Z^{\pi'}(x_0)],$$

implying $v_\psi^{\pi'}(x_0) = \psi[Z^{\pi'}(x_0)] > \psi[Z^{\pi^*}(x_0)] = v_\psi^{\pi^*}(x_0)$ which contradicts the optimality of $\pi^*$ for $v_\psi^\pi(x_0)$. We must conclude

$$\pi^* \in \arg\max_\pi \psi[Z^\pi(x_0)],$$

completing the proof. □

**E.4. Proof of Prop. 3.4**

> **Proposition 3.4.** *Let $\pi'$ denote the policy induced by Gumbel selection at the root (9) and the improved policy (11) at non-root nodes, with parameters updated according to (12). Then the resulting policy satisfies*
>
> $$v_\psi^{\pi'}(x_0) \geq v_\psi^\pi(x_0).$$

*Proof.* Improvement at the root follows from Lem. D.3. Improvement at non-root nodes follows from Lem. D.4, with step size $\xi = 1$ giving the result for the improved policy (target) itself. These improvements are for a specific history; applying Lem. D.2 then implies that the resulting policy satisfies

$$v_\psi^{\pi'}(x_0) \geq v_\psi^\pi(x_0).$$

□

# F. Proof of Theorem 3.1

In this section, we prove convergence of *RiskZero* to an *optimal* policy under the expected risk objective and, by Prop. 3.3, also under the $\psi$ objective. Our analysis focuses on *tabular* softmax policy parameterizations, with one parameter $\theta_{h,a}$ for each history-action pair, and closely follows the approach of Agarwal et al. (2020).

**Proof Sketch.** We established monotone improvement in Lem. D.4, but this alone does not guarantee that the limit is optimal. Our goal here is to rule out bad cases in which the policy converges to a suboptimal fixed point. Specifically, we want to exclude situations where, for some history $h$, an action $a_+$ has a strictly higher limiting action-value than the policy's average value, yet $\pi(a_+ \mid h)$ fails to converge to 1.

To do so, we consider a partition of actions at every history by virtue of their limiting advantages

- $\mathcal{I}_+^h$: actions with strictly better advantage in the limit $q_\psi^{(\infty)}(h, a) > v_\psi^{(\infty)}(h)$.
- $\mathcal{I}_-^h$: actions with strictly worse advantage in the limit $q_\psi^{(\infty)}(h, a) < v_\psi^{(\infty)}(h)$.
- $\mathcal{I}_0^h$: the remaining actions with tied limiting values.

Note that if $\mathcal{I}_+^h$ is empty for every $h$, then we are done; every action considered by the policy attains its limiting value, none achieves more, so the limit is optimal. It then remains to rule out the case $\mathcal{I}_+^h \neq \emptyset$ for some $h$, which we do by contradiction.

**Contradiction Strategy.** The argument consists of two threads that must reconcile, but cannot, leading to a contradiction.

Lems. F.4 to F.6 establish a notion of vanishing mass. For the policy to converge, it can no longer change due to advantages. Hence, actions on $\mathcal{I}_+^h$ or $\mathcal{I}_-^h$, where advantages are bounded away from 0 in the limit, must eventually have vanishingly small probability. For $a_+ \in \mathcal{I}_+^h$, this forces $\pi(a_+ \mid h) \to 0$, and this means some *logits* need to go to $\infty$ (see Lem. F.6).

Lems. F.7 to F.11 then focus more on this group of logits that blow up. Assuming $\mathcal{I}_+^h$ is non-empty, we fix some arbitrary $a_+ \in \mathcal{I}_+^h$. We then partition the actions in $\mathcal{I}_0^h$ further, into $\mathcal{B}_0^h(a_+)$—those *never* dominated by $a_+$—and $\overline{\mathcal{B}}_0^h(a_+)$—those that are eventually dominated (i.e., $\pi_\theta^{(t)}(a_+ \mid h) > \pi_\theta^{(t)}(a \mid h)$ for some $t$). Lem. F.8 intuitively shows that all diverging probability mass concentrates on the set of actions that is never dominated $\mathcal{B}_0^h(a_+)$, i.e., $\sum_{a \in \mathcal{B}_0^h(a_+)} \theta_{h,a} \to \infty$.

Concentration of diverging mass on $\mathcal{B}_0^h(a_+)$ means the sum of expected updates over $\mathcal{B}_0^h(a_+)$ must eventually always be *positive*, on average. However, we can show that the combination of (i) $a_+$'s advantages being bounded below by some non-zero quantity, (ii) the support in $\mathcal{I}_-^h$ having vanishing probability, and (iii) advantages on $\overline{\mathcal{B}}_0^h$ vanishing; then the identity $\sum_a \pi(a \mid h) A_\sigma(h, a) = 0$ forces the sum of updates over $\mathcal{B}_0^h$ to eventually always be *negative* on average.

This is our contradiction, completing the proof.

**Assumptions and Notation.** We summarize our assumptions here, and will restate them, as needed, in the statements that appear in our argument below. We assume rewards $R$ are bounded. Then, for all histories $h$ and actions $a$ in a finite-horizon MDP, the return distribution $Z^\pi(h, a)$ has bounded support (one may use $\gamma \in (0, 1)$ for infinite horizons). We further assume that the risk measure $\psi$ preserves boundedness, so that $\psi[Z^\pi(h, a)]$ is always finite. Let $v_\psi^{\max}$ and $q_\psi^{\max}$ denote the corresponding upper bounds, and $v_\psi^{\min}$ and $q_\psi^{\min}$ lower bounds, on the risk value and action-values, respectively. We will also make use of the history visitation distribution defined in (51). Following Agarwal et al. (2020), we assume $d^\pi(h_t) > 0$ for all $h_t \in \mathcal{H}_t$, while leaving open the question of whether this assumption holds in general.

We write $\pi_\theta^{(t)}$ as a *softmax* policy with logits $\theta^{(t)}$. We index the logits (and the policy) with superscripts $t$ to convey change by updates across time. We first provide a useful characterization of the expected parameter update induced by (12).

> **Lemma F.1.** *Let $\pi_\theta^{(t)}$ be a tabular softmax policy, with one parameter $\theta_{h,a}$ for each history-action pair. For the RiskZero algorithm, the expected parameter update at time $t$ satisfies*
>
> $$\theta_{h,a}^{(t+1)} = \theta_{h,a}^{(t)} + \xi \Delta_{\theta_{h,a}} v_\psi^{(t)},$$
>
> *where*
>
> $$\Delta_{\theta_{h,a}} v_\psi^{(t)} = d^{\pi_\theta^{(t)}}(h) \, \pi_\theta^{(t)}(a \mid h) \, A_\sigma^{(t)}(h, a), \qquad A_\sigma^{(t)}(h, a) = \sigma\left(q_\psi^{(t)}(h, a)\right) - \sigma\left(v_\psi^{(t)}(h)\right).$$

*Proof.* Recall the parameter update rule from (12)

$$\theta_{h,a}^{(t+1)} = \theta_{h,a}^{(t)} + \xi \sigma\left(\text{completed}Q_\psi^{(t)}(h, a)\right), \ \xi > 0$$

By translation invariance of the softmax parameterization, we rewrite this update for *sampled* actions $a$ as

$$\theta_{h,a}^{(t+1)} = \theta_{h,a}^{(t)} + \xi \left[\sigma\left(q_\psi^{(t)}(h, a)\right) - \sigma\left(v_\psi^{(t)}(h)\right)\right] = \theta_a^{(t)} + \xi A_\sigma^{(t)}(h, a), \tag{65}$$

while non-sampled actions receive zero update.

Introduce indicator random variables for a single rollout under policy $\pi_\theta$ at time $t$

$$I_h = \mathbf{1}\{\text{history } h \text{ is observed}\}, \qquad I_{a|h} = \mathbf{1}\{\text{action } a \text{ is chosen at } h\}. \tag{66}$$

By the *RiskZero* algorithm, the parameter $\theta_{h,a}^{(t)}$ is incremented by $\xi A_\sigma^{(t)}(h,a)$ if and only if both $I_h = 1$ and $I_{a|h} = 1$. Thus, the update increment for $\theta_{h,a}^{(t)}$ on a single rollout is

$$\Delta\theta_{h,a}^{(t)} = \xi\, I_h\, I_{a|h}\, A_\sigma^{(t)}(h,a). \tag{67}$$

Taking expectation with respect to the rollout distribution induced by $\pi_\theta$ yields

$$\mathbb{E}[\Delta\theta_{h,a}^{(t)}] = \xi\, \mathbb{E}[I_h I_{a|h}]\, A_\sigma^{(t)}(h,a) \tag{68}$$

$$= \xi\, \Pr(I_h = 1, I_{a|h} = 1)\, A_\sigma^{(t)}(h,a) \tag{69}$$

$$= \xi\, d^{\pi_\theta^{(t)}}(h)\, \pi_\theta^{(t)}(a \mid h)\, A_\sigma^{(t)}(h,a), \tag{70}$$

completing the proof. $\qquad\square$

Note that, given bounds on $v_\psi^{(t)}$ and $q_\psi^{(t)}$ exist then $A_\sigma^{(t)}$ is also bounded. We define

$$\Lambda := \max_{h,a,t} \left| A_\sigma^{(t)}(h,a) \right|. \tag{71}$$

Next, we establish convergence for the sequence of risk values $v_\psi^{(t)}$ and action-values $q_\psi^{(t)}$ induced by the updates.

**Lemma F.2.** *Let rewards $R$ be bounded, and assume the risk measure $\psi$ preserves boundedness. Then, for all histories $h$ and actions $a$, there exist limits $v_\psi^{(\infty)}(h)$ and $q_\psi^{(\infty)}(h,a)$ such that, as $t \to \infty$,*

$$v_\psi^{(t)}(h) \to v_\psi^{(\infty)}(h), \qquad q_\psi^{(t)}(h,a) \to q_\psi^{(\infty)}(h,a).$$

*Proof.* By Lem. D.4, the sequence $\{v_\psi^{(t)}(h)\}_{t\geq 0}$ is monotone non-decreasing for every history $h$. Moreover, since $\gamma \in (0,1)$ (or the MDP is finite) and returns are bounded by the assumption of bounded rewards and $\psi$ preserving boundedness, the sequence is uniformly bounded above by some constant $v_\psi^{\max}$. Therefore, by monotone convergence, $v_\psi^{(t)}(h)$ converges to a finite limit $v_\psi^{(\infty)}(h)$. The same argument applies to the sequence $\{q_\psi^{(t)}(h,a)\}_{t\geq 0}$ for each $(h,a)$. $\qquad\square$

**Corollary F.1.** *Let $\sigma$ be a strictly monotonically increasing function and define*

$$\Delta := \min_{h,a:A_\sigma^{(\infty)}(h,a)\neq 0} \left| A_\sigma^{(\infty)}(h,a) \right|. \tag{72}$$

*Then there exists a time $T_0$ such that for all $t \geq T_0$, $h \in \mathcal{H}$, and $a \in \mathcal{A}$,*

$$\sigma\left( q_\psi^{(t)}(h,a) \right) \geq \sigma\left( q_\psi^{(\infty)}(h,a) \right) - \Delta/4. \tag{73}$$

*and*

$$\sigma\left( v_\psi^{(t)}(h) \right) \geq \sigma\left( v_\psi^{(\infty)}(h) \right) - \Delta/4. \tag{74}$$

*Proof.* Since Lem. F.2 guarantees the existence of the limits $q_\psi^{(\infty)}(h,a)$ and $v_\psi^{(\infty)}(h)$, convergence implies that for every $(h,a)$ we can select $T_0$ sufficiently large so that the respective inequalities hold for $q_\psi^{(t)}$ and $v_\psi^{(t)}$. Since $\sigma$ is monotonic increasing, its application maintains inequalities. $\qquad\square$

Based on these limiting values, we introduce the following partitioning of actions for any history $h$

$$\mathcal{I}_0^h = \{a \in \mathcal{A} \mid q_\psi^{(\infty)}(h,a) = v_\psi^{(\infty)}(h)\} \tag{75}$$

$$\mathcal{I}_+^h = \{a \in \mathcal{A} \mid q_\psi^{(\infty)}(h,a) > v_\psi^{(\infty)}(h)\} \tag{76}$$

$$\mathcal{I}_-^h = \{a \in \mathcal{A} \mid q_\psi^{(\infty)}(h,a) < v_\psi^{(\infty)}(h)\} \tag{77}$$

In the lemmas that follow Lem. F.3 - Lem. F.10, we first show that probabilities vanish $\pi_\theta^{(t)}(a \mid h) \to 0$ for actions $a \in \mathcal{I}_+^h \cup \mathcal{I}_-^h$ as $t \to \infty$. We then show that for actions $a \in \mathcal{I}_-^h$, $\lim_{t\to\infty} \theta_{h,a}^{(t)} = -\infty$ and for all actions $a \in \mathcal{I}_+^h$, $\theta^{(t)}(a \mid h)$ is bounded from below as $t \to \infty$.

**Lemma F.3.** *There exists a time $T_1 \geq T_0$ such that*
$$A_\sigma^{(t)}(h, a) < -\Delta/4 \; \forall a \in \mathcal{I}_-^h \quad and \quad A_\sigma^{(t)}(h, a) > \Delta/4 \; \forall a \in \mathcal{I}_+^h.$$

*Proof.* For the first claim, we have for $t \geq T_0$ and $a \in \mathcal{I}_-^h$

$$A_\sigma^{(t)} = \sigma\left(q_\psi^{(t)}(h, a)\right) - \sigma\left(v_\psi^{(t)}(h, a)\right) \tag{78}$$

$$\leq \sigma\left(q_\psi^{(t)}(h, a)\right) - \sigma\left(v_\psi^{(\infty)}(h, a)\right) + \Delta/4 \text{ by (74)} \tag{79}$$

$$\leq \sigma\left(q_\psi^{(\infty)}(h, a)\right) - \sigma\left(v_\psi^{(\infty)}(h, a)\right) + \Delta/4 \text{ by Lem. F.2} \tag{80}$$

$$\leq -\Delta + \Delta/4 \text{ since } a \in \mathcal{I}_-^h, \text{ and (72)} \tag{81}$$

$$\leq -\Delta/4 \tag{82}$$

Similarly, for the second claim, we have for $t > T_0$ and $a \in \mathcal{I}_+^h$

$$A_\sigma^{(t)} = \sigma\left(q_\psi^{(t)}(h, a)\right) - \sigma\left(v_\psi^{(t)}(h, a)\right) \tag{83}$$

$$\geq \sigma\left(q_\psi^{(t)}(h, a)\right) - \sigma\left(v_\psi^{(\infty)}(h, a)\right) \text{ by Lem. F.2} \tag{84}$$

$$\geq \sigma\left(q_\psi^{(\infty)}(h, a)\right) - \Delta/4 - \sigma\left(v_\psi^{(\infty)}(h, a)\right) \text{ by (73) and Lem. F.2} \tag{85}$$

$$\geq \Delta - \Delta/4 \text{ since } a \in \mathcal{I}_+^h, \text{ and (72)} \tag{86}$$

$$\geq \Delta/4 \tag{87}$$

$\square$

**Lemma F.4** (Vanishing Support). *For any $a \in \mathcal{I}_+^h \cup \mathcal{I}_-^h$, we have $\pi_\theta^{(t)}(a \mid h) \to 0$ as $t \to \infty$, which implies*
$$\sum_{a \in \mathcal{I}_0^h} \pi_\theta^{(t)}(a \mid h) \to 1.$$

*Proof.* Suppose, for sake of contradiction, that there exists some $a \in \mathcal{I}_+^h \cup \mathcal{I}_-^h$ such that $\lim_{t\to\infty} \pi_\theta^{(t)}(a \mid h) \neq 0$. Then, there is exists some infinitely long subsequence of timesteps $\{t_k\} \subseteq \{t\}$ for which $\pi_\theta^{(t_k)}(a \mid h) \geq \kappa > 0$ for all $t_k$.

From Cor. F.1, we have for $t \geq T_0$

$$\left|\sigma\left(q_\psi^{(t)}(h, a)\right) - \sigma\left(v_\psi^{(t)}(h_t)\right)\right| \geq \Delta, \tag{88}$$

which implies

$$\left|q_\psi^{(t)}(h, a) - v_\psi^{(t)}(h_t)\right| \geq \Delta_\psi \quad and \quad \left|e^{\xi\sigma\left(q_\psi^{(t)}(h,a)\right)} - e^{\xi\sigma\left(v_\psi^{(t)}(h)\right)}\right| \geq \Delta_e \tag{89}$$

for some constants $\Delta_\psi, \Delta_e > 0$, by strict monotonicity of $\sigma$.

Now recall the following identity from (64)

$$W\left(v_\psi^{\pi'}(h) - v_\psi^\pi(h)\right) = \sum_{a \in \mathcal{A}_S} \pi(a \mid h)\left(q_\psi^\pi(h, a) - v_\psi^\pi(h)\right)\left(e^{\xi\sigma\left(q_\psi^\pi(h,a)\right)} - e^{\xi\sigma\left(v_\psi^\pi(h)\right)}\right).$$

Consider the contribution of the fixed action $a$ to this sum whenever it is sampled during the subsequence of timesteps $\{t_k\}$. Since $\pi_\theta^{(t_k)}(a \mid h_t) \geq \kappa$, we have

$$\pi_\theta^{(t)}(a \mid h_t)\left(q_\psi^{(t)}(h, a) - v_\psi^{(t)}(h)\right)\left(e^{\xi\sigma\left(q_\psi^{(t)}(h,a)\right)} - e^{\xi\sigma\left(v_\psi^{(t)}(h)\right)}\right) \geq \kappa\Delta_\psi\Delta_e.$$

From the definition of $W$ in (58), we can upper bound by the optimal value $W \leq e^{\sigma(v_\psi^*(h))}$, and so it follows that

$$v_\psi^{(t+1)}(h) - v_\psi^{(t)}(h) \geq \frac{\kappa \Delta c}{e^{\sigma(v_\psi^*(h))}},$$

whenever $a$ is sampled. Recall that all the terms in the sum (64) are positive since $\pi(\cdot) > 0$, and both other factors in the summand have the same sign. We are only considering one of these terms, which maintains the inequality.

Since $\pi_\theta^{(t_k)}(a \mid h) \geq \kappa > 0$ for all $t_k$ and actions are sampled from $\pi$ independently, then by the second Borel Cantelli Lemma, the action $a$ is sampled infinitely often. Let $\{t_j\}$ be the subsequence of times where $a$ is sampled. Then, for every $j$

$$v_\psi^{(t_j+1)} - v_\psi^{(t_j)} \geq \frac{\kappa \Delta_\psi \Delta_e}{e^{\sigma(v_\psi^*)}}.$$

Summing over $j$ gives

$$\sum_{j=1}^{\infty} \left( v_\psi^{(t_j+1)} - v_\psi^{(t_j)} \right) \geq \sum_{j=1}^{\infty} \frac{\kappa \Delta_\psi \Delta_e}{e^{\sigma(v_\psi^*)}},$$

where the left-hand side telescopes, giving

$$v_\psi^{(\infty)} - v_\psi^{(0)} \geq \sum_{j=1}^{\infty} \frac{\kappa \Delta_\psi \Delta_e}{e^{\sigma(v_\psi^*)}}.$$

But the right-hand side diverges, while $v_\psi^{(t)}$ is bounded above by $v_\psi^*$, leading to a contradiction. Hence, our assumption must be false. There is no infinitely long subsequence $\{t_k\}$ of timesteps where $\pi_\theta^{(t)}(a \mid h)$ is non-zero, and so we conclude that

$$\pi_\theta^{(t)}(a \mid h) \to 0 \quad \text{for all } a \in \mathcal{I}_+^h \cup \mathcal{I}_-^h \text{ as } t \to \infty.$$

Since $\pi_\theta^{(t)}$ is a probability distribution over $\mathcal{A} = \mathcal{I}_-^h \cup \mathcal{I}_+^h \cup \mathcal{I}_0^h$, and the mass on $\mathcal{I}_-^h \cup \mathcal{I}_+^h$ vanishes, then it follows that

$$\sum_{a \in \mathcal{I}_0^h} \pi_\theta^{(t)}(a \mid h) \to 1 \text{ as } t \to \infty.$$

$\square$

---

**Lemma F.5** (Monotonicity in $\theta$). *Suppose $d^{\pi_\theta^{(t)}}(h) > 0$ for all histories $h$ and times $t$. Then, $\theta_{h,a}^{(t)}$ is strictly increasing for $a \in \mathcal{I}_+^h$ and strictly decreasing for $a \in \mathcal{I}_-^h$.*

*Proof.* From the update rule, we have in expectation that

$$\theta_{h,a}^{(t+1)} \leftarrow \theta_{h,a}^{(t)} + \xi \Delta_{\theta_{h,a}} v_\psi^{(t)}.$$

From Lem. F.3, we have for $t \geq T_1$

$$A_\sigma^{(t)}(h, a) > 0 \ \forall a \in \mathcal{I}_+^h \quad \text{and} \quad A_\sigma^{(t)}(h, a) < 0 \ \forall a \in \mathcal{I}_-^h.$$

By assumption that $d^{\pi_\theta^{(t)}}(h) > 0$, and since $\pi_\theta^{(t)}(a \mid h) > 0$ for the softmax parameterization, then

$$d^{\pi_\theta^{(t)}}(h) \pi_\theta^{(t)}(a \mid h) A_\sigma^{(t)}(h, a) > 0,$$

for all $h \in \mathcal{H}$, and actions $a \in \mathcal{I}_+^h$, which implies $\theta_{h,a}^{(t+1)} - \theta_{h,a}^{(t)} = \xi d^{\pi_\theta^{(t)}}(h) \pi_\theta^{(t)}(a \mid h) A_\sigma^{(t)}(h, a) > 0$ to establish the first claim. The second claim follows similarly with strictly negative updates on $\mathcal{I}_-^h$. $\square$

**Lemma F.6.** *Assume RiskZero is run with positive linear isomorphism $\sigma$. If $\mathcal{I}_+^h \neq \emptyset$, we have $\max_{a \in \mathcal{I}_0^h} \theta_{h,a}^{(t)} \to \infty$ and $\min_{a \in \mathcal{A}} \theta_a^{(t)} \to -\infty$.*

*Proof.* By Lem. F.4, there exists $a_+ \in \mathcal{I}_+^h$ such that $\pi_\theta^{(t)}(a_+ \mid h) \to 0$ for $t \to \infty$. Equivalently

$$\frac{\exp(\theta_{a_+}^{(t)})}{\sum_{b \in \mathcal{A}} \exp(\theta_b^{(t)})} \to 0.$$

By Lem. F.5, $\theta_{a_+}^{(t)}$ is monotonically increasing which, coupled with the above, implies the denominator must diverge

$$\sum_{b \in \mathcal{A}} \exp(\theta_b^{(t)}) \to \infty.$$

By Lem. F.4, we have $\sum_{a \in \mathcal{I}_0^h} \pi_\theta^{(t)}(a \mid h) \to 1$, or equivalently

$$\frac{\sum_{a \in \mathcal{I}_0^h} \exp(\theta_a^{(t)})}{\sum_{b \in \mathcal{A}} \exp(\theta_b^{(t)})} \to 1,$$

from which it follows that

$$\sum_{a \in \mathcal{I}_0^h} \exp(\theta_a^{(t)}) \to \infty,$$

since the denominator diverges. Hence

$$\max_{a \in \mathcal{I}_0^h} \theta_{h,a}^{(t)} \to \infty.$$

For the second claim, we have $\sigma(x) = Bx$ for some $B > 0$ by assumption that $\sigma$ is a positive linear isomorphism. As a result, we have that the sum of expected updates over all actions is zero

$$\sum_{a \in \mathcal{A}} d^{\pi_\theta}(h) \pi_\theta^{(t)}(a \mid h) A_\sigma^{(t)}(h, a) = d^{\pi_\theta}(h) B \sum_{a \in \mathcal{A}} \pi_\theta^{(t)}(a \mid h) \left( q_\psi^{(t)}(h, a) - v_\psi^{(t)}(h) \right) = 0.$$

From updates summing to zero, we have for some $c \in \mathbb{R}$

$$\sum_{a \in \mathcal{A}} \theta_a^{(t)} = \sum_{a \in \mathcal{A}} \theta_a^{(0)} = c,$$

hence

$$(|\mathcal{A}| - 1) \min_a \theta_a^{(t)} + \max_a \theta_a^{(t)} \leq c,$$

and so $\min_a \theta_a^{(t)} \to -\infty$ since $\max_a \theta_a^{(t)} \to \infty$. $\square$

For all arguments that follow, namely Lem. F.11, we implicitly assume $\sigma$ is a positive linear isomorphism.

**Lemma F.7** (Persistent Dominance)**.** *Suppose $a_+ \in \mathcal{I}_+^h$. For any $a \in \mathcal{I}_0^h$, if there exists $t \geq T_0$ such that $\pi_\theta^{(t)}(a \mid h) \leq \pi_\theta^{(t)}(a_+ \mid h)$ then $\forall \varsigma > t : \pi_\theta^{(\varsigma)}(a \mid h) \leq \pi_\theta^{(\varsigma)}(a_+ \mid h)$.*

*Proof.* We proceed by induction on $t$. Suppose $\pi_\theta^{(t)}(a \mid h) \leq \pi_\theta^{(t)}(a_+ \mid h)$ then we have

$$\begin{aligned}
\Delta_{\theta_{h,a}} v_\psi^{(t)} &= d^{\pi_\theta}(h) \pi_\theta^{(t)}(a \mid h) \left( \sigma \left( q_\psi^{(t)}(h, a) \right) - \sigma \left( v_\psi^{(t)}(h) \right) \right) \\
&\leq d^{\pi_\theta}(h) \pi_\theta^{(t)}(a_+ \mid h) \left( \sigma \left( q_\psi^{(t)}(h, a_+) \right) - \sigma \left( v_\psi^{(t)}(h) \right) \right) \\
&\leq \Delta_{\theta_{h,a_+}} v_\psi^{(t)}
\end{aligned}$$

where the second step follows from $t \geq T_0$ so that

$$
\begin{aligned}
\sigma\left(q_\psi^{(t)}(h, a_+)\right) &\geq \sigma\left(q_\psi^{(\infty)}(h, a_+)\right) - \Delta/4 \text{ by (73)} \\
&\geq \sigma\left(v_\psi^{(\infty)}(h)\right) + \Delta - \Delta/4 \text{ by definition of } \mathcal{I}_+^h, \text{ and } \Delta \\
&\geq \sigma\left(q_\psi^{(\infty)}(h, a)\right) \text{ by definition of } \mathcal{I}_0^h \\
&\geq \sigma\left(q_\psi^{(t)}(h, a)\right)
\end{aligned}
$$

which implies that $\pi_\theta^{(t+1)}(a \mid h) \leq \pi_\theta^{(t+1)}(a_+ \mid h)$, yielding the result. $\qquad\square$

Now consider $a_+ \in \mathcal{I}_+^h$ arbitrary and partition $\mathcal{I}_0^h$ into $\mathcal{B}_0^h(a_+)$ and $\overline{\mathcal{B}}_0^h(a_+)$ as follows

$$
\begin{aligned}
\mathcal{B}_0^h(a_+) &:= \text{all } a \in \mathcal{I}_0^h \text{ such that } \forall t \geq T_0 : \pi_\theta^{(t)}(a_+ \mid h) < \pi_\theta^{(t)}(a \mid h), \\
\overline{\mathcal{B}}_0^h(a_+) &:= \mathcal{I}_0^h \setminus \mathcal{B}_0^h(a_+).
\end{aligned}
$$

So $\mathcal{B}_0^h(a_+)$ contains all $a \in \mathcal{I}_0^h$ which are never dominated by $a_+$. We write $\mathcal{B}_0^h$ and $\overline{\mathcal{B}}_0^h$ when the context is clear.

**Lemma F.8.** *Suppose $\mathcal{I}_+^h \neq \emptyset$. For all $a_+ \in \mathcal{I}_+^h$, $\mathcal{B}_0^h(a_+) \neq \emptyset$, and $\sum_{a \in \mathcal{B}_0^h} \pi_\theta^{(t)}(a \mid h) \to 1$ as $t \to \infty$. This implies $\max_{a \in \mathcal{B}_0^h} \theta_{h,a}^{(t)} \to \infty$.*

*Proof.* Let $a_+ \in \mathcal{I}_+^h$. If $\overline{\mathcal{B}}_0^h$ is empty, then we are done. Otherwise, consider arbitrary $a \in \overline{\mathcal{B}}_0^h$. By definition, there exists $t' > T_0$ such that $\pi_\theta^{(t')}(a_+ \mid h) \geq \pi_\theta^{(t')}(a \mid h)$. Hence, by Lem. F.7

$$
\forall \varsigma > t' : \pi_\theta^{(\varsigma)}(a_+ \mid h) \geq \pi_\theta^{(\varsigma)}(a \mid h),
$$

From which it follows that $\forall a \in \overline{\mathcal{B}}_0^h : \pi_\theta^{(t)}(a \mid h) \to 0$ since $\pi_\theta^{(t)}(a_+ \mid h) \to 0$ from Lem. F.4. Hence,

$$
\sum_{a \in \overline{\mathcal{B}}_0^h} \pi_\theta^{(t)}(a \mid h) \to 0,
$$

but $\mathcal{I}_0^h = \mathcal{B}_0^h \cup \overline{\mathcal{B}}_0^h$ and, from Lem. F.4, we also have $\sum_{a \in \mathcal{I}_0^h} \pi_\theta^{(t)}(a \mid h) \to 1$ which implies $\mathcal{B}_0^h \neq \emptyset$ and further that

$$
\sum_{a \in \mathcal{B}_0^h} \pi_\theta^{(t)}(a \mid h) \to 1.
$$

This implies $\max_{a \in \mathcal{B}_0^h} \theta_{h,a}^{(t)} \to \infty$ by the same argument from Lem. F.6. $\qquad\square$

**Lemma F.9.** *Consider any $h$ with $\mathcal{I}_+^h \neq \emptyset$, then for arbitrary $a_+ \in \mathcal{I}_+^h$, there exists an iteration $T_{a_+}$ such that $\forall t > T_{a_+}, a \in \overline{\mathcal{B}}_0^h(a_+) : \pi_\theta^{(t)}(a_+ \mid h) > \pi(a \mid h)$.*

*Proof.* By definition of $\overline{\mathcal{B}}_0^h$, we have $a \in \overline{\mathcal{B}}_0^h$ implying existence of some $t_a > T_0$ such that $\pi_\theta^{(t_a)}(a \mid h) < \pi_\theta^{(t_a)}(a_+ \mid h)$, and by Lem. F.7

$$
\forall t > t_a : \pi_\theta^{(t)}(a \mid h) < \pi_\theta^{(t)}(a_+ \mid h),
$$

Since this holds for all $a \in \overline{\mathcal{B}}_0^h$, we take

$$
T_a = \max_{a \in \mathcal{B}_0^h(a_+)} t_a,
$$

to complete the proof. $\qquad\square$

**Lemma F.10.** *For all actions $a \in \mathcal{I}_+^h$, we have $\theta_{h,a}^{(t)}$ is bounded from below as $t \to \infty$. For all actions $a \in \mathcal{I}_-^h$, we have $\theta_{h,a}^{(t)} \to -\infty$.*

*Proof.* For the first claim, we have from Lem. F.5 that after $T_1$: $\theta_{h,a}^{(t)}$ is strictly increasing for $a \in \mathcal{I}_+^h$ so $\forall t > T_1 : \theta_{h,a}^{(t)} \geq \theta_{h,a}^{(T_1)}$ establishing the lower bound.

For the second claim, we proceed by contradiction. We have from Lem. F.5 that $\theta_{h,a}^{(t)}$ is strictly decreasing for $a \in \mathcal{I}_-^h$ after $T_1$. By monotonone convergence theorem, $\lim_{t \to \infty} \theta_{h,a}^{(t)}$ exists and is either some constant $\theta_0$ or $-\infty$. For sake of contradiction, let $a \in \mathcal{I}_-^h$ and suppose there exists $\theta_0$ for which $\forall t \geq T_1 : \theta_{h,a}^{(t)} \geq \theta_0$ holds.

Recall from Lem. F.6, there exists $a' \in \mathcal{A}$ for which $\liminf_{t \to \infty} \theta_{h,a'}^{(t)} = -\infty$. Consider $\delta > 0$ such that $\theta_{h,a'}^{(T_1)} \geq \theta_0 - \delta$. For $t \geq T_1$, define $\varsigma(t) = k$ where $k$ is the latest time in $[T_1, t]$ such that $\theta_{h,a'}^{(k)} \geq \theta_0 - \delta$. Define $\mathcal{T}^{(t)}$ as the subsequence of iterations $\varsigma(t) < t' < t$ such that $\theta_{h,a'}^{(t')}$ decreases, i.e., $\Delta_{\theta_{h,a'}} v_\psi^{(t')} \leq 0$ for $t' \in \{\varsigma(t) + 1, \ldots, t - 1\}$. Next, define

$$Z_t = \sum_{t' \in \mathcal{T}^{(t)}} \Delta_{\theta_{h,a'}} v_\psi^{(t)},$$

with $Z_t = 0$ if $\mathcal{T}^{(t)} = \emptyset$. For non-empty $\mathcal{T}^{(t)}$, we have

$$
\begin{aligned}
Z_t &= \sum_{t' \in \mathcal{T}^{(t)}} \Delta_{\theta_{h,a'}} v_\psi^{(t)} \\
&\leq \sum_{t' = \varsigma(t)+1}^{t-1} \Delta_{\theta_{h,a'}} v_\psi^{(t)} \text{ since } \mathcal{T}^{(t)} \text{ contains strictly negative update entries} \\
&\leq \sum_{t' = \varsigma(t)}^{t-1} \Delta_{\theta_{h,a'}} v_\psi^{(t)} + |\Gamma| \text{ where } \Gamma \text{ is the maximum (negative) update due to boundedness of } A_\sigma^{(t)} \\
&\leq \frac{1}{\xi} \sum_{t' = \varsigma(t)}^{t-1} \left[ \theta_{h,a'}^{(t'+1)} - \theta_{h,a'}^{(t')} \right] + |\Gamma| \text{ by definition of the update (12)} \\
&\leq \frac{1}{\xi} \left( \theta_{h,a'}^{(t)} - \theta_{h,a'}^{\varsigma(t)} \right) + |\Gamma| \text{ by telescoping sum} \\
&\leq \frac{1}{\xi} \left( \theta_{h,a'}^{(t)} - (\theta_0 - \delta) \right) + |\Gamma| \text{ since } \theta_{h,a'}^{(\varsigma(t))} \geq \theta_0 - \delta.
\end{aligned}
$$

By $\theta_{h,a'}^{(t)} \to -\infty$, then $\mathcal{T}^{(t)} \neq \emptyset$ eventually and $Z_t \to -\infty$.

Remark that for $t' \in \mathcal{T}^{(t)}$

$$
\begin{aligned}
\left| \frac{\Delta_{\theta_{h,a}} v_\psi^{(t')}}{\Delta_{\theta_{h,a'}} v_\psi^{(t')}} \right| &= \left| \frac{\pi_\theta^{(t')}(a \mid h) A_\sigma^{(t')}(h, a)}{\pi_\theta^{(t')}(a' \mid h) A_\sigma^{(t')}(h, a')} \right| \\
&\geq \exp\left( \theta_0 - \theta_{h,a'}^{(t')} \right) \left| \frac{\sigma\left( q_\psi^{(t')}(h, a) \right) - \sigma\left( v_\psi^{(t')}(h) \right)}{\sigma\left( q_\psi^{(t')}(h, a') \right) - \sigma\left( v_\psi^{(t')}(h) \right)} \right| \text{ by assumption that } \theta_{h,a}^{(t')} \geq \theta_0 \text{ for } t' \geq T_1 \\
&\geq \exp(\delta) \left| \frac{\Delta/4}{\Lambda} \right| \text{ since } a \in \mathcal{I}_-^h \text{ and Lem. F.3, and using the bound on } A_\sigma^{(t')} \text{ from (71)}
\end{aligned}
$$

where the last inequality uses the fact that for all $t' \in \mathcal{T}^{(t)}$, $\theta_{h,a'}^{(t')}$ strictly less than $\theta_0 - \delta$, by definition of $\varsigma(t)$, and decreasing.

Hence, we have $\theta_{h,a'}^{(t')} \leq \theta_0 - \delta \Rightarrow \theta_0 - \theta_{h,a'}^{(t')} \geq \delta$.

Since $\Delta_{\theta_{h,a}} v_\psi^{(t')} < 0$ by $a \in \mathcal{I}_-^h$ for $t' \geq T_1$ (Lem. F.5), and $\Delta_{\theta_{h,a'}} v_\psi^{(t)} < 0$ for all $t' \in \mathcal{T}^{(t)}$ then the inequality reverses

$$\Delta_{\theta_{h,a}} v_\psi^{(t')} \leq \exp(\delta) \frac{\Delta}{4\Lambda} \Delta_{\theta_{h,a'}} v_\psi^{(t')}, \tag{90}$$

which yields

$$\begin{aligned}
\frac{1}{\xi} \left( \theta_{h,a}^{(t)} - \theta_{h,a}^{(T_1)} \right) &= \sum_{t'=T_1}^{t-1} \Delta_{\theta_{h,a}} v_\psi^{(t')} \\
&\leq \sum_{t' \in \mathcal{T}^{(t)}}^{t-1} \Delta_{\theta_{h,a}} v_\psi^{(t')} \text{ since } \Delta_{\theta_{h,a}} v_\psi^{(t')} < 0 \text{ } \textit{always} \text{ } \forall t' > T_1 \text{ as } a \in \mathcal{I}_-^h \text{ (Lem. F.5)} \\
&\leq \exp(\delta) \frac{\Delta}{4\Lambda} \sum_{t' \in \mathcal{T}^{(t)}} \Delta_{\theta_{h,a'}} v_\psi^{(t')} \text{ from (90)} \\
&\leq \exp(\delta) \frac{\Delta}{4\Lambda} Z_t \text{ by definition of } Z_t
\end{aligned}$$

which implies $\theta_{h,a}^{(t)} \to -\infty$ since $Z_t \to -\infty$, and contradicts the assumption of a lower bound. □

**Lemma F.11.** *Consider any $h$ with $\mathcal{I}_+^h \neq \emptyset$, then for any $a_+ \in \mathcal{I}_+^h$ we have*

$$\sum_{a \in \mathcal{B}_0^h(a_+)} \theta_{h,a}^{(t)} \to \infty \text{ as } t \to \infty.$$

*Proof.* From Lem. F.8, $\mathcal{B}_0^h$ is non-empty so consider $a \in \mathcal{B}_0^h$ then by definition $\forall t > T_0 : \pi_\theta^{(t)}(a \mid h) > \pi_\theta^{(t)}(a_+ \mid h)$. By definition of softmax, it follows that $\theta_{h,a}^{(t)} \geq \theta_{h,a_+}^{(t)}$. From Lem. F.10, $\theta_{h,a_+}^{(t)}$ is bounded from below so $\theta_{h,a}^{(t)}$ is as well.

Since $\theta_{h,a}^{(t)}$ is bounded from below, for all $a \in \mathcal{B}_0^h$, and we have $\max_{a \in \mathcal{B}_0^h} \theta_{h,a}^{(t)} \to \infty$ from Lem. F.8, then we also have

$$\sum_{a \in \mathcal{B}_0^h} \theta_{h,a}^{(t)} \to \infty.$$

□

We are now ready to complete the proof for Thm. 3.1. We prove it by showing that $\mathcal{I}_+^h$ is empty for all histories $h$ or, equivalently, $v_\psi^{(t)}(x_0) \to v_\psi^*(x_0)$ as $t \to \infty$.

**Theorem 3.1** (Global convergence). *Let rewards $R$ be bounded and let risk measure $\psi$ be preserving of boundedness. Assume the update in (12) is followed under a softmax policy with a positive linear isomorphism $\sigma$, and that every history $h$ is seen with positive probability. Then for all $h$, $v_\psi^{(t)}(h) \to v_\psi^*(h)$ as $t \to \infty$.*

*Proof.* Note that if $\mathcal{I}_+^h$ is empty then there are no actions with positive limiting advantage. The final policy then attains the maximum action value, so we are done. Now, suppose $\mathcal{I}_+^h$ is non-empty for some $h$. Let $a_+ \in \mathcal{I}_+^h$, then from Lem. F.11

$$\sum_{a \in \mathcal{B}_0^h(a_+)} \theta_{h,a}^{(t)} \to \infty.$$

Now, we will show a contradiction. Our strategy is to examine the contribution of each action subset to the identity $\sum_{a \in \mathcal{A}} \pi_\theta^{(t)}(a \mid h) A_\sigma^{(t)}(h,a) = 0$, which holds by definition of the advantage. We will show that, for large $t$, the contributions from $\mathcal{I}_-^h$ (where $\pi_\theta^{(t)}(a \mid h)$ vanishes relative to $\pi_\theta^{(t)}(a_+ \mid h)$) and from $\overline{\mathcal{B}}_0^h$ (where $A_\sigma^{(t)}(h,a) \to 0$) become negligible compared to the contribution of $a_+$, which is bounded below by $\Delta/4$. The zero-sum invariance then forces the contribution from $\mathcal{B}_0^h$ to be strictly negative, contradicting its divergence to $+\infty$ established directly above.

For $a \in \mathcal{I}_-^h$, we have

$$\frac{\pi_\theta^{(t)}(a \mid h)}{\pi_\theta^{(t)}(a_+ \mid h)} = \exp\left(\theta_{h,a}^{(t)} - \theta_{h,a_+}^{(t)}\right) \to 0,$$

since $\theta_{h,a}^{(t)} \to -\infty$ and $\theta_{h,a_+}^{(t)}$ is lower bounded by Lem. F.10. We can then select $T_2 > T_0$ such that

$$\frac{\pi_\theta^{(t)}(a \mid h)}{\pi_\theta^{(t)}(a_+ \mid h)} < \frac{\Delta}{16|\mathcal{A}|\Lambda},$$

or, equivalently

$$-\Lambda \pi_\theta^{(t)}(a \mid h) > -\frac{\pi_\theta^{(t)}(a_+ \mid h)\Delta}{16|\mathcal{A}|}$$

$$\Rightarrow -\sum_{a \in \mathcal{I}_-^h} \Lambda \pi_\theta^{(t)}(a \mid h) > -\sum_{a \in \mathcal{I}_-^h} \frac{\pi_\theta^{(t)}(a_+ \mid h)\Delta}{16|\mathcal{A}|} > -\sum_{a \in \mathcal{A}} \frac{\pi_\theta^{(t)}(a_+ \mid h)\Delta}{16|\mathcal{A}|}$$

Hence

$$-\sum_{a \in \mathcal{I}_-^h} \Lambda \pi_\theta^{(t)}(a \mid h) > -\frac{\pi_\theta^{(t)}(a_+ \mid h)\Delta}{16}. \tag{91}$$

For $a \in \overline{\mathcal{B}}_0^h$, we have $A_\sigma^{(t)}(h,a) \to 0$ by definition of $\mathcal{I}_0^h$ and $\overline{\mathcal{B}}_0^h \subset \mathcal{I}_0^h$. By Lem. F.9, we have for all $t > T_{a_+} : \pi_\theta^{(t)}(a_+ \mid h) > \pi_\theta^{(t)}(a \mid h) \Rightarrow 1 < \frac{\pi_\theta^{(t)}(a_+|h)}{\pi_\theta^{(t)}(a|h)}$. Hence, we can select $T_3 > T_2, T_{a_+}$ such that for $t > T_3$

$$|A_\sigma^{(t)}(h,a)| < \frac{\pi_\theta^{(t)}(a_+ \mid h)}{\pi_\theta^{(t)}(a \mid h)} \frac{\Delta}{16|\mathcal{A}|}$$

$$\Rightarrow \sum_{a \in \overline{\mathcal{B}}_0^h} \pi_\theta^{(t)}(a \mid h)|A_\sigma^{(t)}(h,a)| < \sum_{a \in \mathcal{A}} \pi_\theta^{(t)}(a_+ \mid h)\frac{\Delta}{16|\mathcal{A}|} < \pi_\theta^{(t)}(a_+ \mid h)\frac{\Delta}{16}$$

from which it follows that

$$-\pi_\theta^{(t)}(a_+ \mid h)\frac{\Delta}{16} < \sum_{a \in \overline{\mathcal{B}}_0} \pi_\theta^{(t)}(a \mid h)A_\sigma^{(t)}(h,a) < \pi_\theta^{(t)}(a_+ \mid h)\frac{\Delta}{16}. \tag{92}$$

Now, consider $t > T_3$. From $\sum_{a \in \mathcal{A}} \pi_\theta^{(t)}(a \mid h)A_\sigma^{(t)}(h,a) = 0$, we have

$$0 = \sum_{a \in \mathcal{I}_0^h} \pi^{(t)}(a \mid h)A_\sigma^{(t)}(h,a) + \sum_{a \in \mathcal{I}_+^h} \pi^{(t)}(a \mid h)A_\sigma^{(t)}(h,a) + \sum_{a \in \mathcal{I}_-^h} \pi^{(t)}(a \mid h)A_\sigma^{(t)}(h,a)$$

$$\overset{(a)}{\geq} \sum_{a \in \mathcal{B}_0^h} \pi^{(t)}(a \mid h)A_\sigma^{(t)}(h,a) + \sum_{a \in \overline{\mathcal{B}}_0^h} \pi^{(t)}(a \mid h)A_\sigma^{(t)}(h,a) + \pi^{(t)}(a_+ \mid h)A_\sigma^{(t)}(h,a) + \sum_{a \in \mathcal{I}_-^h} \pi^{(t)}(a \mid h)A_\sigma^{(t)}(h,a)$$

$$\overset{(b)}{\geq} \sum_{a \in \mathcal{B}_0^h} \pi^{(t)}(a \mid h)A_\sigma^{(t)}(h,a) + \sum_{a \in \overline{\mathcal{B}}_0^h} \pi^{(t)}(a \mid h)A_\sigma^{(t)}(h,a) + \pi^{(t)}(a_+ \mid h)\frac{\Delta}{4} - \sum_{a \in \mathcal{I}_-^h} \pi^{(t)}(a \mid h)\Lambda$$

$$\overset{(c)}{>} \sum_{a \in \mathcal{B}_0^h} \pi^{(t)}(a \mid h)A_\sigma^{(t)}(h,a) - \pi^{(t)}(a_+ \mid h)\frac{\Delta}{16} + \pi^{(t)}(a_+ \mid h)\frac{\Delta}{4} - \pi^{(t)}(a_+ \mid h)\frac{\Delta}{16}$$

$$> \sum_{a \in \mathcal{B}_0^h} \pi^{(t)}(a \mid h)A_\sigma^{(t)}(h,a)$$

where in step (a), we used the definition of $\mathcal{I}_0^h$, and $\mathcal{I}_+^h \neq 0$. In step (b), we used $A_\sigma^{(t)}(h, a_+) > \Delta/4$ for $t > T_3 > T_0$ from Lem. F.3 and the bound on $A_\sigma^{(t)}$. In step (c), we used (92) and (91). Together, these imply that $\sum_{a \in \mathcal{B}_0^h} \Delta_{\theta_{h,a}} v_\psi^{(t)} < 0$ for $t > T_3$ but this contradicts $\sum_{a \in \mathcal{B}_0^h} \theta_{h,a}^{(t)} \to \infty$ from Lem. F.8 which requires

$$\lim_{t \to \infty} \sum_{a \in \mathcal{B}_0^h} \left( \theta_{h,a}^{(t)} - \theta_{h,a}^{(T_3)} \right) = \lim_{k \to \infty} \xi \sum_{k=T_3}^{k} \sum_{a \in \mathcal{B}_0^h} \Delta_{\theta_{h,a}} v_\psi^{(t)} \to \infty.$$

We must accordingly conclude $\mathcal{I}_+^h$ to be empty, which concludes the proof. □

As a result, $\pi_\theta^{(t)}(a \mid h)$ converges to a distribution supported exclusively on $\mathcal{I}_0^h$. All actions $a \in \mathcal{I}_0^h$ share the same optimal risk-sensitive action-value $q_\psi(h, a)$ by definition of $\mathcal{I}_0^h$. We may therefore apply any systematic tie-breaking procedure to select among them during execution. This policy is still optimal in expected risk, and deterministic, therefore yielding an optimal stationary risk-sensitive policy by Prop. 3.3.

