# OpenReview forum: "RiskZero: Plan More to Risk Less with a Learned Model"
_ICML.cc/2026/Conference — ICML 2026 regular_

### Official Review · Reviewer_gi6K · 2026-03-02

**Soundness:** 3
**Presentation:** 3
**Significance:** 3
**Originality:** 3
**Overall Recommendation:** 5
**Confidence:** 2

**Summary:**

An-MCTS based approach for model-based RL (e.g. Alpha/MuZero) for standard risk-based optimization objectives (in contrast to the standard RL expected-return maximization objective).

**Compliance With Llm Reviewing Policy:**

Affirmed.

**Final Justification:**

I read the other reviews, and acknowledge the authors rebuttal.

I maintain my opinion that this is a good contribution paired with a good submssion which is ready to be published, and maintain my score of 5.

**Key Questions For Authors:**

1. Please see the weaknesses section.
2. It seems that RMZ consistently outperforms RAZ, even in the tabular environments, which I find curious (although not necessarily problematic). I'm curious how the authors understand this behavior.
3. Why are AZ/MZ not evaluated in Figures 1/2 B? I think it will be interesting to see the performance difference (even if A/MZ significantly outperform).
4. I'm not sure I understand the purpose of (and the conclusion from) sub-figures C in Figures 1 and 2. Could the authors explain what exactly is evaluated, and with what research question in mind?

**Limitations:**

1. I would like a more verbose discussion in the paper of the consequences for stochastic environments of the method being designed for determinsitic models.

**Strengths And Weaknesses:**

Strengths:
1. Complete contribution: clear motivation, theoretical analysis and support, method, sufficient experiments.
2. Clear presentation.
3. Underlying idea seems natural, sound, and is well explained.

Weaknesses:

1. Novelty in preliminaries (?): It's not clear to me if Proposition 2.1, which is presented in preliminaries, is novel and part of the contributions of the paper, or not. I believe that it is (point 1 in the introduction, line 77). If it is - novelty belongs in my opinion outside of the preliminaries, in the contribution section/s. If it is not novel, than it should be made clear and a citation should be added.

2. Usage of color: There is a rather significant usage of color. I'm in favor of using colors generally, but I find the presentation here a bit confusing:
    1. Line 075 marks values in red, but in figure 1 the return random variable is in red. Is red used for returns? values? both? other?
    2. In figure 1 there are checkmarks in green and orange. Checkmarks imply correctness, but the green (and the text) implies that only one of them is correct.
    3. The colors of the curves in figure 3 are hard to distinguish (especially the variations of pink). Perhaps dashed / regular lines can be used to seperate A/MZ agents, for example.

    Less colors and / or an explanation of the color scheme would help here, in my opinion.

4. Limitations: Is the method (/fundementally) limited to deterministic environments? Do the authors believe that it will be easy to extend the method to stochastic environments? (perhaps with standard StochasticMZ [1] machinary?) Limitation to deterministic environments is not a major issue, for me, and will not influence my score. However, I would like a more robust description of the limitations of the method to be made in the paper.

Overall, this is in my opinion a very good paper, the minor weaknesses above should be easy to address, and I would recommend the paper for acceptance.
However, my expertize is more in the areas of MCTS / AZ / MZ than risk-based RL, so my ability to review the theory was limited, and I adjust the confidence of the review accordingly.

---

> ### Author Rebuttal · Authors · 2026-03-31
>
> We thank the reviewer for the thoughtful feedback and recognition of our contributions. We address the questions below.
>
> > ### **Q1. RMZ vs RAZ performance**
>
> The only difference is that RMZ learns dynamics, while RAZ uses true dynamics. In principle, both should converge to the same optimum, with differences driven by exploration.
>
> We hypothesize RAZ becomes overconfident early and exploits suboptimal behavior, while RMZ’s learned dynamics introduce noise that sustains exploration and leads to better solutions once stabilized. As it is not a main focus of our method, we leave verification of this hypothesis to future work.
>
> We will briefly note this observation in the experimental discussion, as we agree it is both interesting and potentially informative to readers.
>
> > ### **Q2. Why AZ/MZ not shown**
>
> We believe this refers to Figures 3/4. Our goal was to show that naive risk-sensitive optimization can be biased and may fail for risk-sensitive objectives, while AZ/MZ are already known to perform well in risk-neutral settings. That said, we do agree that comparing both techniques for the risk-neutral case can be useful.
>
> We ran the additional experiments on MiniGrid, Space Invaders, and Graphs during the rebuttal period. AZ/MZ perform very similarly to RiskZero in the risk-neutral case. We cannot include the results here due to space, but will plot and discuss them in the revision.
>
> > ### **Q3. Purpose of sub-figures C**
>
> We believe the reviewer is again referring to Figures 3/4 and respond accordingly. The sub-figures C report risk-adjusted performance (0.25-CVaR) of the risk-neutral policies (1.0-CVaR) from sub-figures B.
>
> They show that risk-neutral policies are clearly suboptimal under risk-sensitive metrics. This validates the need for risk-sensitive optimization and highlights RiskZero’s gains. We will clarify this in the text.
>
> ## Weaknesses
>
> > ### **Q4. Limitation to deterministic environments**
>
> Please see our response to Reviewer 5Zbt (Q3), where we address this question. In short, the current theory assumes deterministic dynamics. Extending to stochastic settings requires other expectation-based statistics (e.g., [R1], [R2]) and, indeed, may be supported by Stochastic MuZero.
>
> As we note to Reviewer 5Zbt, we agree that a more detailed description of the limitations would be beneficial to readers and facilitate future work. We will incorporate this discussion into a clearer, more precise accounting in the limitations section, with additional details in an appendix.
>
> > ### **Q5. Novelty of Proposition 2.1**
>
> To our knowledge, the result and proof are novel. However, existing value-based approaches for CVaR and, as pointed out by reviewer UFgF, coherent distortion risk measures suggest that the result could be inferred as a corollary from their convergence guarantees, since these methods optimize over deterministic policies
>
> For clarity, we believe it would be more appropriate to retain Proposition 2.1 in the preliminaries as it underpins the value based methods we discuss there. However, to highlight our contribution, we will first clarify that prior work establishes convergence, and then that our contribution is a direct proof of existence of a deterministic optimum.
>
> > ### **Q6. Usage of color:**
> > ### **Q6.1 Line 075 marks values in red, but in figure 1 the return random variable is in red. Is red used for returns? values? both?**
>
> Red denotes the return distribution (Figure 1 caption). We will align Section 1 with the caption of Figure 1, which is more precise.
>
> > ### **Q6.2 In figure 1 there are checkmarks in green and orange. Checkmarks imply correctness, but the green (and the text) implies that only one of them is correct.**
>
> The intent here is to convey a ranking: green indicates the best, yellow second best, and red the worst. While the third option is optimal, the second may be selected by some algorithms based solely on future risk, even if suboptimal.
>
> To clarify, we will replace the yellow checkmark with a yellow X, indicating that the choice is suboptimal but may be selected if only future risk is considered. We will add emphasis in the caption to make this clearer as well.
>
> > ### **Q6.3 The colors of the curves in figure 3 are hard to distinguish**
>
> We appreciate the reviewer’s care for clarity and will adopt this suggestion. We will use a dashed line for the RAZ and RMZ agents to further emphasize that these are “our” methods.
>
> Again, we sincerely thank the reviewer for diligently and carefully engaging with our work; your feedback and suggestions directly improve the quality and clarity of our paper.
>
> ---
>
> [R1] Moghimi, Mehrdad, and Hyejin Ku. "Beyond CVaR: Leveraging Static Spectral Risk Measures for Enhanced Decision-Making in Distributional Reinforcement Learning." ICML 2025
>
> [R2] Bäuerle, N., & Ott, J. "Markov decision processes with average-value-at-risk criteria." Mathematical Methods of Operations Research (2011)

---

> > ### Author Rebuttal · Reviewer_gi6K · 2026-04-01
> >
> > I acknoweldge the rebuttal by the authors. My concerns have largely been addressed.
> > I've read the other reviews, and my opinion remains that this is a good paper that is ready to be published, and thus I maintain my score.
> >
> > I understand the tension between narrative and structure, but I maintain the opinion that readers should not expect novelty in preliminaries, by definition.
> >
> > I would request the authors choose one of two presentation paths:
> > 1. Present Proposition 2.1 as a corollary in the preliminaries, include a citation, and leave the narrative as is.
> > 2. Maintain that Proposition 2.1 is novel, present is as such, and move it from preliminaries to the contribution sections, and adjust the narrative accordingly.

---

> > > ### Author Response · Authors · 2026-04-08
> > >
> > > We thank the reviewer for engaging with our rebuttal and, once again, for their recognition of our contributions and their insightful feedback which we will incorporate to further improve our paper. We appreciate the reviewer’s emphasis on clarity and will adopt their second suggestion in our revision; we will adjust the narrative to move Proposition 2.1 into Section 3 rather than leaving it in the preliminaries.

---

### Official Review · Reviewer_i5wn · 2026-03-12

**Soundness:** 3
**Presentation:** 3
**Significance:** 3
**Originality:** 3
**Overall Recommendation:** 5
**Confidence:** 3

**Summary:**

The paper considers environments with deterministic state transitions and stochastic rewards, and aims to learn policies which maximize a risk sensitive ulity function instead of the expected return. They show that one can always find a deterministic policy which optimizes that utility, which together with the deterministic state transitions then implies that one can find a single trajectory which minimizes the risk sensitive utility, with the risk only coming from the stochastic rewards.

They then introduce RiskZero which combines the trajectory-level risk modeling with the planning and MCTS of MuZero. They then prove certain improvement and convergence guarantees. Then they specify how RiskZero is implemented on top of MuZero.

In the experiments, they evaluate it one Mini Grid toy example, in which RiskZero is doing better than MuZero and other baselines in terms of episode risk and sample efficiency. They then do the same in a modified Space Invaders environment from MinAtar, in which again RiskZero does best in terms of episode risk and sample efficiency. Finally, they evaluate RiskZero in Stochastic Bipartite Matching and Stochastic Maximum Independent Set, in which again RiskZero consistently outperforms competing methods.

**Compliance With Llm Reviewing Policy:**

Affirmed.

**Final Justification:**

The authors have addressed all the minor issues I raised, and have laid out clearly how why their current method is restricted to deterministic state transitions, and what the difficulties are with extending it to stochastic state transitions.

I keep my assessment that this paper is solid and relevant work, thus I am happy to keep my score at "5: Accept."

**Key Questions For Authors:**

My only real question is how the algorithm would have to be modified in the case of stochastic state transitions. But this has been addressed as a limitation and proposed as future work, so an answer to this question is not going to much impact my assessment of the paper.

**Limitations:**

Yes.

**Strengths And Weaknesses:**

Strenghts:
- The paper is clearly a result of high effort, and takes great care in describing its own methods properly and comparing against related work.
- Combining risk sensitivity with MCTS/MuZero type planning is clearly a relevant research problem.
- Their algorithm outperforms other algorithms they tested against both in terms of finding risk-sensitive policies and in terms of sample efficiency.


Weaknesses:
- The main weakness of the paper is that it resticts itself to deterministic state transitions, which then leaves the random rewards as the only source of risk. I guess that Proposition 3.2 is not true anymore for stochastic state transitions, which means that then one needs very different algorithms. In the context of risk management, assuming deterministic state transitions is a massive simplification which will not cover many interesting applications. It is noteworthy though that the authors are aware of this limitation, and they consider the extension to stochastic state transitions as future work.

Minor issues:
- Some wording around the distinction between optimizing for expected return vs optimizing for a certain risk utility is quite vague and a bit misleading. E.g. in line 40, they say that "Planning, in this sense, is not only about seeking reward but also about avoiding undesirable outcomes." This is a bit ambiguous, since when optimizing for reward, one is automatically also trying to avoid undesirable outcomes, as they are specified by a low reward.
- It is very unclear and confusing whether policies do condition on histories or on states. In line 66 policies are defined to condition on histories, without any mention yet why one would need that, given that one is in an MDP in which one typically only needs to condition on the current state. Then in line 128, for computational tractability one restricts attention to policies which condition on the state only. Then in line 138 the paper says that it build on TQL, which does condition on histories, but then in line 146 again the Bellman equation does not condition on histories. Then in  line 157 the there is again a history conditioned Bellman equation, which is followed by the contradictory phrasing "resulting in policies which are stationary w.r.t. history". Similarly in line 216 the contradictory term "history-stationary" policy is used.
- The definition of the quantile function in line 80 has a typo. It should be $F(z) >= w$ not $F(z) <= w$.
- coherent measures of risk are mentioned in line 077 but then they are never actually defined in the main paper, and the pointer towards the appendix only happens in line 99. Better sign-posting would be nice.
- $v_{\psi}^\pi(ha)$ in line 188 is never defined.
- Lemma 4.1 in line 270 is a well-known standard result which doesn't need to be stated and proven as a lemma.

---

> ### Author Rebuttal · Authors · 2026-03-31
>
> We thank the reviewer for their thoughtful and constructive feedback, and for recognizing the clarity, relevance, and empirical strength of our work. We address their suggestions below.
>
> > ### **Q1. Extension to stochastic state transitions**
>
> We agree this is an important direction and refer the reviewer to Q3 in our response to Reviewer 5Zbt.
>
> ## Minor Issues
>
> > ### **Q2. Wording for expected return vs optimizing for a certain risk utility is vague and misleading.**
>
> We thank the reviewer for this clarification and agree that the original wording may suggest a separation between reward maximization and avoidance of undesirable outcomes, when in fact the latter may sometimes be subsumed by the former.
>
> Our intent here is to emphasize that, under uncertainty, planning differs not in whether undesirable outcomes matter, but in _how much_ they matter. In particular, rare but severe failure may be tolerated under expectation maximization and require no planning. By contrast, a risk-averse actor may plan _more_ in such a situation to avoid such severe failure.
>
> In that sense, it is the degree of aversion to the undesirable that drives planning, and we want to motivate it. To that end, we will revise the sentence accordingly:
>
> _Planning, in this sense, is not only about maximizing expected reward, but also the degree to which one seeks the avoidance of undesirable outcomes._
>
> > ### **Q3. It is very unclear and confusing whether policies condition on histories or states.**
>
> We appreciate the reviewer for raising this point, which will help us improve clarity. The challenge is that we draw on background from multiple subfields of RL, some of which use policies that condition on histories, while others do not. We intended to follow the original formulations where possible (e.g., distributional RL does not condition on histories) to avoid confusion, but agree that the presentation can be clearer.
>
> To improve clarity, we will make the following revisions to our paper:
>
> In line 066, we will state explicitly that policies may depend on either states or histories, and briefly explain why and when the latter may be needed. In line 128, we reference a restriction to stationary policies; as noted in our response to Reviewer UFgF (Q2), this is not specifically with respect to states. Since 'stationary' is typically associated with states, we will revise this sentence first to emphasize an absence of conditioning on realized returns, which is a main source of tractability, and then note that this also coincides with a notion of stationarity. Accordingly, we do not view the use of 'history-stationary' as inherently contradictory with our interpretation of 'stationary', as noted in our response to Reviewer UFgF (Q2). Again, we will elaborate on our interpretation to clarify.
>
> For lines 138, 146, and 157, we will make the structure more explicit. We will first introduce TQL in the context of distributional RL, then define distributional RL with respect to states, and only afterward introduce TQL's conditioning on histories, continuing to use histories thereafter.
>
> > ### **Q4. Quantile definition typo**
>
> We sincerely appreciate the reviewer's attention to detail in reviewing our work; thank you for noticing this typo. We have made the correction.
>
> > ### **Q5. Better coherent measure sign-posting**
>
> We appreciate this feedback and agree that a change will improve clarity. We will remove the initial mention of coherent measures in line 077, and defer the only mention to line 099 with reference to the appendix. Due to space constraints, we do not formally define coherence in the main text and instead focus on the intuition that coherent measures satisfy natural axioms for risk-averse decision-making. The formal definitions are included in Appendix A.2 and support the proofs, but are not required for the main development.
>
> > ### **Q6.  $v_\psi^\pi(ha)$ in line 188 is never defined.**
>
> We define $v_\psi^\pi$ as a function of histories in Equation 8 (right-hand side), and note in line 067 that, under deterministic dynamics, we write $ha$ and $hax$ interchangeably, where it is implied that $x = M(h, a)$. Thus, $v_\psi^\pi(ha)$ intuitively denotes the value of the next history, i.e., $v_\psi^\pi(hax)$.
>
> We agree this is easy to miss, given the separation between these statements. To improve clarity, we will add a brief reminder of this interpretation for $ha$ immediately below Equation 8.
>
> > ### **Q7. Lemma 4.1 is well-known**
>
> We agree and will remove the Lemma from the main text. We will retain it in the appendix (with reference to the proof) for completeness, given its importance to our approach. It provides a quick, convenient reference for the connection to the quantile functions used in our formulation.
>
> ---
> We offer sincere thanks, once again, to the reviewer for their careful, precise reading of our work and for their suggestions. All of them directly improve the clarity of our presentation.

---

> > ### Author Rebuttal · Reviewer_i5wn · 2026-04-02
> >
> > The authors have addressed all the minor issues I raised, and have laid out clearly how why their current method is restricted to deterministic state transitions, and what the difficulties are with extending it to stochastic state transitions.
> >
> > I keep my assessment that this paper is solid and relevant work, thus I am happy to keep my score at "5: Accept."

---

> > > ### Author Response · Authors · 2026-04-08
> > >
> > > We thank the reviewer for engaging with our rebuttal and, once again, for their recognition of our work and their constructive and thoughtful feedback which we will incorporate to further improve our paper.

---

### Official Review · Reviewer_5Zbt · 2026-03-12

**Soundness:** 2
**Presentation:** 2
**Significance:** 2
**Originality:** 2
**Overall Recommendation:** 4
**Confidence:** 3

**Summary:**

The paper proposes RiskZero, a MuZero-style method for risk-sensitive reinforcement learning that plans using trajectory-level return distributions instead of expected return. The method combines historical return, reward, and future value distributions during search so that MCTS can optimize coherent risk measures such as CVaR. The paper also provides theoretical justification that optimizing an expected-risk surrogate over stationary policies can recover optimal deterministic risk-sensitive policies, and shows empirical improvements over prior risk-sensitive baselines on toy grid, MinAtar, and combinatorial tasks.

**Compliance With Llm Reviewing Policy:**

Affirmed.

**Final Justification:**

The rebuttal addressed my main concerns of the limitations of deterministic dynamics, discrete actions, and stationary policies.

**Key Questions For Authors:**

(1) What exactly is plotted as “episode risk” in the figures? Is this $\psi(Z)$, the expected-risk surrogate, or average episodic return under the distortion? Please clarify in the captions.

(2) How does the method handle model misspecification during rollouts? The algorithm seems to model return uncertainty but not epistemic uncertainty in the learned dynamics/value models. Why should the method remain reliable when the rollout model is biased?

(3) The theory assumes deterministic dynamics and stationary policies. How essential are these assumptions for the algorithm, and would the main results still hold in stochastic environments?

**Limitations:**

The current method is limited to deterministic dynamics, discrete actions, and stationary policies.
Robustness to model misspecification is not demonstrated empirically.
The experimental environments are relatively simple compared to the safety-motivated claims.

**Strengths And Weaknesses:**

Strengths:

(1) The paper addresses an important problem: incorporating trajectory-level risk into model-based planning.

(2) The algorithm is conceptually clean and well motivated, and the theory attempts to justify the surrogate objective used for planning.

(3) Experiments support the claim that combining historical and future returns improves risk-sensitive decision making.

Weaknesses:

(1) The setting is more restricted than the framing suggests (deterministic dynamics, stationary policies, discrete actions).

(2) The method models stochastic returns but does not clearly address model misspecification in learned rollouts.

(3) The evaluation metric (“episode risk”) is not clearly defined, which makes the plots harder to interpret.

---

> ### Author Rebuttal · Authors · 2026-03-31
>
> We thank the reviewer for the constructive feedback and for recognizing the importance and clarity of our work. We address the key points below.
>
> > ### **Q1. Definition of “episode risk” in plots**
>
> In all figures, _episode risk_ is $\psi[Z]$, computed empirically by rolling out the policy several times, collecting returns, and applying the distortion risk measure to the empirical distribution of returns. We will clarify this in the captions.
>
> > ### **Q2. Model misspecification**
>
> The reviewer is correct that RiskZero addresses aleatoric, not epistemic, uncertainty, and does not guarantee robustness under model bias. Consistent with prior RSRL and MuZero work, we did not make a claim about reliability under a biased rollout model. Handling misspecification would require additional machinery, orthogonal to our contribution. We will clarify this distinction in the paper and expand the discussion of the limitations (alongside existing pointers to relevant directions, such as RobustZero) to encourage future work in this area, which we agree is useful and interesting.
>
> > ### **Q3. Deterministic dynamics and stationarity**
>
> Though we see avenues to address them, both assumptions are used in the current formulation for different reasons.
>
> * **Stationarity** is to limit scope and increase tractability. In principle, one can augment the state with the discounted accumulated return and update it during search via reward sampling. Each node would then maintain a family of value distributions over an augmented state $(h, s)$, where $s$ is the realized return. This yields a policy that is stationary in the augmented space but non-stationary in the original one.
>
> * **Deterministic dynamics** are more central. Proposition 3.2 relies on the fact that, under deterministic transitions and convergence to deterministic $\pi^*$, the expectation is taken over a fixed trajectory, so expected risk equals true risk. This equivalence breaks under stochastic dynamics, where the gap is non-trivial and not easily bounded.
>
>     One alternative is to learn full, mixed trajectory distributions at each node, but this requires a compact, updatable representation and still does not resolve policy improvement: Proposition 3.1 (with a concrete counterexample in its proof in Appendix E) shows that improving risk is non-trivial even with full distributions.
>
> We therefore view expectation-based updates as the most practical for search. Extending to stochastic, non-stationary settings will likely require revising the theory while retaining expectation-based updates, as in prior work for non-stationary policies in stochastic settings [R1, R2]. We believe combining such statistics with Stochastic MuZero is a promising direction to extend RiskZero.
>
> An extension to continuous actions may be achieved via Sampled MuZero, without significant modifications otherwise. We consider discrete actions to limit scope.
>
> We thank the reviewer(s) for initiating this discussion and encouraging the authors to consider the limitations further. We will summarize this discussion in the limitations, and reference a more elaborate section added to the appendix, as it will be useful to readers.
>
> > ### **Q4. The experimental environments are relatively simple compared to the safety-motivated claims**
>
> We thank the reviewer for raising this point and agree that evaluating in more complex, safety-critical settings is an important direction. We also believe RiskZero makes progress in this direction.
>
> Our experimental design is guided by two goals: (i) validating the correctness of the trajectory-level formulation with expected risk, and (ii) benchmarking against prior risk-sensitive RL methods. The chosen environments reflect this:
>
> * **MiniGrid** enables controlled validation with a known structure and solution.
>
> * **Space Invaders** introduces scaling to image-based observations with stochastic rewards.
>
> * **Combinatorial tasks** stress scalability in large state/action spaces and connect to practical decision-making under uncertainty.
>
> To our knowledge, prior RSRL work has not considered stochastic combinatorial optimization. Our bipartite matching and MIS benchmarks are novel, large-scale, and directly connected to applications such as resource allocation, scheduling, and network design under uncertainty.
>
> We view this as a step toward more complex domains. The goal is not immediate deployment in safety-critical systems, rather establishing a scalable, principled foundation for risk-sensitive planning that can be built upon. We will clarify this positioning and emphasize scaling to richer domains as an interesting direction for future work.
>
> ---
>
> [R1] Moghimi, Mehrdad, and Hyejin Ku. "Beyond CVaR: Leveraging Static Spectral Risk Measures for Enhanced Decision-Making in Distributional Reinforcement Learning." ICML 2025
>
> [R2] Bäuerle, N., & Ott, J. "Markov decision processes with average-value-at-risk criteria." Mathematical Methods of Operations Research (2011)

---

> > ### Author Rebuttal · Reviewer_5Zbt · 2026-04-04
> >
> > Thank you for your detailed response. I don't have further questions. I've increased my score to 4.

---

> > > ### Author Response · Authors · 2026-04-08
> > >
> > > We thank the reviewer for engaging with our rebuttal and, once more, for their constructive feedback that will improve our paper. We are also appreciative for the increase in score.

---

### Official Review · Reviewer_UFgF · 2026-03-20

**Soundness:** 3
**Presentation:** 4
**Significance:** 2
**Originality:** 4
**Overall Recommendation:** 5
**Confidence:** 3

**Summary:**

The manuscript introduces RiskZero, the first risk-sensitive MuZero RL agent. It uses MuZero’s representation, state dynamics, and policy models without modification, but replaces reward-related (reward, value, and historical return) models with distributional counterparts to enable risk-sensitive planning. Trajectory-level risk is estimated via CVaR and by combining historical return with future risk, as considering future risk alone would bias the estimation. In addition to distributional modelling of reward-related quantities, RiskZero also introduces an additional loss term for training the new historical return model. RiskZero is evaluated on several stochastic benchmarks, demonstrating significantly better risk-sensitive policy learning while also converging faster in risk-neutral optimization. Ablation studies highlight the clear contribution of Gumbel planning, showing that it outperforms heuristic search with as little as two simulations and is less sensitive to the number of samples.

**Compliance With Llm Reviewing Policy:**

Affirmed.

**Final Justification:**

The authors provided a thorough rebuttal that put the work in context with a related work I had mentioned in my review. The rebuttal also clarified terminology and described the used hyperparameter selection strategy. I think raising the score from 4 to 5 is appropriate.

**Key Questions For Authors:**

1. SRM-DRL (Moghimi and Ku, 2025) seems to be a strong recent method for risk-sensitive RL. How does it compare to RiskZero? If this question is addressed sufficiently, I’d likely raise the significance rating.
2. What does “class history-stationary policies” (in Proposition 3.2) mean? I interpret it as if we see the same history at different times $t$ and $t’$, the policy produces the same action distribution for both. However, if the history includes every state and action from the initial state, I think there is no meaningful concept of equal histories at different times or does the history only include the most recent T steps?
3. How were the hyperparameters chosen?

**Limitations:**

yes

**Strengths And Weaknesses:**

## Soundness
I did not study all proofs in detail, but the propositions seem correct, and implementation choices seem appropriate. The benchmark results and ablation studies provide strong support for the presented claims.

I would appreciate a note on how hyperparameters for all models were chosen.

## Presentation
The paper seems polished and does a good job of communicating the subject matter including motivation, theory, and implementation. I particularly like the compact but easy to parse plots in the ablation studies.

I believe it would be helpful for readers to provide a few more cross-references to corresponding proofs/derivations in the appendix, e.g., for Proposition 2.1.

Minor note: In Algorithm 1->procedure GumbelMCTS: “simulation” should be changed to “Simulation”

## Significance
The contribution is addressing an important problem that is relevant to many real-world applications, where some actions can have disastrous consequences. As far as I can tell, it presents the first trajectory-level risk-sensitive MuZero variant and the authors present several worthwhile avenues for future research. The performance improvements, at least on the considered benchmarks, are considerable.

SRM-DRL (Moghimi and Ku, 2025) seems to be a strong recent method in distributional RL. How does RiskZero compare to it? (see Question 3). This would be relevant for judging whether RiskZero advances the state-of-the-art.

Moghimi, Mehrdad, and Hyejin Ku. "Beyond CVaR: Leveraging Static Spectral Risk Measures for Enhanced Decision-Making in Distributional Reinforcement Learning." International Conference on Machine Learning. PMLR, 2025.

## Originality
The contribution is original, integrating trajectory-level return estimation with Gumbel-based search in MuZero. The construction of this framework is well-motivated. The work introduces both a novel method and insights into effective strategies for risk-sensitive RL.

---

> ### Author Rebuttal · Authors · 2026-03-31
>
> We thank the reviewer for the thoughtful feedback and for recognizing the novelty, empirical strength, and clarity of our work. We address the main points below.
>
> > ### **Q1. Comparison to SRM-DRL**
>
> We appreciate the pointer to SRM-DRL, which is a relevant and recent work that we will explicitly discuss in the revision.
>
> SRM-DRL extends value-based CVaR-based methods (e.g., [R1]) by generalizing to spectral risk measures, improving convergence guarantees, and enabling non-stationary policies via state augmentation. We will reference these contributions in the related work and Appendix A.3 accordingly.
>
> We view SRM-DRL and RiskZero as addressing different regimes rather than directly competing:
> * **RiskZero** is **model-based**, introducing a tractable trajectory-level risk formulation compatible with planning, enabling efficient search and improved sample efficiency. As in prior work [R1–R3], we restrict to **(history-)stationary policies** for tractability.
> * **SRM-DRL** is **model-free** and targets **non-stationary policies** via a continuous risk budget and alternating optimization.
>
> These design choices lead to different trade-offs. State augmentation and alternating optimization can reduce sample efficiency. Consistent with this, SRM-DRL and related approaches evaluate in relatively low-dimensional environments (e.g., stock-trading environments where the state is simply the time index, and Lunar Lander with an 8D vector state).
>
> This motivates RiskZero: a planning-based approach that scales better to complex environments while retaining risk sensitivity. We will make this positioning explicit.
>
> At the same time, we believe the approaches are complementary. SRM-DRL suggests a path to extend RiskZero beyond stationary policies and deterministic dynamics. Its expectation-based updates may integrate with our framework, though this requires further analysis. If a sufficient statistic can be identified, implementation could follow via Stochastic MuZero [R4] and our extension for non-stationary policies (see response to Reviewer 5Zbt, Q3). We will expand this discussion in the limitations.
>
> We thank the reviewer again for highlighting this connection, which opens an important direction for future work.
>
> > ### **Q2. Meaning of “history-stationary policies”**
>
> We say a policy is _history-stationary_ if identical histories yield identical action distributions. In other words, **our use of _stationary_ is not tied to time**, but to invariance with respect to the input in general: identical inputs yield identical outputs.
>
> This distinction is important because more general formulations exist. Non-stationary methods (e.g., SRM-DRL) condition on additional variables (e.g., risk budget), so identical histories may still lead to different actions due to differing auxiliary inputs, such as different realized returns. While identical histories do not occur within a single trajectory, they can arise across different trajectories with different realized returns. In these cases, optimal behavior may differ, motivating non-stationary formulations.
>
> We agree that our terminology could be clearer. In the revision, we will:
>
> (i) explicitly define what we mean by stationary at first use (line 130),
>
> (ii) emphasize the distinction from time-based stationarity, and
>
> (iii) strengthen the reference to Appendix A.3, where Figure 6 provides a concrete example showing that history-stationary policies can be suboptimal.
>
> > ### **Q3. Hyperparameter selection**
>
> RiskZero introduces one additional hyperparameter relative to baselines: the number of quantile samples $D$. We fix $D = 1024$ as we find it sufficient (Fig. 5B).
>
> Other choices follow prior work:
>
> * 64 quantile atoms [R6]
>
> * Hidden size 256 following TQL (64 for graph settings [R5])
>
> * MuZero and Gumbel MuZero hyperparameters are taken directly.
>
> We use the largest feasible batch size and tune learning rate and replay buffer via small sweeps. We will add these details to Appendix C.4.
>
> > ### **Q4. Useful cross-references to corresponding proofs/derivations**
>
> We agree and will add explicit cross-references to the appendix for all relevant propositions and derivations.
>
> > ### **Q5. ‘simulation’ should be changed to 'Simulation’**
>
> Thank you for catching this. We have corrected “simulation” to “Simulation” in Algorithm 1.
>
> ---
> [R1] Lim, Shiau Hong, and Ilyas Malik. "Distributional reinforcement learning for risk-sensitive policies." NeurIPS 2022
>
> [R2] Tamar, Aviv, et al. "Policy gradient for coherent risk measures." NeurIPS 2015
>
> [R3] Keramati, Ramtin, et al. "Being optimistic to be conservative: Quickly learning a CVaR policy." AAAI 2020
>
> [R4] Antonoglou, Ioannis, et al. "Planning in stochastic environments with a learned model." ICLR 2021
>
> [R5] Abe, Kenshin, et al. "Solving np-hard problems on graphs with extended alphago zero." arXiV
>
> [R6] Dabney, Will, et al. "Implicit quantile networks for distributional reinforcement learning." NeurIPS 2019

---

> > ### Author Rebuttal · Reviewer_UFgF · 2026-04-04
> >
> > I thank the authors for the thorough rebuttal and clarifications regarding related work (SRM-DRL), terminology, and hyperparameter tuning. I will raise my score.

---

> > > ### Author Response · Authors · 2026-04-08
> > >
> > > We thank the reviewer for engaging with our rebuttal and, once again, for their insightful feedback which we will incorporate to further improve our paper. We also appreciate their recognition of our work, and increase in score.

---

### Decision · Program_Chairs · 2026-04-30

**Decision:**

Accept (regular)

**Comment:**

This paper presents RiskZero, an extension of MuZero for risk-sensitive decision-making. Standard MuZero relies on optimizing expected return, while the authors instead propose using trajectory-level return distributions. RiskZero leverages CVaR and combines historical returns, rewards, and future return distributions during search, allowing MCTS to optimize a risk-sensitive objective. The authors also provide a detailed theoretical analysis of the convergence of optimizing risk-sensitive policies. Experiments on Mini Grid, MinAtar, and combinatorial optimization tasks show that RiskZero achieves better risk-sensitive performance compared to existing baselines.

All reviewers agree that this paper is well-motivated and addresses an important problem in risk-sensitive environments, especially in model-based RL. Real-world decision-making is complex, so incorporating risk-aware evaluation is crucial. The idea of considering risk into MuZero-style planning is also novel, and the approach of using value distributions in MCTS is also well-designed and smoothly integrated. The proposed method is technically solid, with both theoretical justification and strong empirical results. Overall, I agree with the reviewers and recommend acceptance.